# Medical thinking with multiple images

**Zonghai Yao**[*1,2], **Benlu Wang**[*3], **Yifan Zhang** [2,4], **Junda Wang** [1,2], **Iris Xia** [3], **Zhipeng Tang** [1], **Shuo Han** [5], **Feiyun Ouyang** [2,4], **Zhichao Yang** [1], **Arman Cohan** [3], **Hong Yu** [1,2,4]

[1]Manning College of Information and Computer Sciences, UMass Amherst
[2]Center for Healthcare Organization and Implementation Research, VA Bedford Health Care
[3]Department of Computer Science, Yale University
[4]Miner School of Computer and Information Sciences, UMass Lowell
[5]Department of Electrical and Computer Engineering, UMass Lowell
zonghaiyao@umass.edu, benlu.wang@yale.edu

 **Code**: https://github.com/benluwang/MedThinkVQA
 **Leaderboard**: https://benluwang.github.io/MedThinkVQA/
 **Dataset**: https://huggingface.co/datasets/bio-nlp-umass/MedThinkVQA

## Abstract

Large language models perform well on many medical QA benchmarks, but real clinical reasoning is harder because diagnosis often requires integrating evidence across multiple images rather than interpreting a single view. We introduce Med-ThinkVQA, an expert-annotated benchmark for thinking with multiple images, in which models must interpret each image, combine cross-view evidence, and solve diagnostic questions under intermediate supervision and step-level evaluation. The dataset contains 8,067 cases, including 720 test cases, with an average of 6.62 images per case, substantially denser than prior work (earlier maxima $\leq 1.43$). On the test set, the best closed-source models, Claude-4.6-Opus, Gemini-3-Pro, and GPT-5.2-xhigh, achieve only 54.9%–57.2% accuracy, while smaller proprietary variants, GPT-5-mini/nano, drop to 39.7% and 30.8%. Top open-source models perform worse overall, with Qwen3.5-397B-A17B (52.2%) and Qwen3.5-27B (50.6%) leading, followed by Lingshu-32B (43.2%), InternVL3.5-38B (40.7%), and MedGemma-27B (31.8%). Further analysis points to a single bottleneck: current models struggle with grounded multi-image reasoning, i.e., reliably extracting, aligning, and composing evidence across views before higher-level inference can help. This is supported by three consistent findings: adding expert-provided single-image cues and integrating cross-image evidence improve performance, whereas replacing them with models' self-generated intermediates reduces accuracy. Step-level analysis shows that over 70% of errors come from image reading and cross-view integration, with reasoning failures increasing on decisive steps. Scaling results show that while accuracy increases with more images, additional inference-time computation is beneficial only when the underlying visual grounding is already reliable. When early evidence extraction is weak, longer reasoning yields limited or unstable gains and can even amplify misread cues. Together, these results show that the main barrier is not simply insufficient reasoning length or depth, but the lack of reliable mechanisms for grounding, aligning, and composing distributed evidence across real-world, cross-view, multimodal clinical inputs.

## 1 Introduction

Medical question answering has advanced quickly with large language models (LLMs) and vision-language models (VLMs) (Yan et al., 2023; Li et al., 2024; Chen et al., 2024b; Yao & Yu, 2025; Xie et al., 2024; Jiang et al., 2025). Yet recent evidence shows that strong final-answer accuracy in medical multimodal QA can still mask substantial failures in image understanding and inference (Yang et al., 2025; Jin et al., 2024; Bean et al., 2026; Asadi et al., 2026; Machcha et al., 2026). Many benchmark scores are now high, and several exam-style settings appear close to saturation (Jin et al., 2020;

---
*Equal contribution

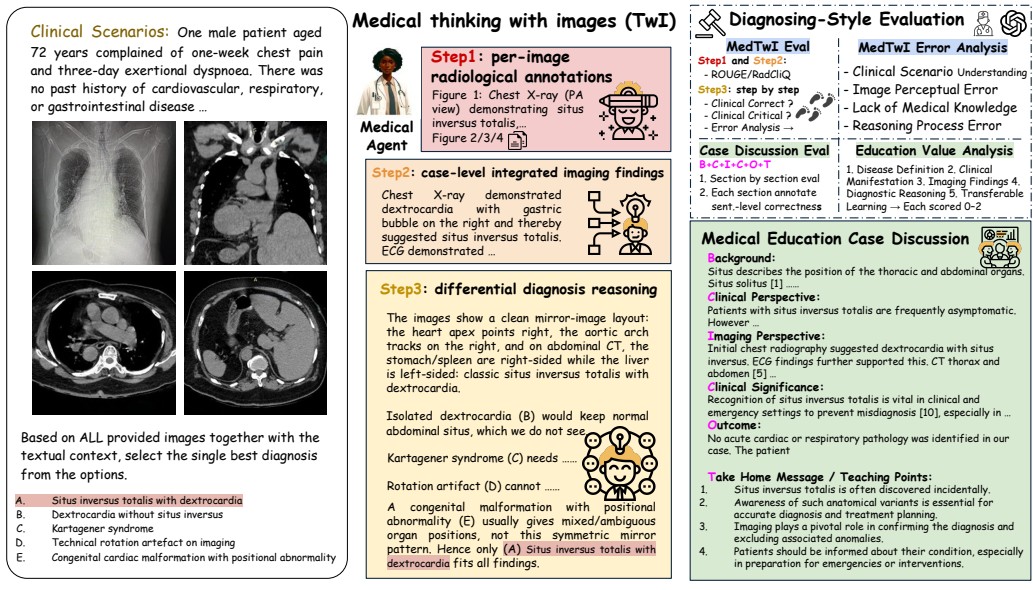

**Figure 1:** Medical Thinking with Images (TwI): task and diagnosing-style evaluation. Left: a sample case with a clinical scenario, multi-view images (e.g., radiograph + CT), and a five-option single-best-answer diagnosis. Middle: TwI's three supervised steps: (1) *Per-Image Findings* (detect and name key radiological signs for each image, expert-annotated, brief statements); (2) *Case-level Integrated Imaging Summary* (synthesize cross-view evidence into a single case summary); (3) *Differential-Diagnosis (DDx) reasoning* (align the summary with candidate diagnoses, rule out distractors with image-grounded arguments, and pick the most consistent answer). Right: Beyond-accuracy evaluation. Steps 1–2 use automatic metrics (ROUGE / RadCliQ), while Step 3 and the Medical Education Case Discussion are assessed with structured human- and LLM-judge rubrics that check clinical correctness and educational value, localizing failures in image reading, cross-view fusion, and teaching quality (see Section 3).

Hendrycks et al., 2020; Pal et al., 2022; Jin et al., 2019; Yao et al., 2026; Chen et al., 2024a; Gilson et al., 2023). But real diagnostic reasoning is usually not a single question over a single image Wang et al. (2024). In practice, clinicians read the clinical scenario, inspect several views, combine findings across images, and only then narrow the differential diagnosis. Figure 1 (left) illustrates this workflow. This gap matters because a model can perform well on standard medical QA while still failing at the harder step of grounding, aligning, and composing evidence across multiple images. We therefore need a benchmark that tests diagnosis in the way it is often performed, with multiple informative views and explicit intermediate reasoning.

MedThinkVQA is designed around this *think-with-multiple-images* setting. As shown in Fig. 1 (middle), the benchmark defines a three-step *think-with-images* (TwI) process: models first produce *per-image findings*, then synthesize them into a *case-level integrated imaging summary*, and finally perform *differential-diagnosis reasoning* to choose the best diagnosis. This structure makes the diagnostic process observable, instead of evaluating only the final answer. MedThinkVQA also includes a *Medical Education Case Discussion* task, where models generate teaching-style explanations grounded in the case. This better reflects how clinicians explain findings and share reasoning in practice. Figure 1 (right) further shows our diagnosing-style evaluation setup, which goes beyond answer accuracy to examine stepwise correctness, error types, and education value, so failures in image reading, cross-view fusion, and higher-level reasoning can be localized rather than hidden inside a single accuracy number.

Table 1 places MedThinkVQA in the context of prior multimodal medical QA datasets (Hu et al., 2024; Chen et al., 2024c; Zuo et al., 2025). Compared with earlier benchmarks, MedThinkVQA combines several properties that rarely appear together: expert annotation, real clinical scenarios, multi-modal imaging, longitudinal follow-up studies, intermediate supervision for think-with-images reasoning, and beyond-accuracy evaluation. To our knowledge, it is the only benchmark in Table 1 that satisfies all of these conditions. The dataset contains 8,067 cases, including 720 test cases, with

| Benchmark | # Case | Expert Annotation | Clinical Scenarios | # Img per Cas | Multi-Mod Imaging | Longitud Studies | Think-with-Images Intermediate Signals | Beyond-ACC Evaluation |
|---|---|---|---|---|---|---|---|---|
| VQA-Rad Lau et al. | 451 | ✗ | ✗ | 0.45 | ✗ | ✗ | ✗ | ✗ |
| VQA-Med Ben Abacha et al. | 500 | ✗ | ✗ | 1.00 | ✗ | ✗ | ✗ | ✗ |
| Path-VQA He et al. | 6,719 | ✗ | ✗ | 0.13 | ✗ | ✗ | ✗ | ✗ |
| SLAKE-En Liu et al. | 1,061 | ✗ | ✗ | 0.09 | ✗ | ✗ | ✗ | ✗ |
| PMC-VQA Zhang et al. | 33,430 | ✗ | ✗ | 0.87 | ✗ | ✗ | ✗ | ✗ |
| OmniMedVQA Hu et al. | 127,995 | ✗ | ✗ | 0.92 | ✗ | ✗ | ✗ | ✗ |
| GMAI-MMBench Chen et al. | 21,281 | ✗ | ✗ | 1.00 | ✗ | ✗ | ✗ | ✗ |
| GEMeX Liu et al. | 1,605,575 | ✗ | ✗ | 1.00 | ✗ | ✗ | ✓ | ✓ |
| Medical-Diff-VQA Hu et al. | 700,703 | ✗ | ✗ | 1.23 | ✗ | ✓ | ✗ | ✗ |
| MedFrameQA Yu et al. | 2,851 | ✗ | ✗ | 3.24 | ✗ | ✗ | ✓ | ✗ |
| ICG-CXR Ma et al. | 11,439 | ✗ | ✗ | 2.00 | ✗ | ✓ | ✗ | ✗ |
| MedRAX[1] Fallahpour et al. | 2,500 | ✗ | ✗ | 1.85 | ✗ | ✗ | ✗ | ✗ |
| GEMeX-ThinkVG Liu et al. | 206,071 | ✗ | ✗ | 1.00 | ✗ | ✗ | ✓ | ✓ |
| MMMU (H & M) Yue et al. | 1,752 | ✓ | ✗ | 1.14 | ✗ | ✗ | ✗ | ✗ |
| MMMU-Pro (H & M) Yue et al. | 346 | ✓ | ✗ | 1.25 | ✗ | ✗ | ✗ | ✗ |
| S-Chain Le-Duc et al. | 12,000 | ✓ | ✗ | 1.00 | ✗ | ✗ | ✓ | ✓ |
| MedXpertQA MM Zuo et al. | 2,000 | ✓ | ✓ | 1.43 | ✗ | ✗ | ✗ | ✗ |
| **MedThinkVQA** | **8,067** | ✓ | ✓ | **6.62** | ✓ | ✓ | ✓ | ✓ |

**Table 1: Comparisons with multimodal medical QA benchmarks.** *MedThinkVQA* is expert-annotated, averages **6.62** images per case (prior maxima $\leq 1.43$; $\geq 4.5\times$ more), and is the largest corpus at the expert-annotation level; a checkmark in the *Expert Annotation* column indicates that items are curated and labeled by clinicians rather than automatically generated. **Clinical Scenarios.** Prior work lacks broad, fine-grained coverage of real diagnostic scenarios; only *MedThinkVQA* and *MedXpertQA-MM* include scenario labels. **Multi-Modal Imaging / Longitudinal Studies.** We mark *Multi-Modal Imaging* when at least some questions require joint interpretation of images from multiple distinct medical imaging modalities for the same case (e.g., radiograph+CT), and *Longitudinal Studies* when questions are built from follow-up imaging of the same patient at different time points (e.g., baseline vs follow-up studies). **Think-with-Images Intermediate Signals.** This merged column indicates whether a benchmark provides intermediate supervision for think-with-images reasoning, including *per-image findings*, a *case-level imaging summary*, and a *case discussion* (teaching note). **Beyond-ACC Evaluation.** Leveraging these signals, only *MedThinkVQA* supports fine-grained, end-to-end assessment of think-with-images reasoning and teaching discussions: stepwise checks, error-type tags, education-value scoring, and automatic intermediate metrics, rather than accuracy alone.

an average of 6.62 images per case. Prior expert-level medical VQA benchmarks use far fewer images per case, with a previous maximum of at most 1.43. This difference is not only a matter of scale. It changes the task itself, from recognizing clues in one image to integrating distributed evidence across views, modalities, and time. We further design MedThinkVQA so that questions depend on imaging evidence rather than textual shortcuts; Section 2 details the ICD–10 coverage, rare-disease cases, and the filtering and option policies used to control distractors, leakage, and surface biases.

Figure 3 shows that this setting remains difficult even for the strongest current models. On the test split, the best closed-source systems, Claude-4.6-Opus, Gemini-3-Pro, and GPT-5.2-xhigh, reach only 57.2%, 55.3%, and 54.9% accuracy, respectively. Strong open-source models trail behind, led by Qwen3.5-397B-A17B at 52.2%. These results remain far below both the strongest result reported on the hardest prior benchmark of a related kind and clinician performance on our manually reviewed subset, indicating substantial headroom. The lower panels of Fig. 3 further clarify the source of the difficulty. First, accuracy rises monotonically as more images are provided, which shows that additional views contain real information gain and that the benchmark truly rewards using more visual evidence. Second, accuracy also tends to increase with greater reasoning effort, measured by output tokens. Third, within both Dense and Mixture-of-Experts (MoE) model families, larger models generally perform better, but their ability to benefit from extra reasoning budget differs: the medium-to-large ($> 2B$) Qwen3.5 family translates additional reasoning more effectively as model size grows, whereas Qwen3-VL and small-scale Qwen3.5 models show limited and unstable returns. Together, these trends point to a central conclusion: longer reasoning can help, but only after the model has already extracted and aligned the right visual evidence.

This conclusion is reinforced by our intermediate-signal and error analyses. Providing *expert* per-image findings and case summaries improves final accuracy, while replacing them with models' *self-generated* intermediates yields only small gains or even hurts performance. This means the main bottleneck is not simply the last reasoning step, but the earlier stages of grounded visual understanding and cross-image composition. The expert audit shows that 88.05% of images are supportive for the final diagnosis, the test set contains about 2.30 imaging modalities per case on

average, and 30.4% of test cases involve longitudinal follow-up studies. In other words, most cases require aggregating multiple useful views rather than finding one decisive image. Our step-level analysis supports the same story: across 202 labeled steps, 44 contain non-empty error tags; among these error-bearing steps, 77.27% involve image-understanding errors and 22.73% involve reasoning errors, while medical-knowledge and scenario-setup errors are much rarer. Even within *Critical* error-bearing steps, image understanding remains dominant. These findings align with the abstract and with Fig. 3: current models still struggle most with reliably extracting, aligning, and composing evidence across views before higher-level inference can fully help.

**Contributions.** (1) We introduce MedThinkVQA, a benchmark for *multi-image* diagnostic reasoning that supervises three explicit think-with-images steps. (2) We provide a *beyond-accuracy* evaluation suite, including automatic intermediate metrics, structured error-type tagging, and education-value scoring, together with scoring scripts and output formats. (3) We release a large, image-dense, expert-annotated corpus (8,067 cases, 6.62 images per case) that, to our knowledge, is the only benchmark that checks all columns in Table 1. (4) We present evidence that the main bottleneck of current medical VLMs is not raw reasoning length alone, but reliable cross-image evidence extraction and integration.

## 2 MEDTHINKVQA

### 2.1 SOURCE CORPUS

MedThinkVQA is adapted from *Eurorad*, a peer-reviewed online database of radiology teaching cases curated by the European Society of Radiology (eur). The corpus covers major subspecialties (neuro, musculoskeletal, thoracic, abdominal, pediatric, etc.) and common imaging modalities (X-ray, CT, MRI, ultrasound, etc.). Each case includes: (i) a brief clinical history; (ii) a multi-image set (average 6.62 images per case); (iii) radiologist-annotated, per-image hints; (iv) a case-level *Integrated Imaging Summary* section; (v) an *Expert Reasoning & Teaching Note* that interprets the findings, highlights key diagnostic reasoning, and links to clinical relevance; (vi) the final diagnosis; and (vii) a differential-diagnosis list.

Cases are contributed by radiologists and researchers worldwide, typically based on real clinical examinations. Submissions are reviewed by the Eurorad Editorial Board (radiology experts) before publication to ensure authenticity and educational value (eur). We collected **8,067** cases and curated them into MedThinkVQA. After post-processing, we formed a held-out test set with **720** cases. These cases span 9 major imaging modalities in the test split and typically involve more than two distinct modalities per case; detailed modality statistics are provided in Appendix N. For concreteness, detailed field-to-annotation examples and six representative Eurorad case studies are provided in Appendix G. Details of the MCQ transformation and option policy are provided in Section 2.3.

Eurorad materials use CC BY-NC-SA 4.0; MedThinkVQA follows the same license and is for research and education only, with attribution and ShareAlike, and no commercial use. We worked with Eurorad and use the materials with permission. Cases are de-identified to the best of our knowledge; we did not collect new personal data; IRB review was not required; we remove items if residual identifiers are suspected. The benchmark is not a clinical device and must not be used for diagnosis, treatment, or triage. To lower leakage risk, we release collection and filtering scripts, run de-duplication, and drop items that text-only models can solve; we also keep a path to refresh held-out items.

### 2.2 DATASET COVERAGE

**Task framing.** We characterize dataset coverage along two orthogonal axes: (i) a *disease* axis using ICD–10 chapters, and (ii) a *radiology/medical imaging* axis grouped by anatomy and subspecialty. The ICD–10 taxonomy contains 22 chapters. Using GPT-5.2 to map case labels to ICD–10, our test set covers 308 categories, providing coverage of long-tail conditions. A complete breakdown of ICD–10 chapters and subcategories is reported in the Appendix S.

To assess breadth from an imaging perspective, we aggregated the *full dataset* by radiology subspecialties (*anatomy & subspecialty*). Figure 2 shows the distribution. The cases are not concentrated in a single region but span across all major clinical domains. The largest share comes from *abdominal imaging* (22.0%), followed by *neuroradiology* (16.0%) and *musculoskeletal* (14.3%). Mid-sized categories include chest (9.6%), paediatric (8.0%), and urogenital imaging (7.5%), while cardiovascular

(6.7%) and head & neck (5.7%) also make substantive contributions. Smaller but non-negligible proportions are represented in breast and interventional radiology, with hybrid imaging appearing only rarely (<0.1%). From a temporal-structure perspective, roughly one third of MedThinkVQA cases are longitudinal follow-up studies (multiple time points for the same patient), so temporal disease evolution is explicitly represented; detailed longitudinal statistics are summarized in Appendix O.

## 2.3 MCQ Conversion and Option Policy

Each case is presented as a five-choice single-best-answer MCQ: Given the clinical history and associated radiology images, select the most likely diagnosis from the options. The ground-truth label is the case's *final diagnosis*. While only the *clinical history* and *images* are provided as input context for the QA task, we also retain other curated textual fields (expert caption, Integrated Imaging Summary, and Expert Reasoning & Teaching Note) in the dataset files for potential future use. If the source differential diagnosis list has ≥ 5 candidates, we *prune*

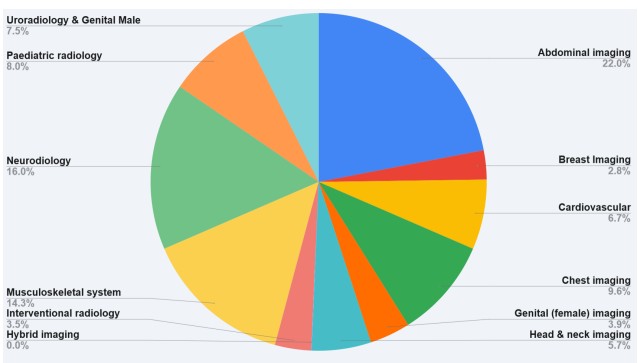

**Figure 2:** Distribution of radiology imaging main categories

to five using a confusion-aware ranking (keep the correct answer plus four distractors that models most often confuse with the truth). Duplicates or contradictions are rejected. Unlike prior LLM-based medical MCQ generation work (Yao et al., 2025), our test options are not freely generated from scratch. Instead, they remain anchored in expert differential diagnoses and are only pruned to five choices, which helps preserve clinical plausibility while reducing synthetic drift. Key held-out test statistics are summarized in Table 2.

## 2.4 Test Set

We design the test split to stay as close as possible to expert reasoning and image-based decision making. *The construction of the training set is described in Appendix B.*

**(1) Expert differential diagnosis as starting point.** We first use cases where the *expert differential* list has ≥ 5 entries. The final diagnosis serves as the key, and the differential entries form the distractor pool. This ensures all candidate options come directly from experts and filters 2,996 data points.

**(2) Leakage Detection.** To ensure the rigor of the dataset, we conduct leakage detection on each clinical history to verify whether it directly reveals the correct diagnosis. Specifically, we examine whether (i) the diagnosis label itself (exact name or ICD-standard term) appears in the text, (ii) synonyms, abbreviations, or eponyms are explicitly present, or (iii) uncertain mentions of the label or its variants occur (e.g., "?X," "rule out X," "suspected X," "possible X"). The detailed prompt used for this detection is provided in Appendix J. In total, 137 leaked cases are identified and removed from the dataset.

**(3) Confusion-aware pruning.** Moreover, if there are more than five distractors, we check which wrong answers the reasoning VLM (`o4-mini`) picks mistakenly. We keep these frequently confused distractors when possible and sample the rest at random. Only deletions are made; the original Expert Reasoning & Teaching Note is lightly edited (via GPT-5-mini) to remove references to deleted options (Appendix I). No new medical content is introduced.

**(4) Remove text-solvable cases.** To ensure that images are genuinely necessary for solving the task, we conduct a two-stage text-only filtering procedure.

| Overall | |
| --- | --- |
| Samples | 720 |
| Images | 5845 |
| **Per-sample** | |
| Imgs/sample | 8.12 |
| Cap. length | 1338.6 |
| Find. length | 875.8 |
| Disc. length | 2732.8 |
| **Option Length** | |
| Avg. | 27.5 |
| **Num. Density** | |
| Macro avg. | 0.0093 |
| **Other** | |
| Pos. correct | 2.84 |
| Mean mod. cnt | 2.30 |
| All mod. types | 9 |
| Longit. share (%) | 30.4 |

**Table 2:** Test stats (*Cap/Find/Disc.* = caption, findings, discussion; *Pos. correct* = avg. position of correct option; *Mean mod. cnt* = mean # of imaging modalities; see Appendix N for modality statistics and Appendix O for longitudinal-study statistics.

First, we evaluate each provisional item using four large-scale *text-only* models—Qwen3-Next-80B, GPT-oss-120B, MedGemma-27B-text-it, and Llama-3.3-70B. Any item that all four models answer correctly is removed. This step filters out questions that can be reliably solved from the textual context alone, retaining only cases where imaging information is essential or plays a substantial role. This stage removes 1,074 cases.

Second, to further eliminate potentially text-solvable samples even after supervised fine-tuning (SFT), we test with four SFT LLMs (Qwen2.5-3B, Mistral-7B-v0.2, Qwen3-4B, and Llama-3.1-8B). We remove cases in which all four models answer correctly. This stricter criterion ensures that even smaller-capacity models cannot consistently solve the question without visual input. This additional filtering step removes 180 cases.

**(5) Surface Bias Mitigation** We observed a systematic surface bias in option length: in over half of the cases, the correct answer was the longest choice, far above the uniform expectation of 20% under a five-option setting. This bias likely arises because correct diagnoses tend to be phrased more specifically and with greater clinical detail for a given patient, whereas distractors are shorter and more generic. Importantly, models achieved 5–10 points higher accuracy on such items, suggesting partial reliance on superficial length heuristics rather than genuine multimodal reasoning. To prevent shortcut learning, we pruned items accordingly.

In addition, to reduce dataset-level structural skew, we further rebalanced the distribution of ICD disease categories and imaging modalities. We removed additional samples to improve the balance of labels and modalities across the dataset. In total, this stage removes 706 cases, improving both surface-level fairness and macro-level distributional balance.

## 2.5 MEDICAL EDUCATION CASE DISCUSSION

In clinical practice, difficult or representative cases are often documented as teaching notes and shared with trainees and colleagues, and the Eurorad "Discussion" sections already serve this role. The human-expert study in Appendix C further shows that even experienced clinicians find a subset of MedThinkVQA cases very difficult, motivating our Medical Education Case Discussion task, in which models are asked to generate structured teaching content rather than predict a single diagnosis. To make this evaluation well defined, we focus on cases whose discussions follow a clear five-section template, namely *Background*, *Clinical Perspective*, *Imaging Perspective*, *Outcome*, and *Take-Home Messages*, yielding a subset of test cases that strictly conforms to this structure and supports section-by-section comparison. Details about Case Discussion Automatic/Human Evaluation and Results can be found at Appendix E.

## 3 EXPERIMENTAL SETUP

### 3.1 MODEL BASELINE

We establish baselines using a diverse set of VLMs to ensure fair and representative evaluation. The selection spans both *Inference-Time Scaled Large Multimodal Models* (e.g., GPT-5.2 and Gemini-3-Pro) and *Vanilla Large Multimodal Models*, which include open-weight generalist and medical-tuned families such as InternVL family, MedGemma, and Phi, at different parameter scales (2B–397B).

### 3.2 AUTOMATIC EVALUATION

**1. Intermediate imaging metrics** For the per-image findings and the case-level integrated imaging summary (Steps 1–2 in Fig. 1), we follow recent radiology-report evaluation work (Yu et al., 2023; Ostmeier et al., 2024) and compute ROUGE as a lexical-overlap baseline together with RadCliQ, which correlates more strongly with radiologist preferences. We apply these metrics to compare model outputs against the expert-written findings and summaries, providing automatic, fine-grained signals for how well models capture clinically salient details.

**2. Stepwise Reasoning Evaluation** We split each model explanation into atomic steps using GPT-5-MINI, then used GPT-5 as an LLM judge to label each step: factual correctness, whether it is *critical* to the final diagnosis, and, if incorrect, an error type. This design is also motivated by recent evidence that answer-only scoring in medicine can hide clinically meaningful reasoning failures, making step-level auditing and explicit error attribution important (Wang et al., 2025a). When a step is incorrect, the judge assigns one of four error types: clinical-scenario misunderstanding, missing or misread

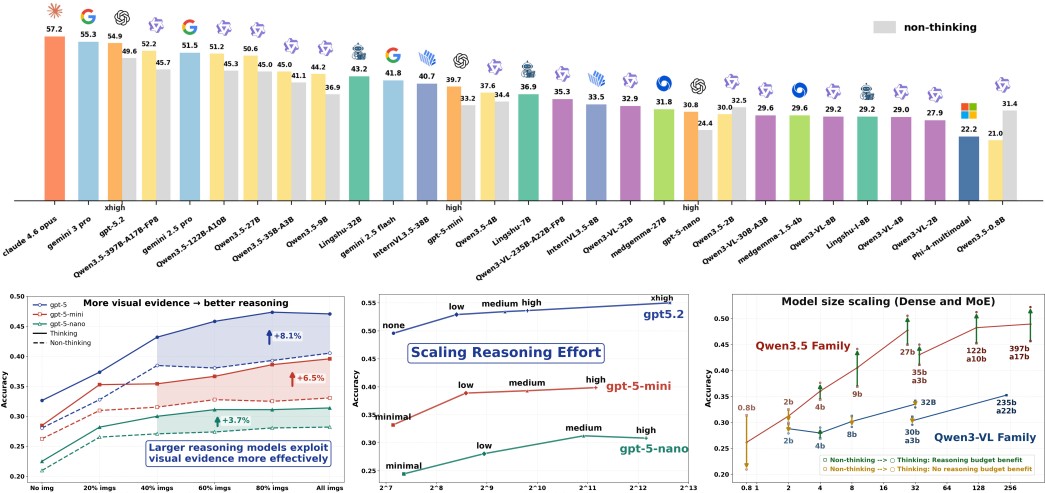

**Figure 3: MedThinkVQA benchmark results on the test set**, highlighting the core challenge of thinking with multiple images: performance is limited less by raw reasoning length alone, and more by whether a model can reliably ground, align, and compose evidence across views. **Top:** Overall test accuracy (%). When available, *paired bars* report the same model in thinking and matched non-thinking mode (gray); *single bars* report the thinking result when such a mode exists, otherwise the standard configuration. Detailed results are provided in Appendix D. **Bottom: (left)** Masking a fraction of images shows stable gains as more visual evidence is provided, indicating that the benchmark contains real cross-image information gain. However, larger reasoning models benefit more, suggesting that success depends on how well additional images are actually used, rather than simply being available. The gains of thinking over non-thinking also peak at +8.1/+6.5/+3.7 points for GPT-5/mini/nano as more visual evidence becomes available. **(middle)** Accuracy generally increases with greater reasoning effort, as measured by average output tokens, indicating a clear scaling trend. **(right)** Both Dense and MoE LLMs within each family broadly follow model-size scaling, but their ability to convert extra reasoning budget into better answers differs. Within the medium-to-large (> 2B) Qwen3.5 family, reasoning gains increase as models scale, whereas Qwen3-VL and small-scale Qwen3.5 models show limited and unstable returns, suggesting that extra inference-time computation helps mainly when early visual evidence extraction is already reliable and cannot reliably compensate for weak multi-image grounding.

image evidence (*Image Understanding Err*), medical knowledge error, or flawed reasoning; Table 3 summarizes the aggregate coverage of these labels, and this taxonomy is reused for both automatic and human evaluations. Overall, most failures stem from image misinterpretation/information extraction, especially on *critical* steps (69.23%). When answers are wrong, *Reasoning Err* and *Medical Knowledge Err* become more prominent alongside the image errors (details in Appendix L and Appendix Q).

### 3.3 HUMAN EVALUATION

Two medical experts did a Stepwise Reasoning Evaluation on 50 cases (202 steps) for step factuality and error types. In total, 44 steps contained errors (21.78%), with *Image Understanding Err* dominant (77.27%), followed by *Reasoning Err*, supporting the automatic evaluation conclusion that image misinterpretation is the primary source of mistakes. Inter-rater agreement was high: Cohen's $\kappa = 0.82$ between the two experts, and human–LLM-judge agreement ranging from $\kappa = 0.70$ to $\kappa = 0.84$, confirming the reliability of the automatic judge.

## 4 RESULTS AND DISCUSSION

### 4.1 MAIN RESULTS

Figure 3 reports the overall MedThinkVQA leaderboard. Performance is still limited across all models. The best result is 57.2% from Claude-4.6-Opus, followed by Gemini-3-Pro at 55.3% and GPT-5.2 at 54.9% in xhigh thinking mode. Strong open models are lower, ranging from the low-50s to the low-20s, including the Qwen3.5 family (21–52.2%), InternVL3.5-38B (40.7%), Qwen3-VL-32B

(32.9%), MedGemma-27B (31.8%), and Phi-4 (22.2%). Detailed results are provided in Appendix D. Overall, these results show that *think-with-multiple-images* diagnosis remains difficult even for current top multimodal systems.

Inference-time *thinking* helps, but the gains are moderate. For models with matched non-thinking variants (gray bars in Fig. 3, top), thinking usually improves accuracy by about 5–7 points. For example, GPT-5.2 increases from 49.9 to 54.9, GPT-5-mini from 33.2 to 39.7, and GPT-5-nano from 24.4 to 30.8. Similar improvements also appear in the Qwen3.5 MoE models. This shows that a larger reasoning budget is useful, but it does not eliminate the task's main difficulty.

Figure 5 makes this bottleneck clearer. It evaluates several representative models under controlled input settings. When models rely solely on images, performance remains limited. When *expert-written* imaging text is added, accuracy improves substantially across models. This suggests that the main difficulty lies in extracting and combining diagnostic evidence from multiple images, rather than in reasoning over text once the key evidence is already provided.

## 4.2 IMAGE REASONING CAPABILITIES

**Expert imaging summaries sharply boost accuracy.** Across all four models, providing expert-written text extracted from images leads to large gains on MedThinkVQA (Fig. 5). Feeding the `Integrated Imaging Summary (expert)` raises accuracy by +41.5, +50.5, +41.5, and +42.0 points for MedGemma-

| Error type | All error steps (N=1509) | Critical error steps (N=182) |
|---|---|---|
| Image Understanding Err | 959 (63.55%) | 126 (69.23%) |
| Reasoning Err | 583 (38.63%) | 71 (39.01%) |
| Medical Knowledge Err | 362 (23.99%) | 60 (32.97%) |
| Clinical Scenario Err | 191 (12.66%) | 22 (12.09%) |

**Table 3:** LLM-judge error-type coverage. *Note:* categories are multi-label; percentages are step-level coverage over error steps and may sum to >100%. Full per-split (answer-correct vs. wrong) breakdowns are in the Appendix.

27B, GPT-5-nano, GPT-5-mini, and GPT-5, respectively, corresponding to 2.19×, 2.60×, 1.95×, and 1.92× their baselines. Once this diagnosis-oriented summary is available, adding an `Image Hint (expert)` provides only modest extra gains (+0.5–5.0 points), and the summary consistently outperforms the hint alone by +8.0–15.0 points. These patterns indicate that *structured findings matter more than caption-like descriptions*: the summary encodes laterality, location, pattern, and extent in a way that provides highly discriminative cues for differential diagnosis. The relative gains are larger for weaker baselines (e.g., GPT-5-nano 2.60× vs. GPT-5 1.92×), suggesting that once visual evidence is *correctly verbalized*, the remaining language inference is largely adequate—and the core obstacle is *extracting and structuring pixel-level radiological evidence* from multi-view inputs.

**Self-generated text is fragile and consistently hurts.** When models first write their own `Hint`/`Summary` and then condition on it, performance *consistently drops* across all models and all self-generated settings (Fig. 5). All four models underperform their baselines: MedGemma-27B decreases by 3.5 points with self-hints and by 12.5 points with self-summaries/`Both`; GPT-5-nano decreases by 3.0–6.0 points; GPT-5-mini decreases by 6.5–12.0 points; and GPT-5 decreases by 7.0–9.5 points. Tab. 7 explains why: Image→Caption/Findings generations achieve low ROUGE-L ($\approx 0.13$–$0.16$) and imperfect RadCliQ-v1 scores, meaning that self-produced descriptions often miss laterality, precise locations, or key patterns and may introduce subtle inaccuracies. In addition, noisy text increases sequence length and can dilute attention over multi-view inputs, and current VLMs may over-trust erroneous text when image–text grounding is weak.

## 4.3 SCALING ANALYSES: VISUAL EVIDENCE AND REASONING EFFORT

**Larger reasoning models exploit visual evidence more effectively, but the payoff is family-dependent.** Figure 3 (bottom-left) ablates visual evidence by masking a fraction of images. Accuracy improves *monotonically* as more images are revealed, confirming that MedThinkVQA genuinely rewards multi-view evidence aggregation rather than single-image shortcuts. Importantly, the slope of this improvement is steeper for larger reasoning models: GPT-5 benefits more from increasing image ratios than GPT-5-mini and GPT-5-nano, suggesting a stronger capability to *select, align, and fuse* cross-view signals. Moreover, the advantage of *thinking* grows as visual evidence increases and

peaks at high image ratios, reaching **+8.1/+6.5/+3.7** points for GPT-5/mini/nano, respectively. This interaction indicates that additional reasoning budget is most useful when there is sufficient visual evidence to integrate and when the model can reliably ground its intermediate steps in that evidence.

The same principle helps interpret the contrasting behavior of Qwen families in Fig. 3 (bottom-right). Across both Dense and MoE variants, performance broadly follows model-size scaling, yet the *marginal return* of extra reasoning differs sharply by family and model performance: **scaling the reasoning budget** helps most Qwen3.5 models increasingly as models grow, whereas Qwen3-VL family and small Qwen3.5 models fail to reliably translate additional reasoning tokens into accuracy gains. A plausible explanation is that Qwen3.5's stronger multimodal stack better supports evidence selection and cross-view alignment, so added "thinking" tends to refine fusion decisions. In contrast, if Qwen3-VL and small Qwen3.5's visual grounding and fusion are weaker, additional tokens may preferentially elaborate on misread or misaligned cues, effectively amplifying early perceptual errors rather than correcting them, yielding flat or unstable gains even with larger reasoning budgets.

This phenomenon is particularly pronounced in Qwen3.5-0.8B: due to its limited visual feature extraction capacity, the reasoning process largely reiterates incorrect visual interpretations. Errors accumulate across reasoning steps, and the final prediction can be substantially worse than its non-reasoning counterpart.

**Scaling reasoning effort helps when the visual-evidence substrate is reliable.** Figure 3 (bottom-middle) shows that accuracy generally increases with reasoning effort, operationalized by average output tokens (from minimal/none to medium/high and `xhigh`). This pattern supports a "more inference-time compute → better diagnosis" trend, but the benefit remains limited. Together with the Qwen3.5 vs. Qwen3-VL contrast, these results suggest that inference-time scaling is *conditional*: additional reasoning tokens are beneficial primarily when the model's image reading and cross-view alignment are sufficiently accurate to serve as a trustworthy substrate. When that substrate is noisy, longer reasoning may not help and can even entrench spurious evidence chains.

## 4.4    DATA CONTAMINATION ANALYSIS

We assess potential test leakage using MELD (Memorization Effects Levenshtein Detector). Results show no evidence of severe contamination. Detailed analysis is provided in Appendix R.

## 4.5    DISCUSSION

Inference-time scaling is a useful but secondary lever in MedThinkVQA. Our results, together with recent medical evaluation studies, suggest that the main limitation is not a shortage of verbal deliberation alone, but a mismatch between the reasoning trace and the evidence actually grounded in the images. High final-answer accuracy can coexist with hidden image-understanding failures, and current medical models also remain weak at recognizing when evidence is missing or unreliable, so a longer chain-of-thought by itself is unlikely to be a sufficient fix (Yang et al., 2025; Griot et al., 2025). From this perspective, progress will likely depend on better control of where evidence enters the reasoning process and how each intermediate step is checked. First, supervision should move from answer-level to evidence-level. Recent work on rationale-guided MedVQA, stepwise medical evaluation, and process reward modeling shows that clinically meaningful errors are often exposed only at the intermediate-step level, and that retrieval-grounded verifiers can improve reasoning by checking steps against guidelines and literature (Gai et al., 2025; Wang et al., 2025a; Yun et al., 2025). In our setting, the per-image findings, case-level summaries, and option-wise eliminations provide precisely the structured targets needed to supervise view selection, cross-view fusion, and elimination reasoning rather than only the last answer token. Second, architectures should model image structure explicitly rather than flattening multiple views into a single undifferentiated context. Recent medical VLM work finds that stronger visual grounding improves downstream VQA and report generation, while multi-view longitudinal fusion improves clinical accuracy by preserving spatial and temporal structure across studies (Luo et al., 2025; Liu et al., 2025c). This suggests that future models for MedThinkVQA may need view-aware memory, temporal indexing, and grounding modules that can localize which image, region, or time point supports each claim. Third, test-time improvement should be case-adaptive and tool-assisted rather than purely free-form. In radiology and medical QA, multi-step retrieval, retrieval-augmented reasoning, and uncertainty-triggered expert control improve factual grounding or diagnostic accuracy by routing low-reliability cases to external knowledge or targeted correction (Wind et al., 2025; Tran et al., 2025; 2026; Liang et al., 2025). For multi-image diagnosis, the most useful extra computation may therefore not be more tokens alone, but targeted

actions such as retrieving radiology references, invoking a finding-grounder, comparing prior studies, or abstaining when cross-view evidence remains inconsistent.

## 5 RELATED WORK

Early MedVQA corpora set task forms but had small scale or shallow reasoning (Ben Abacha et al., 2019; Lau et al., 2018; Liu et al., 2021; He et al., 2020; Zhang et al., 2023). Later unified benchmarks grew breadth across modalities and specialties (Hu et al., 2024; Chen et al., 2024c). General expert-level suites also add a Health/Medicine subset and try to reduce shortcuts (Yue et al., 2024a;b). But most questions are single-image or short-context, and many use automatic labels. Many datasets are built from image captions, so labels do not encode diagnostic reasoning or multi-image context. They also lack the detailed clinical information that real cases need. Coverage of medical image types is still limited compared to practice. Within chest radiography and longitudinal imaging, large-scale corpora such as GEMeX (Liu et al., 2025b), Medical-Diff-VQA (MIMIC-Diff-VQA) (Hu et al., 2023), ICG-CXR (Ma et al., 2025), and the multi-image MedFrameQA benchmark (Yu et al., 2025) expand data scale and introduce explainable, difference-aware, counterfactual, or explicitly multi-image settings. However, their QAs and rationales are largely produced by rule-based or GPT-style pipelines rather than per-item expert traces, most items remain single-view or image-pair based, and they do not provide the per-image findings, case-level imaging summaries, or teaching notes needed for stepwise diagnostic supervision. So evaluation stays answer-centric and lacks stepwise diagnostic supervision, as reflected in the upper rows of our comparison.

MedXpertQA raises difficulty and realism and has a multimodal track with images and histories (Zuo et al., 2025). It also provides scenario labels. But it does not release expert *per-image findings* or a *case-level imaging summary*, and it does not annotate option-wise eliminations. Items also use far fewer images per case (max $\leq 1.43$), so cross-view fusion is not stressed. We fill these gaps with expert step labels (per-image findings and a case summary), with option-wise eliminations, and with a reproducible beyond-accuracy suite (step metrics, error types, and education scoring).

Eurorad-based studies often prompt models with textual descriptions from case reports (Kim et al., 2025). This probes language use, but it does not test reading raw images. Text-only prompting cannot test multi-image fusion or image dependence. In parallel, agent-style evaluation on chest X-rays (MedRAX/ChestAgentBench) orchestrates segmentation, grounding, report-generation, and classification tools on Eurorad-derived multiple-choice cases, but the released benchmark exposes only questions and images without the underlying expert step traces, complementing rather than replacing multi-image diagnostic supervision (Fallahpour et al., 2025). So our setting requires direct multi-image reading and option-wise, evidence-grounded elimination.

Work on reasoning supervision trains or audits how models explain answers (Gai et al., 2025; Liu et al., 2024; Wang et al., 2025b; Fan et al., 2025). Prior efforts include chain-of-thought generation, visually grounded reasoning, and cycle consistency. Recent corpora such as GEMeX-ThinkVG (Liu et al., 2025a) and S-Chain (Le-Duc et al., 2025) further introduce step-by-step rationales with explicit visual grounding and localization metrics (e.g., answer–reason scores, A-score, mIoU), moving beyond answer-only evaluation while still focusing mainly on single-image cases without clinical scenarios or multi-view, case-level synthesis. These help transparency and stability. But most corpora do not release expert, item-specific *diagnostic* traces tied to options. Without option-aligned traces, contrastive fidelity checks and step-level rubrics are hard to standardize. We release expert per-image findings and a case-level summary, and we pair them with option-wise eliminations. This enables contrastive fidelity checks, step-level scoring, and education-oriented evaluation with human and LLM judges. Teaching discussions are also a standard product of medical education, yet benchmarks rarely evaluate this skill.

## 6 CONCLUSION

We introduce MedThinkVQA, a large-scale benchmark for multimodal diagnostic reasoning in radiology, built from authentic multi-image cases with expert-authored reasoning traces. Our results show that this setting remains difficult even for the strongest current models, with performance mainly limited by failures in grounded multi-image evidence extraction and cross-view integration. We hope MedThinkVQA will serve as a rigorous testbed for developing models that can not only answer correctly but also reason more reliably in clinically realistic settings.

## ACKNOWLEDGMENTS

Research reported in this study was supported by the National Center on Homelessness Among Veterans (NCHAV) and by the National Institutes of Health (NIH) under award number 1R01NR020868, and 1I01HX003711-01A1. This study was also in part supported by NIH under award numbers R01DA056470-A1 and 1R01AG080670-01, and by the U.S. Department of Veterans Affairs (VA) Health Systems Research. The content is solely the responsibility of the authors and does not necessarily represent NIH, VA, or the US government.

We sincerely thank Juncheng Huang, MBBS (Department of Diagnostic Imaging, National University Hospital, Singapore) and Chin Siang Ong, MBBS PhD MPH (Assistant Professor of Surgery, Division of Surgical Outcomes, Surgery Center for Health Services and Outcomes Research, Department of Surgery, Yale School of Medicine) for their invaluable clinical expertise and dedication to this project. They contributed substantially to the expert annotation process of MedThinkVQA. Their rigorous review helped ensure the clinical fidelity of per-image findings, integrated imaging summaries, differential diagnosis construction, and case discussions. In addition, their professional insights guided the refinement of the held-out test set, the auditing of image-level supportiveness, and the identification of clinically challenging and longitudinal cases. Their careful clinical judgment and thoughtful feedback significantly strengthened the reliability, educational value, and medical realism of this benchmark. We are deeply grateful for their time, expertise, and commitment to advancing trustworthy clinical AI research.

## REPRODUCIBILITY STATEMENT

We provide full details to ensure reproducibility. Dataset sources and splits are in Section 2; implementation details are in Section 3; and training-set construction details are in Appendix B. We also provide the prompts used for data construction and LLM-judge evaluation in Appendices H, I, J, and L. We include an anonymized code repository link in the abstract.

## ETHICAL STATEMENT

**Data source, licensing, and legal compliance.** All cases are adapted from *Eurorad*, a peer-reviewed educational database maintained by the European Society of Radiology. Eurorad materials are licensed under *Creative Commons Attribution-NonCommercial-ShareAlike 4.0 International license*. MedThinkVQA follows the same license. Released data are for research and education only; commercial use is prohibited. Derivative datasets must preserve attribution, non-commercial use, and ShareAlike terms.

**Human subjects and privacy.** Eurorad cases are intended for education and are de-identified to the best of our knowledge. We did not collect new personal data and did not recruit patients or lay participants; IRB review was not required. We reviewed materials for residual identifiers and removed items when concerns arose.

**Evaluation reliability.** We combine automatic scripts, expert review, and LLM-judges. On step-level labels, human–human agreement is Cohen's $\kappa = 0.822833$, human1–LLM-judge agreement is $\kappa = 0.838357$, and human2–LLM-judge agreement is $\kappa = 0.701566$. These results support the stability of our automated judging, but LLM-judges do not replace expert oversight.

**Bias and fairness.** Educational repositories can encode geographic, demographic, and practice-style biases. Rare conditions and certain protocols are unevenly represented. Models trained or tuned on this benchmark may inherit such biases. We encourage stratified analyses and external validation before any deployment.

**Safety and misuse.** Models evaluated here are research artifacts. They must *not* be used for diagnosis, treatment, triage, or other high-stakes tasks without added clinical validation, regulatory clearance, and domain oversight. Generated discussions may sound authoritative yet still be incomplete or wrong. Any downstream use requires human supervision, documented fail-safes, and monitoring.

**Transparency, reproducibility, and environment.** We document data construction, metrics, and judging protocols. We release code, scoring scripts, and example data, subject to third-party licenses.

No hidden reward models, private test sets, or special samplers were used. We report hardware and runtime where relevant and encourage efficient evaluation to limit environmental impact.

**Conflicts of interest and ethics compliance.** All authors have read and will adhere to the ICLR Code of Ethics for submission, reviewing, and discussion. Any sponsorships or competing interests will be disclosed in the author checklist.

**Data leakage assessment and mitigation.** As discussed in Section 4.4, we conducted internal checks for leakage and found no obvious overlap between our test items and publicly released training artifacts that we were aware of. We remove text-only solvable items, strip explicit textual shortcuts, and stress cross-image fusion. Still, the risk of leakage cannot be ruled out. To reduce risk further, we will (i) release the full data collection and processing code for public audit, and (ii) maintain a rolling test set covering the most recent 6–12 months of newly curated cases, with periodic updates and refreshed scores for reported models. We will also publish de-duplication scripts (exact/near-duplicate filters on images and texts) and document all split procedures.

**Others.** MedThinkVQA is a research benchmark, not a clinical tool. Expert-authored traces are pedagogical; they may overlook interpersonal nuances, local workflows, and institutional contexts. The multiple-choice setting enables standardized scoring; it also simplifies real diagnostic work and stops before treatment planning and longitudinal follow-up. Coverage is broad but not complete across body regions, patient groups, vendors, devices, and acquisition protocols. Although cases span many conditions, some specialties (e.g., pediatrics, psychiatry) and rare diseases remain underrepresented. All cases originate from a single educational repository, so distribution shifts across hospitals, populations, and imaging pipelines are likely. The dataset is currently English-only; multilingual generalization has not been tested. Annotations, while expert-written, can still contain noise or stylistic variation. Our LLM-as-Judge components improve scalability, but they can be prompt-sensitive and may reflect judge-model biases; we therefore report human agreement and keep experts informed. Finally, we evaluate stepwise reasoning for differential diagnosis; reference-free evaluation of clinical reasoning without ground-truth steps is left for future work.

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

## A  LLM USAGE

In accordance with the ICLR 2026 policies on LLM usage, we disclose how LLMs were used in this work. LLMs were employed to assist with grammar polishing, wording improvements, and drafting text during paper preparation. All technical content, proofs, experiments, and analyses were conceived, implemented, and validated by the authors. Authors remain fully responsible for the correctness of the claims and results.

No LLMs were used to generate research ideas, write code for experiments, or produce results. No confidential information was shared with LLMs, and no prompt injections or other inappropriate uses were involved.

This disclosure aligns with the ICLR Code of Ethics: contributions of tools are acknowledged, while accountability and verification rest entirely with the human authors.

## B  TRAINING SET

For training rows whose A–E option slots are incomplete, we use a GPT-5.2 model with an augmentation prompt (full prompt in Appendix H) to preserve existing non-empty options, fill the missing distractors to form a five-choice question, and revise the case-level discussion. The model receives the clinical history, imaging findings, patient images, per-image captions, current options, the original correct answer, and the existing discussion. It outputs a completed five-option set, an updated correct-answer letter aligned with the original diagnosis, and a revised case-level discussion that strengthens the elimination of incorrect options and the justification for the correct answer. The released training split contains 7,347 cases, each formatted as a five-choice single-best-answer question.

## C  EXPERT PERFORMANCE AND DATA QUALITY ANNOTATIONS

**Annotators and protocol.**  All annotations were provided by two board-certified clinicians in active clinical practice.[1] We randomly sampled 96 test cases and ran a two-round expert study aligned with our MCQ and stepwise evaluation. In *Round 1*, experts saw only the clinical history and all study images and selected one diagnosis out of five options, matching our VLM setup. In *Round 2*, they additionally received the full reference materials, including image captions, per-image findings, the integrated imaging summary, the teaching discussion, and the ground-truth answer, and audited each case for internal consistency, difficulty, and image redundancy, distinguishing supportive from redundant views. The same 96-case subset is used to evaluate VLM baselines for a fair human-model comparison.

**Round 1: Diagnostic Performance.**  Experts answered 74/96 cases correctly (**77.10%** accuracy).

**Round 2: Data Quality and Image Redundancy.**  In the audit phase, experts marked 2/96 cases (2.1%) as *possibly inconsistent* and 18/96 (18.8%) as *very difficult*, indicating that the benchmark largely reflects coherent teaching cases while retaining a non-trivial proportion of challenging items. For image redundancy, experts judged whether each view provided supportive evidence toward the final diagnosis. In 65/96 cases (463 images), all views were deemed supportive (100%). The remaining 31/96 cases contained 315 images, of which 222 (70.48%) were judged supportive. Overall, 685/778 images were rated as supportive (**88.05%**), with the rest considered redundant for determining the final diagnosis.

The expert study shows that experienced clinicians still clearly outperform state-of-the-art VLMs on *MedThinkVQA*, that the curated items are overwhelmingly consistent with only a small fraction flagged as potentially problematic, and that most cases require aggregating evidence from many supportive views. As shown in Fig. 3 and 4, when image_ratio = 0 the task reduces to choosing one diagnosis out of five options with nearly a random success probability of 20%, and accuracy then rises steadily as a larger proportion of case images is revealed across all models; together with the

---

[1]One annotator is a diagnostic radiologist at a tertiary academic hospital in Asia with 7 years of post-training experience, and the other is an academic surgeon at a U.S. medical school with 5 years of post-training experience.

| Case group | #Cases | #Images | #Supportive imgs | Supportive ratio (%) |
|---|---|---|---|---|
| All images supportive | 65 | 463 | 463 | 100.00 |
| Mixed supportive / redundant | 31 | 315 | 222 | 70.48 |
| **Overall (96 cases)** | **96** | **778** | **685** | **88.05** |

**Table 4: Round 2 expert audit: image-level redundancy vs. support.** Most cases use all images as supportive evidence; even in mixed cases, the majority of views remain supportive rather than redundant.

expert audit in Table 4, where 88% of images are rated supportive, this monotonic gain suggests that additional views are rarely pure redundancy and usually contribute useful diagnostic information, even though overall performance still remains well below human experts. At the same time, the realistic minority of redundant / non-supportive images means models must both integrate multiple supportive views and learn to down-weight redundant ones, mirroring how radiologists select and prioritize key views before forming a diagnosis.

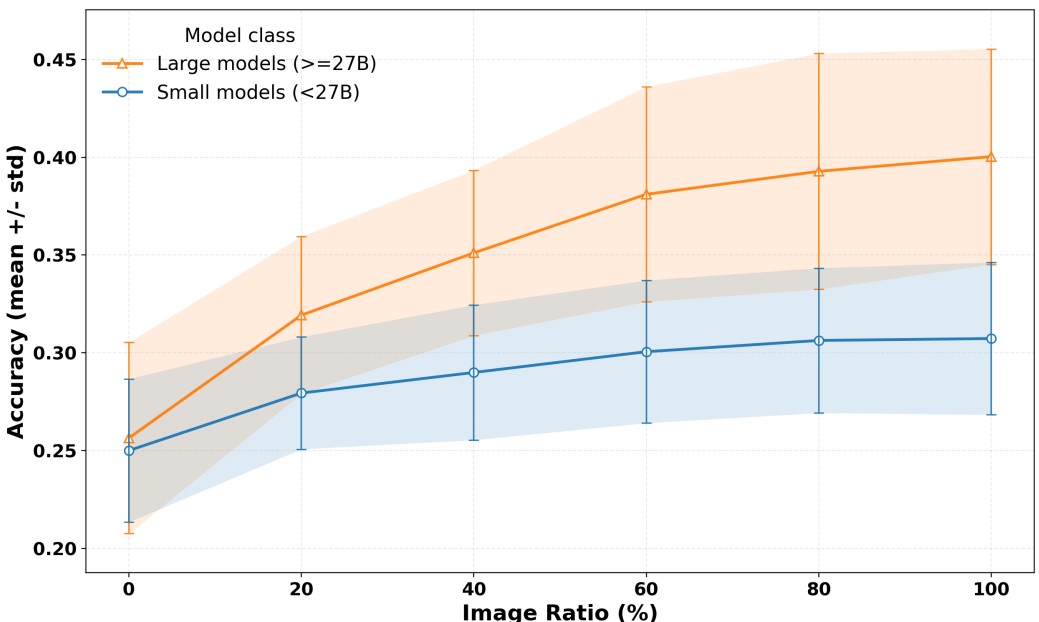

**Figure 4:** Performance of open-source multimodal models on MedThinkVQA under varying visible image ratios (0%–100%). Models are grouped by parameter scale into small ($< 27B$, n=14) and large ($> 27B$, n=11) categories. See Appendix D for full list of models

# D    ACCURACY BY IMAGE RATIO

We report **accuracy** (`acc`) and the number of correct predictions (`acc_correct`) on the full test set ($N = 720$). The **image ratio** $r$ denotes the fraction of available images used as visual evidence: when $r = 0\%$, the input contains only clinical history (text-only); when $r = 100\%$, all available images are provided (full visual evidence).

**Reasoning / Thinking setting.** To make "reasoning effort" (OpenAI GPT models) and "thinking mode" (Qwen models) directly comparable and easier to read, we separate the *base model name* from the *Reasoning/Thinking* setting in the tables below. For models without an explicit reasoning/thinking control, we mark the setting as *default*.

## D.1    FULL TEST RESULTS ($N = 720$, FULL VISUAL EVIDENCE; $r = 100\%$)

**Table 5:** Full-test results on the full test set ($N = 720$) with full visual evidence ($r = 100\%$).

| Model | Reasoning/Thinking | acc | #correct |
|---|---|---|---|
| **OpenAI (GPT-5 family)** | | | |
| GPT-5.2 | none | 0.4958 | 357 |
| GPT-5.2 | low | 0.5292 | 381 |
| GPT-5.2 | medium | 0.5444 | 392 |
| GPT-5.2 | high | 0.5333 | 384 |
| GPT-5.2 | x-high | 0.5486 | 395 |
| GPT-5 mini | minimal | 0.3306 | 238 |
| GPT-5 mini | low | 0.3903 | 281 |
| GPT-5 mini | medium | 0.3958 | 285 |
| GPT-5 mini | high | 0.3972 | 286 |
| GPT-5 nano | minimal | 0.2444 | 176 |
| GPT-5 nano | low | 0.2806 | 202 |
| GPT-5 nano | medium | 0.3125 | 225 |
| GPT-5 nano | high | 0.3083 | 222 |
| **Google (Gemini)** | | | |
| Gemini 2.5 Flash | default | 0.4270 | 307 |
| Gemini 2.5 Pro | default | 0.5097 | 367 |
| Gemini 3 Pro | default | 0.5569 | 401 |
| **Anthropic (Claude)** | | | |
| Claude 4.6 Opus | default | 0.5722 | 412 |
| **Microsoft (Phi)** | | | |
| Phi-4-multimodal-instruct | default | 0.2222 | 160 |
| **MedGemma** | | | |
| MedGemma 1.5 4B | default | 0.2958 | 213 |
| MedGemma 27B | default | 0.3181 | 229 |
| **Qwen-VL** | | | |
| Qwen3-VL-2B | non-thinking | 0.2972 | 214 |
| Qwen3-VL-2B | thinking | 0.2792 | 201 |
| Qwen3-VL-4B | non-thinking | 0.2697 | 194 |
| Qwen3-VL-4B | thinking | 0.2895 | 208 |
| Qwen3-VL-8B | non-thinking | 0.3125 | 225 |
| Qwen3-VL-8B | thinking | 0.2910 | 210 |
| Qwen3-VL-30B-A3B | non-thinking | 0.3109 | 224 |
| Qwen3-VL-30B-A3B | thinking | 0.2952 | 213 |
| Qwen3-VL-32B | non-thinking | 0.3425 | 247 |
| Qwen3-VL-32B | thinking | 0.3292 | 237 |
| Qwen2.5-VL-72B | non-thinking | 0.4000 | 288 |

| Model | Reasoning/Thinking | acc | #correct |
|---|---|---|---|
| Qwen3-VL-235B-A22B (FP8) | non-thinking | 0.3528 | 254 |
| Qwen3-VL-235B-A22B (FP8) | thinking | 0.3528 | 254 |
| **Qwen3.5** | | | |
| Qwen3.5-0.8B | non-thinking | 0.3139 | 226 |
| Qwen3.5-0.8B | thinking | 0.2097 | 151 |
| Qwen3.5-2B | non-thinking | 0.3250 | 234 |
| Qwen3.5-2B | thinking | 0.3000 | 216 |
| Qwen3.5-4B | non-thinking | 0.3444 | 248 |
| Qwen3.5-4B | thinking | 0.3764 | 271 |
| Qwen3.5-9B | non-thinking | 0.3694 | 266 |
| Qwen3.5-9B | thinking | 0.4417 | 318 |
| Qwen3.5-27B | non-thinking | 0.4500 | 324 |
| Qwen3.5-27B | thinking | 0.5056 | 364 |
| Qwen3.5-35B-A3B | non-thinking | 0.4111 | 296 |
| Qwen3.5-35B-A3B | thinking | 0.4500 | 324 |
| Qwen3.5-122B-A10B | non-thinking | 0.4528 | 326 |
| Qwen3.5-122B-A10B | thinking | 0.5125 | 369 |
| Qwen3.5-397B-A17B (FP8) | non-thinking | 0.4569 | 329 |
| Qwen3.5-397B-A17B (FP8) | thinking | 0.5222 | 376 |
| **Lingshu (vLLM)** | | | |
| Lingshu-7B | default | 0.3694 | 266 |
| Lingshu-I-8B | default | 0.2917 | 210 |
| Lingshu-32B | default | 0.4319 | 311 |
| **InternVL3.5 (vLLM)** | | | |
| InternVL3.5-1B | default | 0.3778 | 272 |
| InternVL3.5-2B | default | 0.3556 | 256 |
| InternVL3.5-4B | default | 0.3375 | 243 |
| InternVL3.5-8B | default | 0.3347 | 241 |
| InternVL3.5-14B | default | 0.3569 | 257 |
| InternVL3.5-30B-A3B | default | 0.3542 | 255 |
| InternVL3.5-38B | default | 0.4069 | 293 |

## D.2 IMAGE-RATIO ABLATION ($r \in \{0, 20, 40, 60, 80\}\%$)

The $r = 100\%$ results coincide with the full-test results in Table 5 and are therefore not repeated here. Each cell shows `newacc` (#correct); missing entries indicate that a model was not evaluated at that ratio in `selected_model_acc.csv`. The complete ablation matrix is reported in Table 6.

**Table 6:** Image-ratio ablation results ($r \in \{0, 20, 40, 60, 80\}\%$).

| Model | Reasoning/Thinking | 0% | 20% | 40% | 60% | 80% |
|---|---|---|---|---|---|---|
| **OpenAI (GPT-5 family; API)** | | | | | | |
| GPT-5.1 | low | 0.3389 (244) | – | – | – | – |
| GPT-5.2 | low | 0.3653 (263) | – | – | – | – |
| GPT-5 | medium | 0.3264 (235) | 0.3736 (269) | 0.4319 (311) | 0.4583 (330) | 0.4736 (341) |
| GPT-5 | minimal | 0.2806 (202) | 0.3278 (236) | 0.3847 (277) | 0.3806 (274) | 0.3931 (283) |
| GPT-5 mini | medium | 0.2847 (205) | 0.3528 (254) | 0.3542 (255) | 0.3667 (264) | 0.3861 (278) |
| GPT-5 mini | minimal | 0.2625 (189) | 0.3097 (223) | 0.3153 (227) | 0.3278 (236) | 0.3250 (234) |
| GPT-5 nano | low | 0.2097 (151) | 0.2653 (191) | 0.2708 (195) | 0.2639 (190) | 0.2806 (202) |
| GPT-5 nano | medium | 0.2250 (162) | 0.2819 (203) | 0.3000 (216) | 0.3111 (224) | 0.3111 (224) |
| GPT-5 nano | minimal | 0.2125 (153) | 0.2236 (161) | 0.2208 (159) | 0.2417 (174) | 0.2375 (171) |
| **MedGemma (vLLM)** | | | | | | |
| MedGemma 1.5 4B (IT) | default | 0.2319 (167) | 0.2611 (188) | 0.2569 (185) | 0.2597 (187) | 0.2597 (187) |
| **Lingshu (vLLM)** | | | | | | |

*Continued on next page*

| | ROUGE-L (↑) | | RadCliQ-v1 (↑) | |
|---|---|---|---|---|
| Model | Caption | Findings | Caption | Findings |
| gpt-5-nano | 0.1435 | 0.1585 | 0.8080 | 0.6781 |
| gpt-5-mini | 0.1510 | 0.1636 | 0.8317 | 0.6931 |
| GPT-5 | 0.1534 | 0.1627 | 0.8341 | 0.6818 |
| medgemma-27b-it | 0.1336 | 0.1621 | 0.7810 | 0.7192 |

**Table 7:** Scores of VLMs for Image→Caption and Image→Findings across two metrics (ROUGE-L and RadCliQ).

| Model | Reasoning/Thinking | 0% | 20% | 40% | 60% | 80% |
|---|---|---|---|---|---|---|
| Lingshu-7B | default | 0.2792 (201) | 0.3111 (224) | 0.3458 (249) | 0.3694 (266) | 0.3694 (266) |
| Lingshu-I-8B | default | 0.2903 (209) | 0.2903 (209) | 0.2903 (209) | 0.2903 (209) | 0.2875 (207) |
| Lingshu-32B | default | 0.3208 (231) | 0.3625 (261) | 0.4069 (293) | 0.4250 (306) | 0.4194 (302) |
| **InternVL3.5 (vLLM)** | | | | | | |
| InternVL3.5-1B | default | 0.3292 (237) | 0.3639 (262) | 0.3667 (264) | 0.3639 (262) | 0.3708 (267) |
| InternVL3.5-2B | default | 0.2944 (212) | 0.3181 (229) | 0.3250 (234) | 0.3389 (244) | 0.3389 (244) |
| InternVL3.5-4B | default | 0.2375 (171) | 0.2819 (203) | 0.2931 (211) | 0.3153 (227) | 0.3389 (244) |
| InternVL3.5-8B | default | 0.2208 (159) | 0.2681 (193) | 0.2847 (205) | 0.3167 (228) | 0.3222 (232) |
| InternVL3.5-14B | default | 0.2500 (180) | 0.2944 (212) | 0.3222 (232) | 0.3333 (240) | 0.3361 (242) |
| InternVL3.5-30B-A3B | default | 0.2458 (177) | 0.2958 (213) | 0.3208 (231) | 0.3458 (249) | 0.3542 (255) |
| InternVL3.5-38B | default | 0.2403 (173) | 0.3181 (229) | 0.3569 (257) | 0.3708 (267) | 0.4139 (298) |
| **Qwen-VL (vLLM)** | | | | | | |
| Qwen3-VL-2B | non-thinking | 0.2875 (207) | 0.2944 (212) | 0.3014 (217) | 0.2931 (211) | 0.2931 (211) |
| Qwen3-VL-2B | thinking | 0.2750 (198) | 0.2806 (202) | 0.2625 (189) | 0.2861 (206) | 0.3000 (216) |
| Qwen3-VL-4B | non-thinking | 0.2375 (171) | 0.2514 (181) | 0.2389 (172) | 0.2569 (185) | 0.2625 (189) |
| Qwen3-VL-4B | thinking | 0.2236 (161) | 0.2806 (202) | 0.3042 (219) | 0.2972 (214) | 0.3153 (227) |
| Qwen3-VL-8B | non-thinking | 0.2347 (169) | 0.2569 (185) | 0.2778 (200) | 0.2917 (210) | 0.3181 (229) |
| Qwen3-VL-8B | thinking | 0.2111 (152) | 0.2639 (190) | 0.3014 (217) | 0.3056 (220) | 0.2944 (212) |
| Qwen3-VL-32B | non-thinking | 0.2444 (176) | 0.2931 (211) | 0.2944 (212) | 0.3167 (228) | 0.3250 (234) |
| Qwen3-VL-32B | thinking | 0.2361 (170) | 0.2694 (194) | 0.3097 (223) | 0.3069 (221) | 0.3014 (217) |
| Qwen2.5-VL-72B | non-thinking | 0.2250 (162) | 0.2889 (208) | 0.3111 (224) | 0.3264 (235) | 0.3278 (236) |
| **Qwen3.5 (vLLM)** | | | | | | |
| Qwen3.5-27B | non-thinking | 0.3278 (236) | 0.3514 (253) | 0.3806 (274) | 0.4097 (295) | 0.4319 (311) |
| Qwen3.5-27B | thinking | 0.3875 (279) | 0.3806 (274) | 0.4347 (313) | 0.4569 (329) | 0.4861 (350) |
| Qwen3.5-35B-A3B | non-thinking | 0.2736 (197) | 0.3083 (222) | 0.3639 (262) | 0.3819 (275) | 0.3889 (280) |
| Qwen3.5-122B-A10B | non-thinking | 0.3153 (227) | 0.3708 (267) | 0.3917 (282) | 0.4306 (310) | 0.4500 (324) |
| Qwen3.5-122B-A10B | thinking | – | – | – | 0.4681 (337) | 0.4806 (346) |

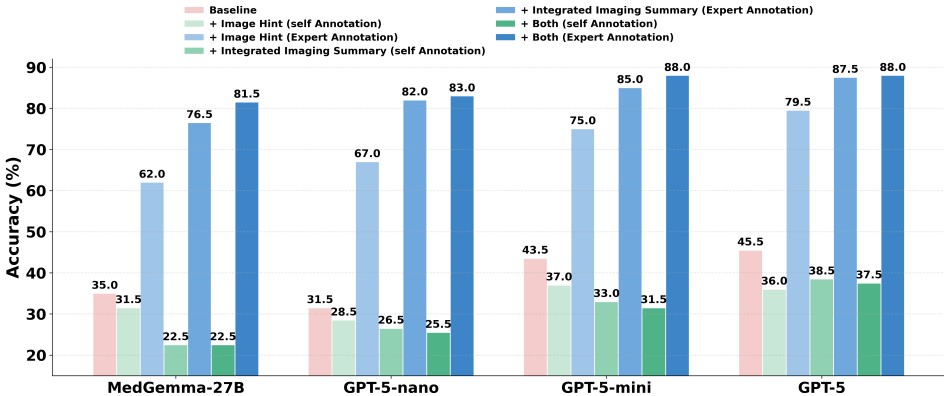

**Figure 5:** Accuracy on 200 randomly sampled cases from the *MedThinkVQA* when augmenting images with text. We compare `Image Hint` (caption-like) and `Integrated Imaging Summary` (diagnosis-oriented findings), each provided either by an *expert* or generated by the *model itself* (self). `Both` combines the two.

# E  MEDICAL EDUCATION CASE DISCUSSION

## E.1  CASE DISCUSSION AUTOMATIC EVALUATION

We implemented a comprehensive automatic evaluation framework to assess the quality of generated case discussions using GPT-5 as the evaluator. Each generated discussion contained multiple subsections, including background, clinical perspective, imaging perspective, outcome, and take-home messages. Our evaluation employed a two-stage approach: first, we conducted sentence-level factual-correctness assessment by splitting each subsection into individual sentences and tasking a prompted LLM (GPT-5) to judge the correctness of each sentence based on the provided case context, imaging findings, differential diagnosis list, image captions, and medical images. The evaluator was instructed to mark sentences as true if explicitly supported or reasonably inferable from the context, and false only if clearly contradictory or incorrect. Second, we performed quality assessment using an expert-curated rubric that scored discussions on five key criteria: disease overview, clinical pathophysiology, imaging analysis, reasoning and differentials, and transferable learning, with each criterion rated on a 0–2 scale. The LLM evaluator provided both numerical scores and brief justifications for each rubric criterion, focusing on medical accuracy, completeness, educational value, and integration of clinical and imaging perspectives. For the automatic evaluation, we randomly sampled 20 case discussions from our dataset for GPT-5 to evaluate using this framework.

## E.2  CASE DISCUSSION HUMAN EVALUATION

To validate our automatic evaluation framework, we conducted human evaluation using two medical experts who independently assessed radiology case discussions. Each evaluator was presented with one case discussion randomly selected from outputs generated by three different models, ensuring blinded assessment without knowledge of the generating model. Following the same two-stage methodology as the automatic evaluation, the human evaluators first performed sentence-level factual-correctness evaluation and then applied the expert-curated rubric to provide quality scores. This human evaluation served as the gold standard for assessing the reliability and validity of our automated evaluation approach.

## E.3  CASE DISCUSSION EVALUATION RESULTS

The generated case discussions demonstrated high factual accuracy across all tested models, with overall correctness rates ranging from 92.81% to 99.22%, as shown in Tab. 8. The GPT-5 series consistently achieved the highest factual correctness, while the Clinical Perspective subsection scored highest across all models (97.89–100%). The Outcome subsection showed some performance differences, with MedGemma-27B achieving 85.71% compared to the other models, which scored above 95%. The rubric-based evaluation revealed that GPT-5 achieved the highest overall score of 9.9/10. MedGemma-27B scored 7.05/10, showing particular weakness in clinical pathophysiology (1.15/2) and reasoning differentials (1.1/2), while all models demonstrated consistent strength in disease overview and imaging findings (Tab. 9).

| Model | Background (%) | Clinical (%) | Imaging (%) | Outcome (%) | Take-Home (%) | Overall (%) |
|---|---|---|---|---|---|---|
| gpt-5 | 100.0 | 100.0 | 97.81 | 98.70 | 100.0 | 99.08 |
| gpt-5-mini | 98.59 | 98.65 | 99.10 | 100.0 | 100.0 | 99.22 |
| gpt-5-nano | 97.87 | 98.99 | 97.39 | 95.89 | 98.46 | 97.76 |
| medgemma-27b-it | 89.0 | 97.89 | 94.93 | 85.71 | 93.65 | 92.81 |

**Table 8:** Sentence-level factual correctness evaluation across discussion subsections

| Model | Total | Disease Overview | Clinical Pathophys. | Imaging | Reasoning Different. | Transfer Learning |
|---|---|---|---|---|---|---|
| gpt-5 | 9.9 | 2.0 | 1.9 | 2.0 | 2.0 | 2.0 |
| gpt-5-mini | 9.4 | 1.95 | 1.6 | 2.0 | 1.85 | 2.0 |
| gpt-5-nano | 8.4 | 1.7 | 1.25 | 2.0 | 1.45 | 2.0 |
| medgemma-27b-it | 7.05 | 1.4 | 1.15 | 1.85 | 1.1 | 1.55 |

**Table 9:** Rubric evaluation scores across different models

## F DATA CONTAMINATION ANALYSIS (MELD)

We assess potential test leakage using a strict sliding-window variant of MELD (Memorization Effects Levenshtein Detector), which measures character-level overlap between each model's generated answer and its input question on the MEDTHINKVQA test set.

Across seven representative LLMs/VLMs, MELD similarities cluster around 20–24% with narrow interquartile ranges (IQRs), and no item reaches the commonly used high-risk threshold of 50%. The distributions are also consistent across text-only and vision-language models, suggesting no family-specific effect.

As shown in Figure 18, all models exhibit low and tightly concentrated similarity scores, well below the high-risk threshold.

## G  EXAMPLE MAPPING FROM EURORAD FIELDS TO MEDTHINKVQA ANNOTATIONS

To make the supervision signals in MedThinkVQA concrete, this section uses a single Eurorad case from the test split, *"Ureteropelvic junction laceration following blunt trauma"*, whose processed JSON is stored at `cases/general/case_219/case.json`. We list the main JSON fields and show how they correspond to the supervision concepts used in the main text.

**Clinical scenario.**

- JSON field: `CLINICAL_HISTORY` (string).

- Content: brief free-text description of the presenting complaint and relevant history (for case 219, an elderly patient with cardiovascular comorbidities presenting with right-sided thoraco-abdominal trauma and microscopic haematuria).

- Main-text concept: this field is used verbatim as the *Clinical Scenario* shown to models before any images or answer options (Fig. 1, left).

**Per-image hints (*Image Hint* / per-image findings).**

- JSON field: `img` (list). Each element is a dictionary with keys `img_id`, `img_path`, `img_alt` (short legend), and `img_alt2` (full descriptive caption).

- Content structure (case 219):

  - Images 1–3: prior multidetector CT study 8 months earlier, showing bilateral peripelvic renal cysts with otherwise normal renal morphology.
  - Images 4–5: current CT for abdominal trauma, with right perirenal and fascial fluid and dependent hyperattenuation in the renal pelvis compatible with acute blood.
  - Image 6: arterial-phase CT and MIP reconstructions, without contrast extravasation, again emphasising hyperattenuation in the renal pelvis and a peripelvic cyst.
  - Images 7–8: delayed excretory-phase images, showing medial perirenal extraluminal opacified urine and normal parenchymal/collecting-system opacification.
  - Images 9–10: delayed images showing extraluminal opacified urine arising from a focal breach at the ureteropelvic junction and an opacified proximal ureter.

- Main-text concept: `img_alt2` provides the expert *per-image hint* used in Step 1 (*Image Hint* / per-image findings). In the TwI setting, models are asked to produce concise radiological finding sentences that are consistent with these captions.

**Case-level Integrated Imaging Summary.**

- JSON field: `IMAGING_FINDINGS` (string).

- Content: a case-level narrative integrating all imaging examinations (prior CT, current CT, delayed acquisitions), key abnormalities (peripelvic cysts, perirenal fluid, extraluminal opacified urine from a UPJ breach), and absence of other traumatic lesions, plus immediate management (e.g., ureteral stenting).

- Main-text concept: this field is the expert reference for the *Integrated Imaging Summary* (Step 2 in Fig. 1); models must fuse per-image findings into a single summary that matches this cross-view evidence.

**Differential diagnosis and MCQ construction.**

- JSON field: `DIF_DIAGNOSIS_LIST` (string with comma-separated diagnoses).

- Content (case 219, simplified): contains the target diagnosis "Ureteropelvic junction laceration following blunt trauma" and related entities such as "Ureteropelvic avulsion", "Renal parenchymal laceration with calyceal disruption", "Urinoma", "Perinephric haematoma", and "Subcapsular haematoma".

- Additional JSON fields used for the MCQ:

- **–** `options`: dictionary mapping option letters (`"A"–"E"`) to diagnosis strings.
- **–** `correct_answer`: the correct option letter (e.g., `"C"`).
- **–** `correct_answer_text`: the correct diagnosis string (e.g., "Ureteropelvic junction laceration following blunt trauma.").
- Main-text concept: these fields instantiate the five-option single-best-answer MCQ used in Step 3 (*Differential-Diagnosis Reasoning*); models compare their Integrated Imaging Summary to the `options` and must select `correct_answer_text`.

**Medical Education Case Discussion.**

- JSON field: `DISCUSSION` (string).
- Content: long-form teaching text covering epidemiology (e.g., rarity of UPJ injuries), mechanisms, imaging pitfalls, management strategies, and prognosis.
- Main-text concept: this field is the expert reference for the *Medical Education Case Discussion* task, where models generate a structured explanation (Background, Clinical Perspective, Imaging Perspective, Clinical Significance, Outcome, Take-Home Notes) that is graded against `DISCUSSION` for clinical correctness and educational value.

Overall, this case illustrates how raw Eurorad sections and figure captions are mapped onto the *Clinical Scenario*, *Image Hint*, *Integrated Imaging Summary*, *Differential Diagnosis*, and *Medical Education Case Discussion* supervision signals defined in the main text and implemented as JSON fields in MedThinkVQA.

## G.1 Multimodal Case Study: Primary Carcinoma of the Rectovaginal Septum

This MedThinkVQA case is a 61-year-old woman with pelvic pain and inguinal lymphadenopathy, ultimately diagnosed with *primary carcinoma of the rectovaginal septum*. In the JSON, all four modalities (Endoscopy, CT, MRI, Histology / pathology) share the same CLINICAL_HISTORY, IMAGING_FINDINGS, DISCUSSION, and each image is referenced by its img_id and stored under images/cases/modality/{img_id}.jpg with img_alt and img_alt2 captions.

**Endoscopy.** The endoscopic modality contains a single sigmoidoscopy frame (img_id = l9iMrGt3) that documents both focal bulging of the sigmoid wall and a 3 cm vegetative rectal lesion; these findings are encoded in the corresponding img_alt and img_alt2 fields and are shown in Fig. 6.

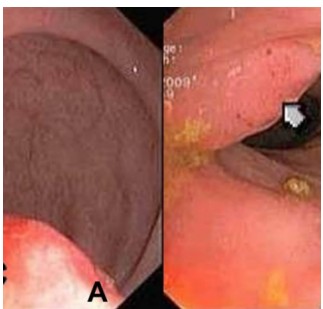

**Figure 6:** Endoscopy image for this case (img_id = l9iMrGt3).

**CT.** The CT modality consists of two axial contrast-enhanced CT images (3uIJtKe-, pfx97TC8) that show a heterogeneous mass in the pouch of Douglas, invasion of adjacent structures, and inguinal lymphadenopathy; the excretory-phase scan with rectal contrast further clarifies rectal wall involvement, as shown in Fig. 7.

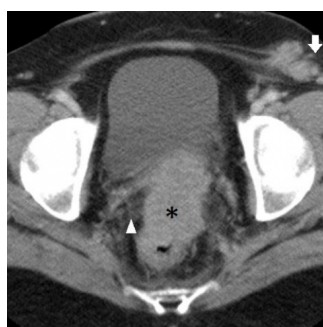

**3uIJtKe-.** Axial contrast-enhanced CT with pelvic mass and enlarged left inguinal nodes.

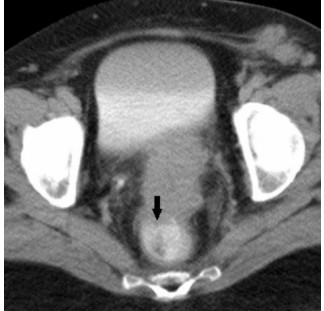

**pfx97TC8.** Excretory-phase CT with rectal contrast, better defining rectal wall involvement.

**Figure 7:** CT images (img_id = 3uIJtKe-, pfx97TC8) stored under images/cases/modality/.

**MRI.** The MRI modality includes five T2-weighted images (O5kEGVZq, IAV1h4UN, FjWYFzXB, F0KIjEeq, U14cWn_5) that jointly characterise mass location (rectovaginal septum), extension to cervix and myometrium, intimate contact with the rectal wall, nodal disease, and preservation of the right ovary and inner cervical stromal layer, all illustrated in Fig. 8.

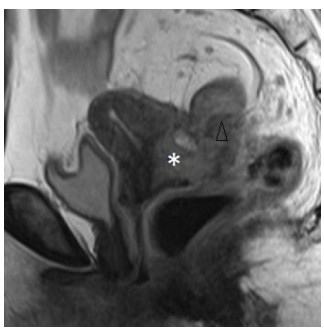

**O5kEGVZq.** Sagittal T2: mass in pouch of Douglas extending to cervix and myometrium.

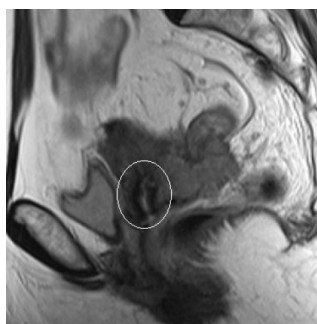

**IAV1h4UN.** Preserved low-signal inner cervical stroma.

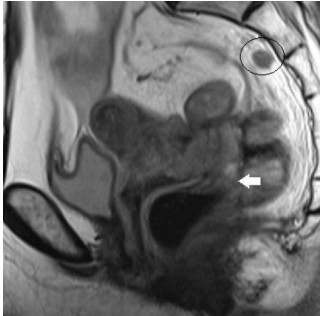

**FjWYFzXB.** Mass inseparable from the anterior rectal wall.

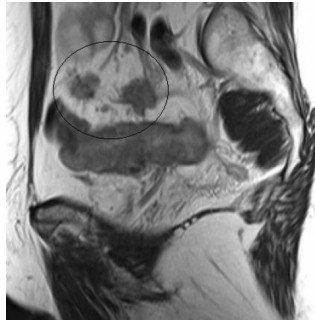

**F0KIjEeq.** Multiple enlarged pelvic and abdominal lymph nodes.

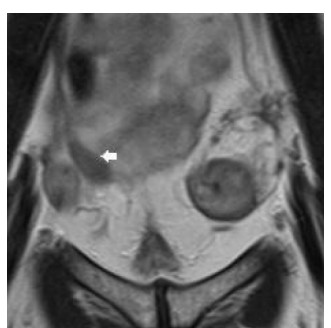

**U14cWn_5.** Coronal T2: normal right ovary separately identified from the mass.

**Figure 8:** MRI images (`img_id = O5kEGVZq, IAV1h4UN, FjWYFzXB, F0KIjEeq, U14cWn_5`) with uniform image size and aligned top edges.

**Histology / pathology.** The histology / pathology modality contains a single composite slide (`3xrCMRPY`) showing solid tumour growth with marked atypia and immunostaining for CAM5.2, CK7, and WT1, all encoded in the `img_alt2` description and visualized in Fig. 9.

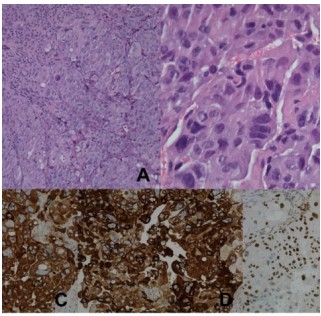

**Figure 9:** Histology / pathology image for this case (`img_id = 3xrCMRPY`).

**Multimodal reasoning signal.** In the dataset, this case is formatted as a five-option diagnostic MCQ with ground-truth label *primary carcinoma of the rectovaginal septum*. Endoscopy and CT highlight an extraluminal pelvic mass with rectal involvement; MRI localises the tumour to the rectovaginal septum and shows preserved cervix and ovaries with nodal spread; histology confirms a Müllerian-type carcinoma. A model must integrate all four modalities together with the shared textual fields in the JSON to distinguish this entity from rectal, cervical, and ovarian primaries.

### G.2 LONGITUDINAL CASE STUDY: CYSTIC PULMONARY TUBERCULOSIS

This MedThinkVQA case is a nine-year-old boy with severe cystic pulmonary tuberculosis, followed radiologically over almost a year. In the JSON, the shared CLINICAL_HISTORY, IMAGING_FINDINGS, and DISCUSSION fields are linked to chest radiographs and CT scans at multiple time points. Below, we group images by clinical time point to illustrate longitudinal disease evolution.

**Baseline imaging at admission.** Baseline chest radiographs (posteroanterior and lateral views) show a bilateral diffuse micronodular acinar infiltrate. A same-day chest CT (lung and mediastinal windows) reveals a diffuse micronodular pattern with random distribution throughout both lungs, suggestive of an inflammatory or infectious process; the full baseline image set is shown in Fig. 10.

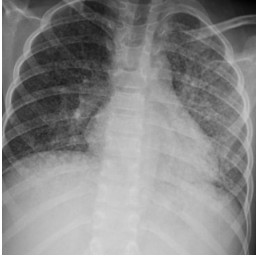

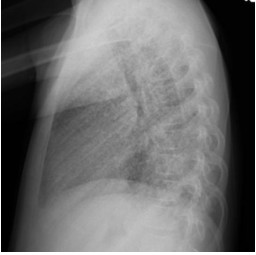

**PA view.** Baseline chest radiograph with diffuse micronodular infiltrates.

**Lateral view.** Baseline chest radiograph confirming bilateral involvement.

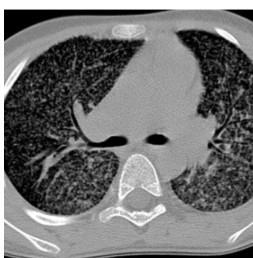

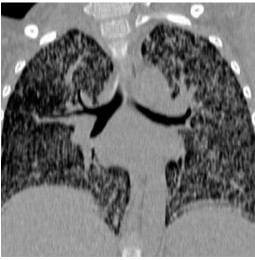

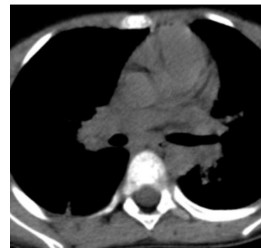

**CT, lung window.** Diffuse micronodular pattern in both lungs.

**CT, lung window.** Randomly distributed nodules throughout the parenchyma.

**CT, mediastinal window.** No large focal mass; diffuse micronodular disease.

**Figure 10:** Baseline chest radiographs and CT at admission, all displayed with a uniform relative size.

**Early course with pneumothoraces.** During the ICU stay, the patient develops spontaneous pneumothoraces requiring chest drainage. Serial radiographs show persistent diffuse micronodular infiltrates with evolving unilateral and bilateral pneumothoraces and multiple chest tubes in place, as illustrated in Fig. 11.

**Development of confluent cystic disease.** A subsequent contrast-enhanced CT demonstrates extensive confluent cystic lesions predominantly in the upper lobes and posterior regions, consistent with cystic pulmonary tuberculosis and explaining the recurrent pneumothoraces (Fig. 12).

**Persistent cysts and larger pneumothoraces.** A further CT shows similar cystic disease but larger bilateral pneumothoraces, pneumomediastinum, and multiple chest drains in place, underscoring the mechanical complications of cystic tuberculosis (Fig. 13).

**Pre-discharge CT.** Before discharge, CT still shows cystic lesions and residual pneumothorax, but with improved overall ventilation, as shown in Fig. 14. The patient tolerates these sequelae after chest tube removal and can leave the hospital.

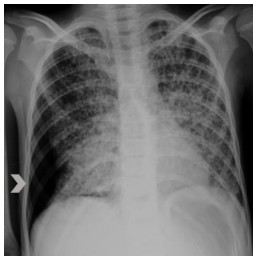

**Chest radiograph.** Right pneumothorax with chest tube in situ.

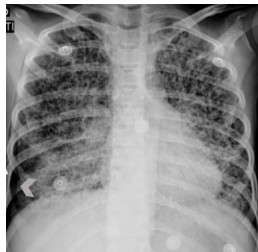

**Chest radiograph.** Persistent diffuse micronodular opacities.

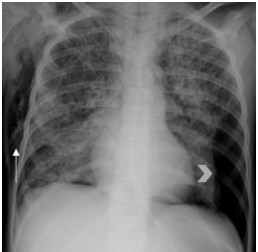

**Chest radiograph.** Bilateral lung involvement with scattered nodules.

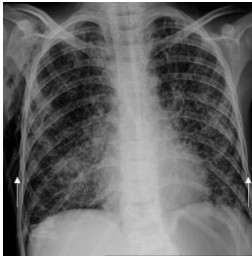

**Chest radiograph.** Multiple thoracic drains for recurrent pneumothoraces.

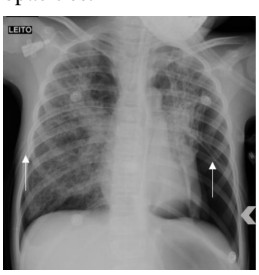

**Chest radiograph.** Persistent residual pneumothorax.

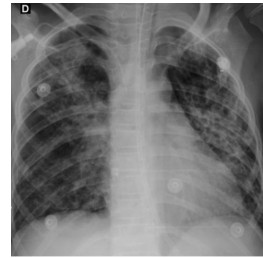

**Chest radiograph.** Diffuse cystic–nodular changes on a background of severe disease.

**Figure 11:** Serial chest radiographs during ICU stay, showing pneumothoraces and multiple chest drains.

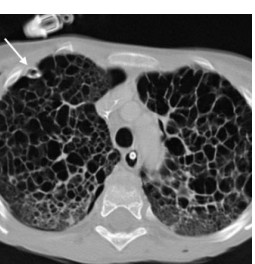

**CT, lung window.** Multiple cystic lesions in both upper lobes.

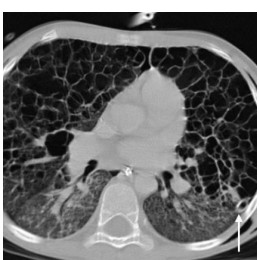

**CT, lung window.** Coalescent cysts forming large air-filled spaces.

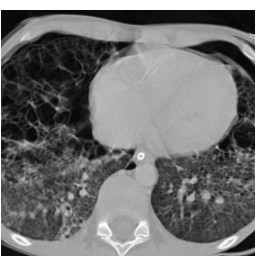

**CT, lung window.** Cystic lesions with surrounding ground-glass opacities.

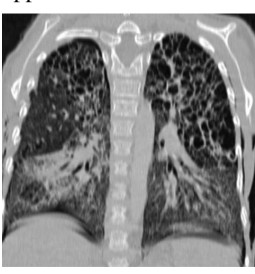

**CT, coronal view.** Upper-lobe predominance of cystic disease.

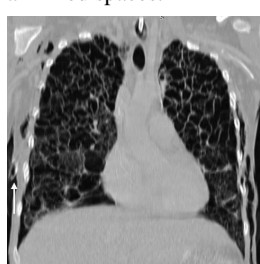

**CT, lung window.** Posterior lung involvement with confluent cysts.

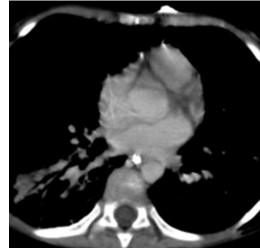

**CT, mediastinal window.** Multiple large cystic spaces without solid mass.

**Figure 12:** CT during peak disease severity, with confluent cystic lesions and diffuse parenchymal involvement.

**Late follow-up.** At eight months after discharge, follow-up CT shows near-complete resolution of the cystic and nodular lesions, with only subtle residual cysts and fibrotic sequelae, as illustrated in Fig. 15.

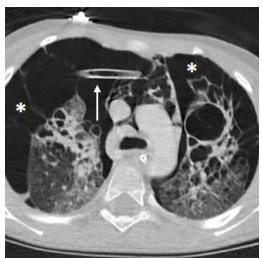

**CT, lung window.** Large right pneumothorax on a cystic background.

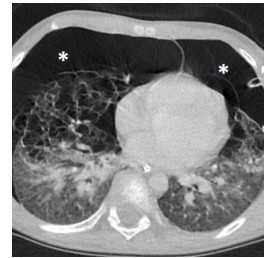

**CT, lung window.** Extensive bilateral cystic changes.

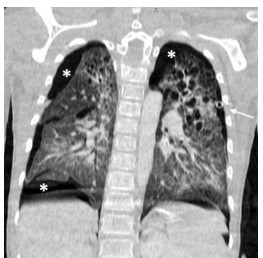

**CT, coronal view.** Bilateral pneumothoraces with upper-lobe cysts.

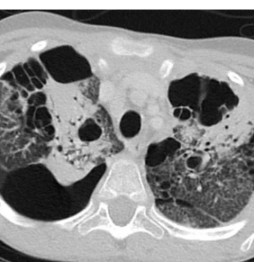

**CT, mediastinal window.** Pneumomediastinum and thoracic drains.

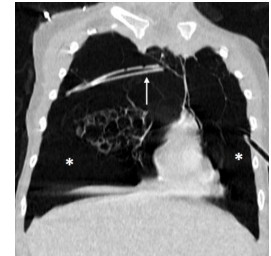

**CT, lung window.** Persistent cystic lesions despite drainage.

**Figure 13:** CT with larger pneumothoraces and pneumomediastinum, on a background of cystic pulmonary tuberculosis.

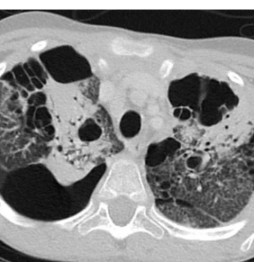

**CT, lung window.** Residual cystic changes with improved aeration.

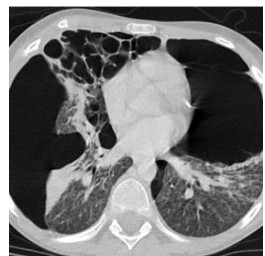

**CT, lung window.** Decreased extent of parenchymal disease.

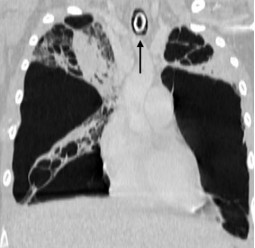

**CT, coronal view.** Tracheostomy in place with residual cysts.

**Figure 14:** Pre-discharge CT: persistent cystic lesions but improved clinical tolerance and removal of chest drains.

**Longitudinal reasoning signal.** In MedThinkVQA, this case is encoded as a five-option diagnostic MCQ with the correct answer *cystic pulmonary tuberculosis*. A model must integrate longitudinal information across all time points—progression from micronodular infiltrates to confluent cystic disease, recurrent pneumothoraces requiring multiple drains, and eventual radiologic recovery—together with the clinical text to distinguish this entity from other cystic lung diseases (e.g., *Pneumocystis jirovecii* pneumonia, Langerhans cell histiocytosis, Birt–Hogg–Dubé syndrome). The unified, uniformly sized image panels highlight how temporal evolution in a single patient can be represented as a structured longitudinal multimodal item in our dataset.

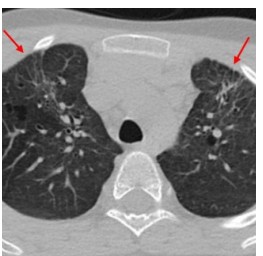 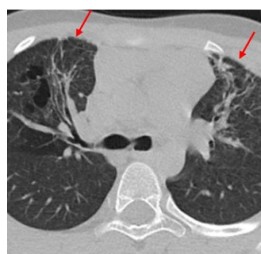 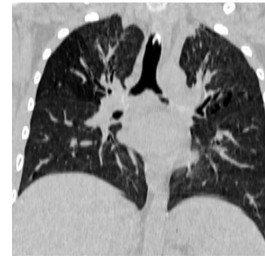

**CT, lung window.** Marked improvement with near-normal parenchyma.

**CT, lung window.** Few residual discrete cysts.

**CT, coronal view.** Almost complete radiologic recovery.

**Figure 15:** Late follow-up CT eight months after discharge, demonstrating almost complete recovery with limited sequelae.

### G.3 GPT-5 Correct Case Study: Hibernoma of the Chest Wall

A 28-year-old woman underwent a conventional chest examination for suspected pneumonia, which incidentally revealed a right paraspinal chest-wall mass. Radiography showed a paraspinal opacity with scoliosis and rib deformities. CT demonstrated a solid, non-mineralised paravertebral lesion with fatty components slightly denser than subcutaneous fat and prominent internal serpiginous vessels. MRI confirmed predominantly fatty signal intensity that was slightly less bright than subcutaneous fat, with incomplete fat suppression, streaky soft-tissue components and slow, inhomogeneous enhancement after contrast—features typical of a hypervascular brown-fat tumour (hibernoma); the full multimodal image set is shown in Fig. 16.

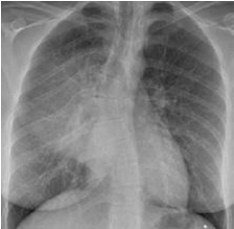

**Chest radiograph.** Right paraspinal mass with scoliosis and rib deformities.

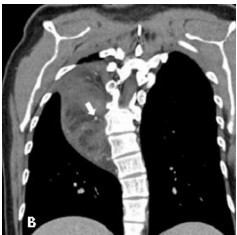

**CT, coronal view.** Fatty mass slightly denser than subcutaneous fat.

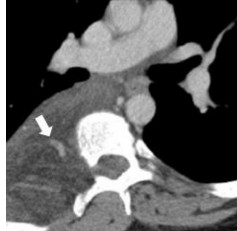

**CT, axial view.** Solid lesion with internal serpiginous enhancing vessels.

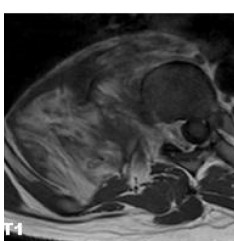

**MRI T1-weighted.** Predominantly fatty high signal, not as bright as subcutaneous fat.

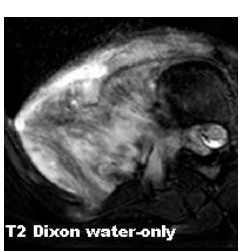

**MRI T2 with fat suppression.** Incomplete fat suppression with heterogeneous appearance.

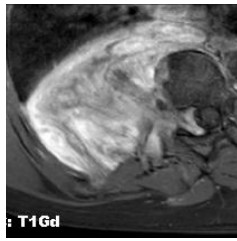

**MRI T1 post-contrast.** Slow, inhomogeneous enhancement of soft-tissue components.

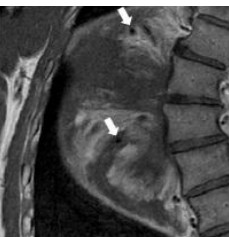

**MRI T1-weighted.** Fatty tissue with streaky muscle-like strands.

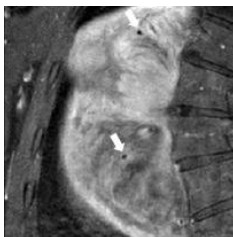

**MRI T1 post-contrast.** Prominent internal vessels with flow voids and enhancement.

**Figure 16:** Imaging from a paraspinal chest-wall mass later diagnosed as hibernoma: chest radiograph, contrast-enhanced CT, and multiparametric MRI (T1, T2 with fat suppression, and post-contrast). All images are stored in the dataset as individual files associated with this case.

**GPT-5 answer.** For the five-option question (Atypical lipomatous tumour; Clear cell sarcoma of the soft tissues; Hibernoma of the chest wall; Liposarcoma; Resolving haematoma), GPT-5 selected *Hibernoma of the chest wall* and produced the following explanation:

> Young woman with a paraspinal chest-wall mass showing predominantly fatty density with prominent internal serpiginous vessels on CT and marked, heterogeneous enhancement on MRI. The lesion is slightly less T1-hyperintense than subcutaneous fat and demonstrates avid enhancement with flow voids, features characteristic of a hypervascular brown-fat tumor (hibernoma). Atypical lipomatous tumor / well-differentiated liposarcoma and other liposarcomas typically occur in older patients, are less vascular and show thick septa or nodular non-fatty components rather than prominent intralesional vessels. Clear cell sarcoma lacks macroscopic fat and classically arises in distal extremities. A resolving hematoma would not show this degree of internal vascularity or persistent avid enhancement. Thus, hibernoma of the chest wall is most likely.

**Why this reasoning is correct.** GPT-5's explanation closely matches the expert interpretation of this case. It correctly recognises a fatty but *not quite normal* fat signal pattern (slightly denser than

subcutaneous fat on CT, slightly less T1-bright on MRI, incomplete fat suppression) and emphasises the key hallmark of hibernoma: a hypervascular intramuscular lesion with prominent serpiginous internal vessels and slow, inhomogeneous enhancement rather than a homogeneous pure-fat mass. It then uses these imaging cues, plus the patient's young age and paraspinal chest-wall location, to rule out the main differentials: atypical lipomatous tumour and other liposarcomas (typically less vascular, in older patients, with thick septa and nodular non-fatty components), clear cell sarcoma (no macroscopic fat, usually distal extremities), and resolving haematoma (lacking persistent vascular flow voids and avid enhancement). This stepwise, modality-aware reasoning is consistent with the teaching point for hibernoma and leads to the correct diagnosis for this MedThinkVQA item.

### G.4 GPT-5 ERROR CASE STUDY

#### G.4.1 GPT-5 ERROR CASE STUDY: COMBINED WILKIE, NUTCRACKER, AND MAY–THURNER SYNDROMES

A 26-year-old woman with a three-year history of weight loss and postprandial abdominal discomfort, prior anorexia nervosa, and known pelvic congestion syndrome underwent contrast-enhanced abdominal CT. Imaging demonstrated: (i) compression of the third portion of the duodenum between the superior mesenteric artery (SMA) and aorta with reduced aortomesenteric angle and distance (Wilkie / SMA syndrome), (ii) compression of the left renal vein between the aorta and SMA with the classic "beak sign," proximal left renal vein dilatation and engorged left ovarian and pelvic veins (Nutcracker syndrome), and (iii) compression of the left common iliac vein by the right common iliac artery against the lumbar spine (May–Thurner syndrome), together with gastric and proximal duodenal dilatation; the complete nine-slice set is shown in Fig. 17.

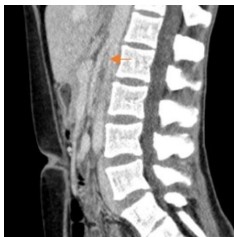

**Sagittal CT.** Third portion of the duodenum compressed between aorta and SMA.

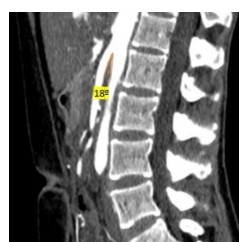

**Aortomesenteric angle.** Markedly reduced SMA–aorta angle.

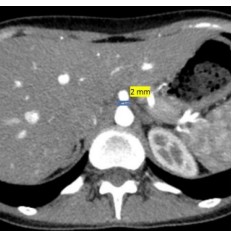

**Aortomesenteric distance.** Narrowed aortomesenteric distance with duodenal compression.

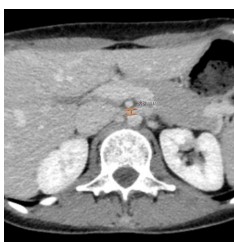

**LRV beak sign.** Abrupt narrowing of the left renal vein between aorta and SMA.

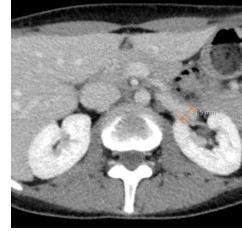

**LRV dilatation.** Proximal enlargement of the left renal vein.

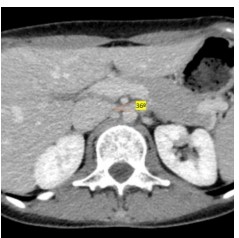

**Beak geometry.** Angulated, tapered appearance of the compressed left renal vein.

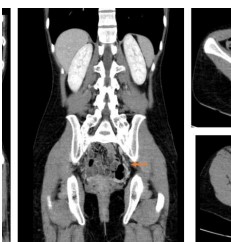

**Pelvic veins.** Tortuous left ovarian vein and dilated pelvic venous plexus.

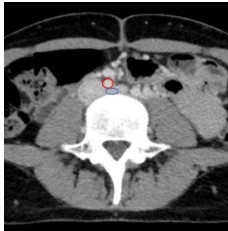

**May–Thurner.** Compression of the left common iliac vein by the right common iliac artery.

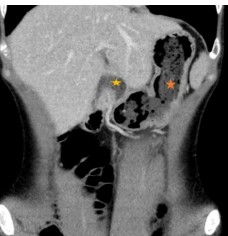

**Global effect.** Dilatation of stomach and proximal duodenum from outflow obstruction.

**Figure 17:** Nine CT slices from a single case demonstrating three concurrent vascular compression syndromes: Wilkie (SMA) syndrome, Nutcracker syndrome with pelvic venous congestion, and May–Thurner syndrome. All images are stored as separate files but belong to one MedThinkVQA item.

**GPT-5 answer (incorrect).** For the five-option question (iliac vein compression by an occult pelvic mass; hereditary megaduodenum; scleroderma with duodenal involvement; superior mesenteric artery syndrome; Nutcracker syndrome and May–Thurner syndrome), GPT-5 selected *superior mesenteric artery syndrome* and produced the following explanation:

> CT demonstrates a markedly reduced aortomesenteric angle ($\approx 18°$) and aortomesenteric distance ($\sim 2\,\mathrm{mm}$), which are diagnostic measurements for compression of the third portion of the duodenum by the SMA, that is, superior mesenteric artery (SMA) syndrome. The clinical context of a young woman with significant weight loss and anorexia nervosa, plus postprandial abdominal discomfort, is classic for SMA syndrome due to loss of the mesenteric fat pad. While other vascular compression findings may be present, the imaging and symptoms most directly support SMA syndrome as the primary diagnosis.

**Why GPT-5 is wrong: failure of multi-image reasoning.** Although this explanation is internally coherent for an *isolated* Wilkie / SMA syndrome, it systematically ignores the additional information carried by the other six images. Three CT slices do quantify the reduced aortomesenteric angle and distance and show duodenal compression, but the remaining views encode two further, independent haemodynamic abnormalities:

- Several slices through the renal hilum show an abrupt, "beak"-shaped narrowing of the left renal vein between the aorta and SMA with proximal dilatation and engorged ovarian and pelvic veins, which is classic for Nutcracker syndrome with pelvic congestion.
- A slice at the iliac bifurcation demonstrates focal compression of the left common iliac vein by the right common iliac artery against the spine, characteristic of May–Thurner syndrome.

The ground-truth label for this MedThinkVQA item reflects the *combination* of vascular compression syndromes documented in the full nine-image set, not just SMA syndrome. However, GPT-5 effectively behaves as if it were solving a single-image task: it focuses on the most salient slice showing SMA–duodenum compression, maps this to the textbook pattern of superior mesenteric artery syndrome, and then dismisses the rest with the remark that other compression findings "may be present."

This behaviour illustrates a central limitation of current multimodal LLMs on genuinely multi-image cases. Instead of integrating heterogeneous evidence across different anatomical levels and mechanisms (duodenal obstruction, renal venous outflow obstruction, iliac venous compression), the model implicitly treats the images as redundant views of one problem and optimises for a single, locally consistent story. In other words, GPT-5 correctly explains one mechanism, but it fails at joint, cross-region reasoning over a curated set of complementary images, which is precisely what this multi-syndrome case is designed to test.

### G.4.2 GPT-5 Error Case Study: Longitudinal Cystic Pulmonary Tuberculosis

This MedThinkVQA item is a longitudinal case of a previously healthy nine-year-old boy with cystic pulmonary tuberculosis. The clinical course spans almost eleven months: initial presentation with fever, cough, dyspnoea and diffuse micronodular infiltrates; multiple bilateral spontaneous pneumothoraces requiring repeated chest drainage and intensive care; evolution on CT from diffuse micronodules and ground-glass opacities to extensive thin-walled cystic change with recurrent pneumothoraces; and, after prolonged anti-tuberculous therapy, near-complete radiologic recovery with only a few residual cysts and fibrotic sequelae. All time points are encoded as separate images but linked to a single question whose ground-truth diagnosis is *cystic pulmonary tuberculosis*.

**GPT-5 answer (incorrect).** For the five-option question (Pneumocystis *jirovecii* pneumonia; Lymphoid interstitial pneumonia; Langerhans cell histiocytosis; Cystic pulmonary tuberculosis; Birt–Hogg–Dubé syndrome), GPT-5 chose *Pneumocystis* jirovecii *pneumonia* and produced the following explanation:

> The imaging shows an initial diffuse interstitial/ground-glass process (crazy-paving appearance) followed by the development of numerous thin-walled cysts/pneumatoceles throughout both lungs and recurrent bilateral spontaneous pneumothoraces. This evolution – diffuse GGO progressing to widespread pneumatoceles with high risk of pneumothorax – is classic for *Pneumocystis jirovecii* pneumonia, especially in children with underlying or occult immunodeficiency. Langerhans cell histiocytosis typically affects young adult smokers with upper-lobe-predominant, irregular cysts and costophrenic angle sparing; lymphoid interstitial pneumonia is a chronic disease with lower-lobe cysts and not an acute febrile presentation; tuberculosis produces cavities rather than numerous thin-walled cysts; Birt–Hogg–Dubé syndrome presents in adults with basilar subpleural cysts without acute diffuse ground-glass opacities. Thus, *Pneumocystis jirovecii* pneumonia best fits the overall picture.

**Error analysis: ignoring the longitudinal time axis.** GPT-5's reasoning is internally consistent for an acute, immunodeficiency-related *Pneumocystis* pneumonia, but it fails as a longitudinal interpretation of this case. Several key aspects of the time series are either misread or ignored:

- **Disease duration and follow-up.** The patient is an immunocompetent nine-year-old boy followed over almost eleven months, with documented near-complete radiologic recovery after prolonged anti-tuberculous therapy. This long clinical evolution with structured follow-up CTs is far more typical of tuberculosis than of uncontrolled *Pneumocystis* infection, which in an undiagnosed immunodeficient child would be expected to progress or relapse rather than steadily resolve.

- **Full temporal chain, not a single snapshot.** GPT-5 effectively compresses the sequence "diffuse micronodules/ground-glass → extensive thin-walled cysts → gradual resolution" into a standard short-course template for *Pneumocystis* pneumonia. It focuses on the middle phase (GGO with cysts and pneumothoraces) and treats the early and late time points as redundant, rather than evidence of a slowly evolving, ultimately reversible granulomatous infection under long-term treatment.

- **Misconception about tuberculosis morphology.** The explanation assumes that tuberculosis "produces cavities rather than numerous thin-walled cysts," implicitly excluding cystic or pneumothorax-prone forms of TB. However, the discussion explicitly describes cystic pulmonary tuberculosis as a rare but recognised entity in which diffuse nodules and ground-glass change can evolve into confluent thin-walled cysts with recurrent pneumothoraces, followed by gradual radiologic improvement once therapy is effective.

- **Underuse of sequential improvement as a diagnostic cue.** A core signal in this longitudinal case is the progressive improvement across follow-up CTs: cysts shrink, pneumothoraces resolve, and only sparse cystic or fibrotic sequelae remain. Such a trajectory is strongly supportive of treated infection (here, tuberculosis) and less compatible with chronic interstitial or cystic diseases (such as LCH, LIP, Birt–Hogg–Dubé) that typically show persistent or progressive structural damage.

### G.4.3 GPT-5 ERROR CASE STUDY: MULTIMODAL-HYDATID

This MedThinkVQA item is a multimodal, multi-organ case of disseminated hydatid disease in a 19-year-old man presenting with seizures and low-grade fever. Cranial CT and MRI show a solitary parietal cystic lesion with mild peripheral enhancement and no diffusion restriction; chest X-ray and body CT demonstrate multiple, well-defined peripheral pulmonary nodules with water attenuation; thoracic ultrasound confirms the cystic nature of the lung lesions; and abdominal CT reveals a cystic hepatic lesion with peripheral calcification. Serology for *Echinococcus granulosus* confirms the diagnosis, and the ground-truth label for the item is *disseminated hydatid disease*.

**GPT-5 answer (incorrect).** For the five-option question (Disseminated hydatid disease; Lung: pulmonary septic emboli; Pulmonary metastases; Brain: cystic brain tumour; Brain abscess), GPT-5 selected *Brain abscess* and produced the following explanation:

> The clinical presentation of seizures and febricula along with laboratory findings of leukocytosis and neutrophilia suggests an infectious or inflammatory process. The absence of respiratory symptoms makes options related to pulmonary pathology less likely as the primary diagnosis. Given the neurological symptoms, the most plausible diagnosis would involve the brain. Between options D and E, brain abscess (E) is more likely to present acutely with seizures and fever, and is often associated with leukocytosis and neutrophilia, whereas a cystic brain tumor (D) might not typically present with such acute infectious symptoms. The imaging findings, although not described, would likely show a localized area of infection in the brain for an abscess, which aligns with the clinical picture provided.

**Error analysis: failure to integrate multimodal, multi-organ evidence.** GPT-5's reasoning treats this case as if it were a purely neuroradiologic vignette: it focuses on seizures, low-grade fever and inflammatory markers, implicitly assumes the key information is "ring-enhancing brain lesion plus infection," and then chooses brain abscess by comparing only options D and E. This single-modality shortcut neglects almost all of the structured multimodal evidence presented:

- **Pulmonary imaging is downgraded to "background".** Chest X-ray and chest CT clearly show numerous, well-defined, peripheral pulmonary nodules with water attenuation and no features of suppurative consolidation or infarction. Thoracic ultrasound further confirms that these nodules are true cysts (anechoic with posterior acoustic enhancement), a pattern much more typical of hydatid disease than septic emboli or metastases. GPT-5's statement that pulmonary options are "less likely" because of absent respiratory symptoms ignores that hydatid cysts are often asymptomatic in the lungs and that imaging, not symptoms, carries the main diagnostic weight here.

- **Hepatic cyst is completely ignored.** Body CT demonstrates a classic hydatid cyst in the left hepatic lobe with a well-defined cystic lesion and peripheral calcification of the pericyst. This second non-brain, non-lung organ involvement is a strong clue for systemic parasitic disease. GPT-5's explanation does not mention the liver at all, indicating that this modality and organ channel are effectively dropped from its reasoning.

- **Brain MRI is interpreted through a generic "ring-enhancement = abscess" template.** The brain lesion in this case is a solitary, CSF-like cyst with a thin rim, mild peripheral enhancement, no diffusion restriction, and only moderate oedema. These features, particularly the absence of diffusion restriction and the characteristic low-signal rim on T2-weighted images, are more consistent with a hydatid cyst than with a pyogenic abscess. GPT-5 instead imagines a typical abscess pattern and even states that the imaging findings "would likely" show a focal infection, revealing that it is reasoning from a mental template rather than actually integrating the provided MRI sequences.

- **Cross-organ pattern is never assembled.** The correct diagnosis requires noticing a triad: (i) multiple cystic pulmonary lesions, (ii) a calcified hepatic cyst, and (iii) a solitary brain cyst with hydatid-like MRI characteristics. Taken together, these represent a classic multi-organ, haematogenously disseminated parasitic infection. GPT-5 never composes this cross-organ, cross-modality picture; instead, it chooses the most salient single modality (brain MRI/CT) and maps the entire case to a focal intracranial infection.

### G.5 STEP-LEVEL EVALUATION CASE STUDY: BILATERAL SUBAREOLAR ABSCESSES

#### G.5.1 ERROR TAXONOMY FOR MODEL RESPONSES

- **Reasoning Error (Reasoning Err).** The imaging and clinical facts themselves are correctly stated, but the model misconstructs the causal chain or diagnostic logic, reaches an incorrect conclusion, selects an inappropriate differential diagnosis, or uses correct facts to support an incorrect judgment.

- **Image Understanding Error (Image Understanding Err).** The model misreads or hallucinates objective visual information that is directly apparent on the images (e.g., location, morphology, signal intensity, enhancement pattern, extent), and the error originates from image interpretation rather than downstream reasoning.

- **Clinical Scenario Error (Clinical Scenario Err).** The model misunderstands, misquotes, or inaccurately restates clinical information provided in the stem (e.g., symptoms, age, duration, physical findings, laboratory data), or treats information that is not given as if it were explicitly provided.

- **Medical Knowledge Error (Medical Knowledge Err).** The model violates general medical knowledge or textbook-level facts—such as typical manifestations of a lesion, epidemiology, key features for differential diagnosis, or common complications—in ways that are independent of how the specific images or clinical history are read.

**Clinical scenario and imaging summary.** A 39-year-old woman presented with unilateral, single-orifice nipple discharge (galactorrhoea) from the right breast and no fever, palpable mass, or skin inflammatory changes. Initial mammography (craniocaudal and oblique views) and ultrasound of both breasts were reported as normal. Galactography of the right breast showed an intraductal filling defect in a canalised nipple duct and raised the suspicion of an intraductal papilloma. Subsequent contrast-enhanced breast MRI revealed bilateral subareolar fluid-filled cavities without internal enhancement, with intensely enhancing peripheral capsules and surrounding inflammatory tissue. A repeat ultrasound, performed after the patient clarified that the discharge was actually bilateral, showed bilateral subareolar hypoechoic nodules with anechoic centres, consistent with abscesses. Cytology confirmed bilateral subareolar abscesses.

**GPT-5 diagnosis and explanation.** The multiple-choice options included galactocele, fat necrosis, intracystic papillomas, and subareolar abscesses (correct answer: subareolar abscesses). GPT-5 instead chose *intracystic papillomas* and produced the following free-text explanation:

> Unilateral, single-duct nipple discharge suggests a focal intraductal process. Mammography is essentially negative. Ultrasound shows a dilated retroareolar duct containing an intracystic solid nodule. MRI demonstrates a small enhancing subareolar intraductal lesion without surrounding inflammatory change. Findings are classic for an intraductal/intracystic papilloma, not a galactocele (no fat–fluid level), abscess (no pain, erythema or inflammatory changes), or fat necrosis.

**Step-level parsing and expert error annotation.** We use GPT-5 to decompose its own explanation into discrete reasoning steps, then ask a breast imaging expert to judge the factual correctness of each step and assign an error type when applicable. Error types follow our taxonomy: *Clinical Scenario Err*, *Image Understanding Err*, *Reasoning Err*, and *Medical Knowledge Err* as defined in Section G.5.1.

1. **Step 1:** "Unilateral, single-duct nipple discharge suggests a focal intraductal process."
   *Expert factual judgment:* **Incorrect**. *Error type:* **Clinical Scenario Err**.
   Although the original history was recorded as unilateral, single-orifice discharge, the case narrative later clarifies that the galactorrhoea is actually bilateral. GPT-5 treats the initial, incomplete history as definitive and over-anchors on a classic teaching pattern for intraductal papilloma, misrepresenting the true clinical scenario.

2. **Step 2:** "Mammography is essentially negative."
   *Expert factual judgment:* **Correct**. *Error type:* None.
   The report states that mammography did not show any pathological findings, so this step accurately reflects the imaging description.

3. **Step 3:** "Ultrasound shows a dilated retroareolar duct containing an intracystic solid nodule."
   *Expert factual judgment:* **Incorrect**. *Error type:* **Image Understanding Err**.
   In reality, the initial ultrasound was normal, and the follow-up ultrasound showed bilateral subareolar hypoechoic nodules with anechoic centres (abscesses), not a duct with an intracystic solid nodule. GPT-5 hallucinates a very specific retroareolar papilloma pattern directly from the images, so the primary error is misreading the imaging rather than subsequent reasoning.

4. **Step 4:** "MRI demonstrates a small enhancing subareolar intraductal lesion without surrounding inflammatory change."
   *Expert factual judgment:* **Incorrect**. *Error type:* **Image Understanding Err**.
   The MRI actually showed bilateral fluid-filled cavities without internal enhancement, surrounded by an intensely enhancing capsule and inflammatory tissue with moderate enhancement. GPT-5 inverts this pattern (solid enhancing nodule, no inflammation), again indicating a failure to correctly translate the described MRI findings into its intermediate reasoning.

5. **Step 5:** "These imaging findings are classic for an intraductal/intracystic papilloma."
   *Expert factual judgment:* **Incorrect**. *Error type:* **Reasoning Err**.
   Given the true imaging pattern (bilateral subareolar fluid collections with enhancing capsule and surrounding inflammation), the correct diagnosis is abscess. Here GPT-5 generalises from its misread ultrasound/MRI steps and commits a diagnostic reasoning error: it asserts a papilloma pattern even though the combination of bilateral disease, fluid-filled cavities, and inflammatory enhancement better matches abscess.

6. **Step 6:** "The findings are not compatible with a galactocele because there is no fat–fluid level."
   *Expert factual judgment:* **Correct as stated**. *Error type:* None.
   The expert accepts that there is no explicit fat–fluid level described. While the accompanying textual explanation overstates how definitively this excludes a galactocele, the literal step ("no fat–fluid level is seen") is factually consistent with the report, so no image or scenario error is assigned at the step text level.

7. **Step 7:** "The findings are not compatible with an abscess because there is no pain, erythema, or inflammatory changes."
   *Expert factual judgment:* **Incorrect**. *Error types:* **Reasoning Err**, **Clinical Scenario Err**, **Medical Knowledge Err**.
   This step bundles three distinct problems:
   - *Clinical Scenario Err*: It equates the absence of external pain, erythema, or skin signs with the absence of inflammation, ignoring the MRI description of surrounding inflammatory tissue and the later clarification of bilateral symptoms.
   - *Medical Knowledge Err*: It assumes that breast abscesses cannot present without overt systemic or cutaneous signs, which contradicts the broader discussion that non-puerperal central abscesses can be subtle.
   - *Reasoning Err*: It uses this flawed premise to exclude the true diagnosis (abscess), despite direct imaging and cytology evidence to the contrary.

8. **Step 8:** "The findings are not compatible with fat necrosis."
   *Expert factual judgment:* **Correct**. *Error type:* None.
   The described lesions are fluid-filled cavities with enhancing capsules and inflammatory tissue, without oil cysts or internal fat signal; this pattern is more characteristic of abscess than fat necrosis, so excluding fat necrosis at this step is reasonable.

# H  OPTION & DISCUSSION AUGMENTATION PROMPT

To ensure reproducibility, we document the exact prompts used for augmenting *Options* and expanding the *Discussion* in the medical multiple-choice QA setting.

## H.1  SYSTEM PROMPT

```
You are a careful medical QA assistant.

# Prompt for Option Generation

### Task
Given a medical multiple-choice question of the form "Select the single best diagnosis"
based on CLINICAL_HISTORY, several patient images, the current provided options,
the correct answer, and an existing discussion (including reasoning about the current options), please:

1. Generate additional incorrect options so that the total number of answer choices
   is exactly 5 (no more, no less).
2. Expand and refine the provided discussion, ensuring it thoroughly explains how
   to eliminate all incorrect answers and why the correct answer is most appropriate,
   using reasoning grounded in the CLINICAL_HISTORY and images.

### Suggested Approaches
1. Consider Erroneous Perspectives: Add distractors that misinterpret or
   overemphasize aspects of the CLINICAL_HISTORY or images.
2. Leverage Common Misconceptions: Create distractors based on common diagnostic
   errors or frequently confused conditions.
3. Logical Misdirection: Introduce distractors grounded in logical reasoning
   that appear plausible but are ultimately incorrect.

### General Requirements
1. Maintain Consistency: Ensure new options match the original ones in length,
   structure, and professional wording.
2. Avoid Oversimplified Distractors.
3. Ensure High Plausibility.
4. Expand Discussion:
   - Include reasoning for the newly generated distractors.
   - Strengthen explanations for ruling out incorrect answers.
   - Deepen justification for selecting the correct answer.

5. Final Output Format:
   Return valid JSON with exactly these fields: options (A-E), correct_answer, discussion.

### Important Output Rules
- Keep all *original* options text unchanged; only add new distractors
  to reach exactly five total options.
- Do NOT reorder existing options; append only the missing letters
  (e.g., add D/E) so that A-E are filled.
- The final correct_answer must correspond to the original correct option's text.
- No extra commentary outside the JSON body.
```

# I   DISCUSSION PRUNING PROMPT

This section documents the prompts used to prune *Discussion* paragraphs by removing references to extra differential diagnoses that are not among the allowed answer options.

## I.1   SYSTEM PROMPT

```
You are a careful clinical editor. Your job is to MINIMALLY edit a medical DISCUSSION.
Goal: remove references to extra differential diagnoses that appear in
DIF_DIAGNOSIS_LIST but are NOT among the five ALLOWED OPTIONS.
Preserve all content related to ALLOWED OPTIONS.
Keep the original clinical reasoning flow, tone, and meaning. Do not add new facts.

Rules:
1) NEVER delete information that relates to any ALLOWED_OPTIONS
   (even if an EXTRA item partially overlaps).
2) Remove sentences/clauses whose main role is to introduce, justify, or
   list items in EXTRA_TO_REMOVE.
   If a sentence mixes allowed and extra diagnoses, keep the allowed part
   and delete only the extra part, then fix grammar to remain fluent.
3) Keep general disease definitions, imaging/lab reasoning, and conclusions
   that support ALLOWED_OPTIONS.
4) Maintain coherence and clinical correctness; do NOT invent new claims.
5) Output strictly as JSON with one key: discussion_new.
6) If EXTRA_TO_REMOVE is empty, return the original discussion as discussion_new.
```

## I.2   USER PROMPT TEMPLATE

```
Edit the DISCUSSION by deleting only the parts about the extra differentials.

ALLOWED_OPTIONS (keep anything related to these):
<ALLOWED_OPTIONS_JSON>

DIF_DIAGNOSIS_LIST_CLEAN:
<DIF_DIAGNOSIS_LIST_CLEAN_JSON>

EXTRA_TO_REMOVE (delete content only about these):
<EXTRA_TO_REMOVE_JSON>

DISCUSSION:
```text
<DISCUSSION>
Return JSON: {"discussion_new": "..."}
```

## J  PROMPTS FOR DATA LEAKAGE AUDITING

### SYSTEM MESSAGE

You are a meticulous clinical QA auditor for multiple-choice diagnosis questions. Your job:
Given ONLY the CLINICAL HISTORY text and the list of candidate diagnosis OPTIONS, decide
whether the history text DIRECTLY REVEALS any option(s).

Definition of DIRECT REVEAL (diagnosis label appears in the text itself, not inferred):
• L3 Explicit label: the exact diagnosis name or ICD/standard label appears, or patterns like
  "Diagnosis: X", "biopsy-proven X".
• L2 Explicit synonym/acronym/eponym/foreign-language variant of the diagnosis label appears
  (e.g., "MI" for myocardial infarction; "Osler-Weber-Rendu" for HHT).
• L1 Explicit but uncertain mention of the diagnosis label (or its synonym/acronym/eponym):
  e.g., "?X", "r/o X", "rule out X", "query X", "suspected X", "possible/probable X",
  "consistent with X", "concern for X", "Hx of/known case of X".

NOT a leak: symptoms, signs, risk factors, imaging descriptors, or lab patterns that merely
SUGGEST a diagnosis. Only mark a leak if the diagnosis LABEL itself (or its standard
synonym/acronym/eponym) occurs in the text.

Use the OPTIONS solely as a dictionary of candidate labels and their widely-used
synonyms/acronyms/eponyms to search for DIRECT textual mentions. Do NOT infer diagnoses
from context. Do NOT mark based on reasoning.

For each leaked option, return:
- option_id, option_text
- overall leak_level (max severity across its evidences; L3>L2>L1)
- evidences: verbatim snippet(s) with [start,end) character indices into the EXACT Clinical
  history string
- a brief justification

If no option is leaked, set has_leak=false and provide non_leak_reason.

Return ONLY valid JSON following the required schema. No extra prose.

### USER MESSAGE (TEMPLATE)

CLINICAL HISTORY (use this exact string when computing char spans):
<<<<HISTORY>>>>
{CLINICAL_HISTORY}
<<<<END_HISTORY>>>>

OPTIONS (candidate diagnoses; DO NOT infer--use only as label dictionary):
A) {option_A_text}
B) {option_B_text}
C) {option_C_text}
D) {option_D_text}
E) {option_E_text}
... (continue as needed, preserving order)

Task: Identify ALL options (if any) that are directly revealed by the HISTORY text
under L1/L2/L3 definitions. Extract verbatim evidence snippet(s) and 0-based [start,end)
char spans into the exact HISTORY string above. If none, set has_leak=false.

## K  PROMPTS FOR DISCUSSION GENERATION

### SYSTEM PROMPTS

You are a board-certified radiologist. Given clinical history, imaging
findings, a differential diagnosis list, the final diagnosis, and one or
more images (with captions), write a Discussion with five sections:
Background; Clinical Perspective; Imaging Perspective; Outcome; Take Home
Message. Be accurate, concise, and grounded in the provided info.

Return strict JSON with keys exactly:
{
  "Background": "...",
  "Clinical Perspective": "...",
  "Imaging Perspective": "...",
  "Outcome": "...",
  "Take Home Message": "..."
}

Example of tone/structure (content is just an example; DO NOT copy text):

```
{
  "Background": "May and Thurner described for the first time in 1956
  a spur-like formation on the left common iliac vein in 22% of autopsies.
  May-Thurner syndrome, also known as Iliac Venous Compression Syndrome
  (IVCS), is a condition of venous compression by the overlying artery,
  usually the left common iliac vein by the right common iliac artery.",

  "Clinical Perspective": "This disease is reported to be more frequent
  in women and the main clinical presentation is deep vein thrombosis.
  The true prevalence of this condition is unknown, but some autopsies
  series reported 22% to 33%. May-Thurner syndrome is a progressive
  vascular disease with long-term disabling complications.",

  "Imaging Perspective": "Iliac vein compression, with or without
  thrombosis, should be treated if symptomatic. The procedure includes
  an ascending venogram through the iliac vein to show the stenotic area.
  A guidewire is advanced through the lesion and a stent is than placed
  over-the-wire.",

  "Outcome": "Since 1995 venous stents have been placed into the narrowed
  vein area. Stents seem to be beneficial, improving the clinical outcome
  and the quality of life of these patients.",

  "Take Home Message": "If a patient has discomfort, swelling or deep
  venous thrombosis (DVT), in the iliofemoral vein territory, especially
  on the left side think about May-Thurner syndrome."
}
```

# L   LLM JUDGE PROMPT

## L.1   SYSTEM PROMPT

```
You are an evaluator for radiology case analyses. Judge the correctness of each step
based on the provided context (Clinical history, Captions, Imaging findings, Discussion)
and relevant teaching value/domain knowledge.
Rules:
1) Evaluate whether each step is correct or reasonably supported; reasonable analysis counts as correct.
2) Mark True if the step is explicitly supported, correctly implied, or logically reasonable given the context
   and your teaching value/domain knowledge.
3) Mark False only if the step is clearly wrong, contradictory, or cannot be reasonably inferred from either
   the context or standard domain knowledge.
4) Ignore style, redundancy, or reasoning quality--focus only on correctness.
5) Provide exactly one concise 1-2 sentence explanation per step.
6) Return ONLY JSON following the provided schema; one verdict per step, same order.
```

## L.2   USER PROMPT (TEMPLATE)

```
Task: For each step below, judge if it is supported by the provided context and relevant
teaching value/domain knowledge.

- Title: {{title}}
- Clinical history: {{clinical_history}}
- Imaging findings: {{imaging_findings}}
- Discussion: {{discussion}}
- Captions (all):
{{captions_block}}   # e.g., lines like "- {{caption_i}}"; if none, use "(none)"

Steps to judge (in order):
{{steps_block}}      # e.g., "1. {{step_1}}\n2. {{step_2}}\n..."

Output strictly as JSON; one verdict per step in the same order, using this schema:
{
  "verdicts": [
    {
      "is_factual": true,
      "explanation": "A brief, self-contained justification (1-2 sentences). If true,
      mention supporting phrase(s) from the context when possible; if false, state the
      contradiction or 'not supported by the provided context'. (2-300 chars)"
    }
    // ... one object per step, in order
  ]
}
```

## L.3   LLM as Judge for Case Discussions

```
You are a board-certified radiologist tasked with evaluating the factual
correctness of radiology case discussions.

Judge the correctness of each sentence from the Discussion section
(Background / Clinical Perspective / Imaging Perspective / Outcome /
Take-Home) based on the provided case context (Clinical history, Imaging
findings, Differential list), the image captions, and the images themselves.

Rules:
1) Mark True if the sentence is explicitly supported, correctly implied,
   or logically reasonable given the context and standard domain knowledge.
2) Mark False only if clearly wrong, contradictory, or not reasonably
   inferable.
3) Ignore style and redundancy--focus only on correctness.
4) Provide exactly one concise 1-2 sentence explanation per sentence.
5) Return ONLY JSON for the schema below.

Return STRICT JSON with this schema:
{
  "sentence_judgments": {
    "<sentence_key>": {
      "text": "<original sentence>",
      "factual": true|false,
      "explanation": "<ONE concise 1-2 sentence explanation>"
    }
  }
}
```

## L.4   Rubric Evaluation Prompt

```
You are a board-certified radiologist tasked with evaluating the quality
of radiology case discussions.

TASK: Evaluate the Discussion section of the provided radiology case
using a standardized rubric.

MATERIALS PROVIDED:
- Clinical history and imaging findings
- Differential diagnosis list
- Medical images with captions
- Discussion section (containing: Background, Clinical perspective,
  Imaging perspective, Outcome, Take-Home messages)

EVALUATION INSTRUCTIONS:
1. Read the entire Discussion section carefully
2. Score each of the 5 rubric criteria on a 0-2 scale.
3. For each rubric score, provide a brief 1-2 sentence justification
4. Calculate total score (sum of all 5 rubrics, range 0-10)

FOCUS ON:
- Medical accuracy and evidence-based content
- Completeness of information
- Educational value for radiology trainees
- Clear communication of key concepts
- Integration of clinical and imaging perspectives

OUTPUT FORMAT:
Return ONLY a valid JSON object following the specified schema.
Do not include any additional text or explanations outside the
JSON structure.

Return STRICT JSON with this schema:
{
  "rubric_scores": {
    "rubric_1_disease_overview": {"score": 0|1|2, "explanation": "<1-2 sentences>"},
    "rubric_2_clinical_pathophysiology": {"score": 0|1|2, "explanation": "<1-2 sentences>"},
    "rubric_3_imaging": {"score": 0|1|2, "explanation": "<1-2 sentences>"},
    "rubric_4_reasoning_differentials": {"score": 0|1|2, "explanation": "<1-2 sentences>"},
    "rubric_5_transferable_learning": {"score": 0|1|2, "explanation": "<1-2 sentences>"},
    "total": 0-10
  }
}
```

## M  ADDITIONAL EVALUATION TABLES FOR TEXT-SOLVABLE CASES

All results below are evaluated on the same **raw test set of 2,859 items**. The four-model joint-correct intersection contains **1,074** items, and the full per-model counts are summarized in Table 10.

**Table 10:** Per-model accuracy and four-model joint-correct intersection on the 2,859-item raw test set.

| Model | Correct | Total | Accuracy |
|---|---|---|---|
| Qwen3-Next-80B-A3B-Instruct | 1,657 | 2,859 | 0.580 (57.96%) |
| gpt-oss-120b | 1,747 | 2,859 | 0.611 (61.11%) |
| medgemma-27b-text-it | 1,644 | 2,859 | 0.575 (57.50%) |
| Llama-3.3-70B-Instruct | 1,613 | 2,859 | 0.564 (56.42%) |
| Four-model joint-correct intersection | 1,074 | 2,859 | 0.376 (37.57%) |

# N MODALITY STATS IN DATASET

Our dataset provides explicit modality annotations for every image. In both the training split and the test split, each case contains up to 49 image slots, and each slot stores a coarse modality label (`image{k}_modality`) and a fine-grained subtype (`image{k}_sub_modality`). We use these provided labels to summarize modality distributions at both the image and case levels.

## N.1 COARSE MODALITY CATEGORIES (LEVEL-1)

We use the Level-1 field `modality` as the coarse modality category. Both splits cover the same nine Level-1 modalities: *CT*, *MRI*, *X-ray*, *Ultrasound*, *Non-modality / Workflow / Post-processing*, *Pathology*, *Clinical photography*, *Nuclear medicine & Molecular imaging*, *Endoscopy*.

## N.2 PER-IMAGE MODALITY DISTRIBUTION

Table 11 reports the per-image distribution of Level-1 modalities. The training split contains 47,571 images, while the test split contains 5,845 images.

**Table 11:** Per-image distribution of coarse (Level-1) modalities in the training and test splits. Counts are absolute image counts, and percentages are relative to the total number of images in each split.

| Modality | Train images | Train (%) | Test images | Test (%) |
|---|---|---|---|---|
| CT | 17,280 | 36.32 | 1,929 | 33.00 |
| MRI | 15,026 | 31.59 | 2,179 | 37.28 |
| X-ray | 6,337 | 13.32 | 540 | 9.24 |
| Ultrasound | 3,989 | 8.39 | 609 | 10.42 |
| Non-modality / Workflow / Post-processing | 2,434 | 5.12 | 230 | 3.93 |
| Pathology | 1,135 | 2.39 | 151 | 2.58 |
| Clinical photography | 674 | 1.42 | 73 | 1.25 |
| Nuclear medicine & Molecular imaging | 534 | 1.12 | 114 | 1.95 |
| Endoscopy | 162 | 0.34 | 20 | 0.34 |
| Total | 47,571 | 100.00 | 5,845 | 100.00 |

CT and MRI dominate both splits, accounting for 67.91% of training images and 70.28% of test images, followed by X-ray and Ultrasound.

## N.3 PER-CASE MODALITY COVERAGE

Because cases can contain multiple images, Table 12 additionally reports the fraction of cases that contain at least one image from each Level-1 modality.

**Table 12:** Per-case coverage of coarse (Level-1) modalities. A case is counted for a modality if it contains at least one image annotated with that modality. Percentages are relative to the total number of cases in each split.

| Modality | Train cases | Train (%) | Test cases | Test (%) |
|---|---|---|---|---|
| CT | 4,552 | 61.96 | 455 | 63.19 |
| MRI | 3,144 | 42.79 | 393 | 54.58 |
| X-ray | 2,749 | 37.42 | 248 | 34.44 |
| Ultrasound | 1,648 | 22.43 | 220 | 30.56 |
| Non-modality / Workflow / Post-processing | 1,411 | 19.21 | 151 | 20.97 |
| Pathology | 571 | 7.77 | 78 | 10.83 |
| Clinical photography | 450 | 6.12 | 48 | 6.67 |
| Nuclear medicine & Molecular imaging | 280 | 3.81 | 50 | 6.94 |
| Endoscopy | 105 | 1.43 | 16 | 2.22 |
| Total | 7,347 | 100.00 | 720 | 100.00 |

## N.4 FINE-GRAINED SUB-MODALITY DISTRIBUTION (LEVEL-2)

Each image also has a fine-grained `sub_modality` label capturing protocol or acquisition type (e.g., contrast-enhanced CT, diffusion MRI, catheter angiography/DSA, mammography, hybrid PET-CT, and pathology slide types). The training split contains 47 distinct sub-modalities, while the test split contains 42; all test sub-modalities are covered by the training split. Table 13 reports per-image frequencies for every Level-2 category.

**Table 13:** Per-image distribution of fine-grained (Level-2) sub-modalities, grouped by their corresponding coarse (Level-1) modality.

| Modality | Sub-modality | Train images | Train (%) | Test images | Test (%) |
|---|---|---|---|---|---|
| CT | Contrast-enhanced CT | 8,822 | 18.54 | 1,041 | 17.81 |
| CT | Non-contrast CT | 4,861 | 10.22 | 503 | 8.61 |
| CT | CT Angiography | 1,538 | 3.23 | 132 | 2.26 |
| CT | HRCT / Thin-slice CT | 1,399 | 2.94 | 202 | 3.46 |
| CT | Other_CT | 632 | 1.33 | 43 | 0.74 |
| CT | CT Perfusion | 28 | 0.06 | 8 | 0.14 |
| MRI | Conventional MRI | 13,006 | 27.34 | 1,852 | 31.69 |
| MRI | Diffusion MRI | 1,178 | 2.48 | 223 | 3.82 |
| MRI | MR Angiography / Venography | 384 | 0.81 | 34 | 0.58 |
| MRI | Other_MRI | 230 | 0.48 | 27 | 0.46 |
| MRI | Perfusion MRI | 133 | 0.28 | 25 | 0.43 |
| MRI | MR Spectroscopy | 88 | 0.18 | 16 | 0.27 |
| MRI | Functional MRI | 7 | 0.01 | 2 | 0.03 |
| X-ray | Plain radiograph | 3,541 | 7.44 | 315 | 5.39 |
| X-ray | Catheter angiography / DSA | 1,402 | 2.95 | 94 | 1.61 |
| X-ray | Fluoroscopy | 981 | 2.06 | 40 | 0.68 |
| X-ray | Mammography | 410 | 0.86 | 91 | 1.56 |
| X-ray | Other_Xray | 3 | 0.01 | 0 | 0.00 |
| Ultrasound | B-mode ultrasound | 2,797 | 5.88 | 405 | 6.93 |
| Ultrasound | Doppler ultrasound | 1,020 | 2.14 | 159 | 2.72 |
| Ultrasound | Contrast-enhanced ultrasound | 95 | 0.20 | 26 | 0.44 |
| Ultrasound | Interventional / Procedure US | 59 | 0.12 | 14 | 0.24 |
| Ultrasound | Elastography | 16 | 0.03 | 5 | 0.09 |
| Ultrasound | Other_US | 2 | 0.00 | 0 | 0.00 |
| Non-modality / Workflow / Post-processing | 3D post-processing | 1,139 | 2.39 | 81 | 1.39 |
| Non-modality / Workflow / Post-processing | Annotated figure / diagram | 651 | 1.37 | 82 | 1.40 |
| Non-modality / Workflow / Post-processing | Reconstruction / Image manipulation | 585 | 1.23 | 66 | 1.13 |
| Non-modality / Workflow / Post-processing | PACS / Teleradiology screenshot | 53 | 0.11 | 1 | 0.02 |
| Non-modality / Workflow / Post-processing | Other_Workflow | 6 | 0.01 | 0 | 0.00 |
| Pathology | Histology (H&E) | 714 | 1.50 | 91 | 1.56 |
| Pathology | Other_Pathology | 213 | 0.45 | 20 | 0.34 |
| Pathology | Immunohistochemistry | 188 | 0.40 | 38 | 0.65 |
| Pathology | Cytology | 20 | 0.04 | 2 | 0.03 |
| Clinical photography | External clinical photo | 341 | 0.72 | 35 | 0.60 |
| Clinical photography | Intraoperative photo | 300 | 0.63 | 35 | 0.60 |
| Clinical photography | Other_ClinicalPhoto | 19 | 0.04 | 2 | 0.03 |
| Clinical photography | Ophthalmic photo | 13 | 0.03 | 1 | 0.02 |
| Clinical photography | Dermoscopy | 1 | 0.00 | 0 | 0.00 |
| Nuclear medicine & Molecular imaging | Hybrid: PET-CT | 257 | 0.54 | 83 | 1.42 |
| Nuclear medicine & Molecular imaging | Planar scintigraphy | 158 | 0.33 | 10 | 0.17 |
| Nuclear medicine & Molecular imaging | PET | 55 | 0.12 | 13 | 0.22 |
| Nuclear medicine & Molecular imaging | Hybrid: SPECT-CT | 40 | 0.08 | 7 | 0.12 |
| Nuclear medicine & Molecular imaging | SPECT | 17 | 0.04 | 0 | 0.00 |
| Nuclear medicine & Molecular imaging | Hybrid: PET-MR | 7 | 0.01 | 1 | 0.02 |
| Endoscopy | GI endoscopy | 92 | 0.19 | 10 | 0.17 |
| Endoscopy | Other_Endoscopy | 40 | 0.08 | 8 | 0.14 |
| Endoscopy | Bronchoscopy | 30 | 0.06 | 2 | 0.03 |
| Total | - | 47,571 | 100.00 | 5,845 | 100.00 |

## N.5 PER-CASE MODALITY DIVERSITY

Beyond per-image counts, we characterize case-level multimodality using `modalities_count`, defined as the number of distinct Level-1 modalities present in a case. Table 14 summarizes its distribution. The training split contains 7,347 cases with an average of 2.03 modalities per case, while the test split contains 720 cases with an average of 2.30 modalities per case.

**Table 14:** Distribution of the number of distinct coarse (Level-1) modalities per case (`modalities_count`) in the training and test splits.

| # Modalities | Train cases | Train (%) | Test cases | Test (%) |
|---|---|---|---|---|
| 1 | 2,476 | 33.70 | 165 | 22.92 |
| 2 | 2,859 | 38.91 | 292 | 40.56 |
| 3 | 1,463 | 19.91 | 166 | 23.06 |
| 4 | 443 | 6.03 | 74 | 10.28 |
| 5 | 87 | 1.18 | 22 | 3.06 |
| 6 | 13 | 0.18 | 1 | 0.14 |
| 7 | 6 | 0.08 | 0 | 0.00 |
| Total | 7,347 | 100.00 | 720 | 100.00 |

In the training split, 72.61% of cases contain at most two modalities. The test split is more multimodal: 63.47% of cases have one or two modalities, and 36.53% contain three or more modalities.

Analogously, we define `sub_modalities_count` as the number of distinct Level-2 sub-modalities per case. The average is 2.68 for training and 3.09 for test. Table 15 provides the full distribution.

**Table 15:** Distribution of the number of distinct fine-grained (Level-2) sub-modalities per case (`sub_modalities_count`) in the training and test splits.

| # Sub-modalities | Train cases | Train (%) | Test cases | Test (%) |
|---|---|---|---|---|
| 1 | 1,593 | 21.68 | 94 | 13.06 |
| 2 | 2,152 | 29.29 | 183 | 25.42 |
| 3 | 1,819 | 24.76 | 196 | 27.22 |
| 4 | 1,051 | 14.31 | 132 | 18.33 |
| 5 | 441 | 6.00 | 66 | 9.17 |
| 6 | 183 | 2.49 | 30 | 4.17 |
| 7 | 78 | 1.06 | 15 | 2.08 |
| 8 | 20 | 0.27 | 3 | 0.42 |
| 9 | 5 | 0.07 | 1 | 0.14 |
| 10 | 4 | 0.05 | 0 | 0.00 |
| 11 | 1 | 0.01 | 0 | 0.00 |
| Total | 7,347 | 100.00 | 720 | 100.00 |

### N.6 COMMON MODALITY COMBINATIONS AT THE CASE LEVEL

We also examine which Level-1 modalities co-occur at the case level. Here, a modality combination is defined as the set of distinct Level-1 modalities present in a case. For the test split (720 cases), the 5 most frequent modality combinations are:

- CT + MRI (70 cases, 9.7%),
- CT (70 cases, 9.7%),
- MRI (68 cases, 9.4%),
- CT + X-ray (48 cases, 6.7%),
- MRI + Ultrasound (33 cases, 4.6%),

For the training split (7,347 cases), the 5 most frequent modality combinations are:

- MRI (944 cases, 12.8%),
- CT (885 cases, 12.0%),
- CT + X-ray (675 cases, 9.2%),
- CT + MRI (547 cases, 7.4%),
- X-ray (433 cases, 5.9%),

## O  LONGITUDINAL STUDIES IN MEDTHINKVQA

We annotate whether each case contains longitudinal follow-up imaging (i.e., multiple imaging time points for the same patient). In both the training split and the held-out test split, we provide: (i) is_longitudinal, (ii) timepoint_count, and (iii) interval_text, a free-text description of the follow-up interval or overall span between time points when available.

Table 16 summarizes the prevalence of longitudinal cases. The held-out test set contains 219 longitudinal cases out of 720 (30.4%). The training set contains 2,154 longitudinal cases out of 7,347 (29.3%). Aggregating both splits, MedThinkVQA includes 2,373 longitudinal cases out of 8,067 total cases (29.4%), indicating that nearly one third of the dataset requires reasoning over temporal disease evolution.

**Table 16:** Prevalence of longitudinal studies in MedThinkVQA.

| Split | # Cases | # Longitudinal | Share (%) |
|---|---|---|---|
| Train | 7,347 | 2,154 | 29.3 |
| Test | 720 | 219 | 30.4 |
| Overall | 8,067 | 2,373 | 29.4 |

Longitudinal cases are typically short sequences. Across longitudinal cases, the median timepoint_count is 2 and the mean is 2.33. About 24.4% of longitudinal cases contain three or more time points, with a maximum of 7 time points in training (8 in test). Table 17 reports the distribution of timepoint_count among longitudinal cases.

**Table 17:** Distribution of the number of imaging time points (timepoint_count) among longitudinal cases. Each cell shows # cases and the percentage relative to the longitudinal subset of the corresponding split.

| timepoint_count | Train | Test | Overall |
|---|---|---|---|
| 2 | 1,631 (75.7%) | 163 (74.4%) | 1,794 (75.6%) |
| 3 | 379 (17.6%) | 41 (18.7%) | 420 (17.7%) |
| 4 | 115 (5.3%) | 9 (4.1%) | 124 (5.2%) |
| 5+ | 29 (1.3%) | 6 (2.7%) | 35 (1.5%) |

For longitudinal cases, interval_text provides a free-text description of the follow-up interval(s) or overall temporal span between imaging time points. Interval descriptions are available for 74.3% of longitudinal cases (1,762 / 2,373), while the remaining 25.7% are marked as unknown/unspecified. Among cases with known interval descriptions, the most common timescales are months (36.7%) and years (22.3%). Table 18 summarizes coarse interval categories derived from interval_text.

**Table 18:** Coarse distribution of follow-up interval descriptions (interval_text) among longitudinal cases. Categories are assigned by keyword matching to temporal units and explicit calendar spans; percentages are relative to the longitudinal subset of each split.

| Interval category | Train | Test | Overall |
|---|---|---|---|
| Unknown / Unspecified | 555 (25.8%) | 56 (25.6%) | 611 (25.7%) |
| Hours | 52 (2.4%) | 12 (5.5%) | 64 (2.7%) |
| Days | 327 (15.2%) | 29 (13.2%) | 356 (15.0%) |
| Weeks | 210 (9.7%) | 16 (7.3%) | 226 (9.5%) |
| Months | 577 (26.8%) | 69 (31.5%) | 646 (27.2%) |
| Years | 363 (16.9%) | 30 (13.7%) | 393 (16.6%) |
| Date range / Span | 56 (2.6%) | 6 (2.7%) | 62 (2.6%) |
| Other | 14 (0.6%) | 1 (0.5%) | 15 (0.6%) |

We additionally estimate the maximum follow-up interval/span described in `interval_text` by parsing explicit numeric durations (e.g., "6 months", "3 years") and calendar spans (e.g., "1971 to 2000"); when multiple time references occur in the same text, we take the maximum. This produces a parsable maximum follow-up for 69.7% of longitudinal cases (1,653 / 2,373). The maximum parsed follow-up interval reaches 60 years in training (20 years in test). Table 19 reports the distribution of these maximum follow-up intervals.

**Table 19:** Distribution of the maximum follow-up interval/span (estimated from `interval_text`) among longitudinal cases. Each cell shows # cases and the percentage relative to the longitudinal subset of the corresponding split.

| Max follow-up | Train | Test | Overall |
|---|---|---|---|
| ≤ 1 week | 230 (10.7%) | 30 (13.7%) | 260 (11.0%) |
| 1 week–1 month | 285 (13.2%) | 20 (9.1%) | 305 (12.9%) |
| 1–6 months | 431 (20.0%) | 56 (25.6%) | 487 (20.5%) |
| 6–12 months | 200 (9.3%) | 22 (10.0%) | 222 (9.4%) |
| 1–3 years | 191 (8.9%) | 14 (6.4%) | 205 (8.6%) |
| 3–5 years | 66 (3.1%) | 8 (3.7%) | 74 (3.1%) |
| > 5 years | 94 (4.4%) | 6 (2.7%) | 100 (4.2%) |
| Unparsed / Unknown | 657 (30.5%) | 63 (28.8%) | 720 (30.3%) |

| Pairwise comparison | Cohen's $\kappa$ |
|---|---|
| Expert 1 vs. Expert 2 | 0.822833 |
| Expert 1 vs. LLM judge | 0.838357 |
| Expert 2 vs. LLM judge | 0.701566 |

**Table 20:** Inter-rater reliability on step factuality (Cohen's $\kappa$). High agreement with Expert 1 and substantial agreement with Expert 2 support the reliability of the LLM judge.

# P PROMPTS FOR STEPWISE EXPLANATION EXTRACTION

## P.1 SYSTEM PROMPT

```
You are a meticulous clinical reasoning editor. Convert a given explanation paragraph
into an ordered list of numbered steps that preserves the original meaning and evidence.
Rules:
1) Preserve content: do NOT introduce facts not present in the explanation.
2) Decompose into atomic inferences or observations -- each step one concise sentence
   (<= ~30 words).
3) Order steps to reflect the reasoning flow (e.g., findings -> interpretation -> decision).
4) Rewrite references like 'option A/B/C' into plain statements; avoid option letters.
5) If the explanation contrasts entities (e.g., 'X not Y'), separate them into distinct steps.
6) Use the same language as the explanation text (typically English).
7) If the explanation is very short, return a single clear step.
Return ONLY the JSON that matches the provided schema.
```

## P.2 USER PROMPT (TEMPLATE)

```
Task: Convert the following explanation into an ordered list of steps.

Context (for referent clarity only - do NOT add facts not present in the explanation):
- Title: {title}
- Clinical history: {clinical_history}
- Imaging findings: {imaging_findings}

Explanation to convert (source of truth):
<<<
{explanation}
>>>

Output strictly as JSON following the schema (no extra text).
```

Table 20 summarizes the pairwise agreement statistics underlying the human-versus-LLM-judge reliability discussion in the main text.

# Q    LLM JUDGE STATS

GPT-5 was evaluated on the **entire** test set, whereas the other three models were evaluated on a **random sample of 100** test cases due to cost and time constraints. Table 3 reports the aggregate main-text summary, while this appendix provides the per-split breakdowns. *Error-type coverage is computed over erroneous steps; since a step may bear multiple error labels, the percentages can exceed* 100%.

## Q.1    GPT-5 (FULL TEST SET WITH **6,425** STEPS )

**Correctly answered (*is_correct=True*).**

- Steps (with valid `is_factual`): **3,903**
- Step factual accuracy: **3311/3903 (84.83**%)
- Critical steps: **1,264**
- Critical-step factual accuracy: **1212/1264 (95.89**%)
- Erroneous steps (all): **592**
- Error-type coverage (among erroneous steps):
    - Reasoning Err: **167/592** (28.21%)
    - Image Understanding Err: **374/592** (63.18%)
    - Clinical Scenario Err: **53/592** (8.95%)
    - Medical Knowledge Err: **91/592** (15.37%)
    - Other/Unspecified: **60/592** (10.14%)
- Erroneous *critical* steps only: **52**
- Error-type coverage (among erroneous critical steps):
    - Reasoning Err: **14/52** (26.92%)
    - Image Understanding Err: **37/52** (71.15%)
    - Clinical Scenario Err: **6/52** (11.54%)
    - Medical Knowledge Err: **11/52** (21.15%)

**Incorrectly answered (*is_correct=False*).**

- Steps (with valid `is_factual`): **2,522**
- Step factual accuracy: **1605/2522 (63.64**%)
- Critical steps: **520**
- Critical-step factual accuracy: **390/520 (75.00**%)
- Erroneous steps (all): **917**
- Error-type coverage (among erroneous steps):
    - Reasoning Err: **416/917** (45.37%)
    - Image Understanding Err: **585/917** (63.79%)
    - Clinical Scenario Err: **138/917** (15.05%)
    - Medical Knowledge Err: **271/917** (29.55%)
    - Other/Unspecified: **9/917** (0.98%)
- Erroneous *critical* steps only: **130**
- Error-type coverage (among erroneous critical steps):
    - Reasoning Err: **57/130** (43.85%)
    - Image Understanding Err: **89/130** (68.46%)
    - Clinical Scenario Err: **16/130** (12.31%)
    - Medical Knowledge Err: **49/130** (37.69%)

## Q.2 INTERNVL3_5-14B_100_SAMPLE (100-SAMPLE SUBSET)

**Overall.** Total number of steps (all samples): **607**.

**Correctly answered (*is_correct=True*).**

- Steps (with valid `is_factual`): **247**
- Step factual accuracy: **189/247** (**76.52**%)
- Critical steps: **91**
- Critical-step factual accuracy: **88/91** (**96.70**%)
- Erroneous steps (all): **58**
- Error-type coverage (among erroneous steps):
    - Reasoning Err: **23/58** (39.66%)
    - Image Understanding Err: **29/58** (50.00%)
    - Clinical Scenario Err: **9/58** (15.52%)
    - Medical Knowledge Err: **24/58** (41.38%)
- Erroneous *critical* steps only: **3**
- Error-type coverage (among erroneous critical steps):
    - Reasoning Err: **0/3** (0.00%)
    - Image Understanding Err: **3/3** (100.00%)
    - Clinical Scenario Err: **0/3** (0.00%)
    - Medical Knowledge Err: **0/3** (0.00%)

**Incorrectly answered (*is_correct=False*).**

- Steps (with valid `is_factual`): **360**
- Step factual accuracy: **195/360** (**54.17**%)
- Critical steps: **61**
- Critical-step factual accuracy: **52/61** (**85.25**%)
- Erroneous steps (all): **165**
- Error-type coverage (among erroneous steps):
    - Reasoning Err: **104/165** (63.03%)
    - Image Understanding Err: **84/165** (50.91%)
    - Clinical Scenario Err: **33/165** (20.00%)
    - Medical Knowledge Err: **81/165** (49.09%)
- Erroneous *critical* steps only: **9**
- Error-type coverage (among erroneous critical steps):
    - Reasoning Err: **4/9** (44.44%)
    - Image Understanding Err: **6/9** (66.67%)
    - Clinical Scenario Err: **2/9** (22.22%)
    - Medical Knowledge Err: **2/9** (22.22%)

### Q.3  MEDGEMMA27B_100_SAMPLE (100-SAMPLE SUBSET)

**Overall.**  Total number of steps (all samples): **1,074**.

**Correctly answered (*is_correct=True*).**

- Steps (with valid `is_factual`): **376**
- Step factual accuracy: **285/376** (**75.80**%)
- Critical steps: **102**
- Critical-step factual accuracy: **97/102** (**95.10**%)
- Erroneous steps (all): **91**
- Error-type coverage (among erroneous steps):
    - Reasoning Err: **22/91** (24.18%)
    - Image Understanding Err: **50/91** (54.95%)
    - Clinical Scenario Err: **15/91** (16.48%)
    - Medical Knowledge Err: **36/91** (39.56%)
- Erroneous *critical* steps only: **5**
- Error-type coverage (among erroneous critical steps):
    - Reasoning Err: **3/5** (60.00%)
    - Image Understanding Err: **4/5** (80.00%)
    - Clinical Scenario Err: **0/5** (0.00%)
    - Medical Knowledge Err: **1/5** (20.00%)

**Incorrectly answered (*is_correct=False*).**

- Steps (with valid `is_factual`): **698**
- Step factual accuracy: **383/698** (**54.87**%)
- Critical steps: **114**
- Critical-step factual accuracy: **78/114** (**68.42**%)
- Erroneous steps (all): **315**
- Error-type coverage (among erroneous steps):
    - Reasoning Err: **156/315** (49.52%)
    - Image Understanding Err: **221/315** (70.16%)
    - Clinical Scenario Err: **72/315** (22.86%)
    - Medical Knowledge Err: **119/315** (37.78%)
- Erroneous *critical* steps only: **36**
- Error-type coverage (among erroneous critical steps):
    - Reasoning Err: **16/36** (44.44%)
    - Image Understanding Err: **22/36** (61.11%)
    - Clinical Scenario Err: **9/36** (25.00%)
    - Medical Knowledge Err: **14/36** (38.89%)

### Q.4 QWEN2.5VL-32B_100 (100-SAMPLE SUBSET)

**Overall.**    Total number of steps (all samples): **781**.

**Correctly answered (*is_correct=True*).**

- Steps (with valid `is_factual`): **337**
- Step factual accuracy: **274/337** (**81.31**%)
- Critical steps: **103**
- Critical-step factual accuracy: **100/103** (**97.09**%)
- Erroneous steps (all): **63**
- Error-type coverage (among erroneous steps):
    - Reasoning Err: **22/63** (34.92%)
    - Image Understanding Err: **36/63** (57.14%)
    - Clinical Scenario Err: **6/63** (9.52%)
    - Medical Knowledge Err: **31/63** (49.21%)
- Erroneous *critical* steps only: **3**
- Error-type coverage (among erroneous critical steps):
    - Reasoning Err: **0/3** (0.00%)
    - Image Understanding Err: **3/3** (100.00%)
    - Clinical Scenario Err: **0/3** (0.00%)
    - Medical Knowledge Err: **0/3** (0.00%)

**Incorrectly answered (*is_correct=False*).**

- Steps (with valid `is_factual`): **444**
- Step factual accuracy: **236/444** (**53.15**%)
- Critical steps: **67**
- Critical-step factual accuracy: **35/67** (**52.24**%)
- Erroneous steps (all): **208**
- Error-type coverage (among erroneous steps):
    - Reasoning Err: **130/208** (62.50%)
    - Image Understanding Err: **113/208** (54.33%)
    - Clinical Scenario Err: **52/208** (25.00%)
    - Medical Knowledge Err: **109/208** (52.40%)
- Erroneous *critical* steps only: **32**
- Error-type coverage (among erroneous critical steps):
    - Reasoning Err: **21/32** (65.62%)
    - Image Understanding Err: **26/32** (81.25%)
    - Clinical Scenario Err: **4/32** (12.50%)
    - Medical Knowledge Err: **11/32** (34.38%)

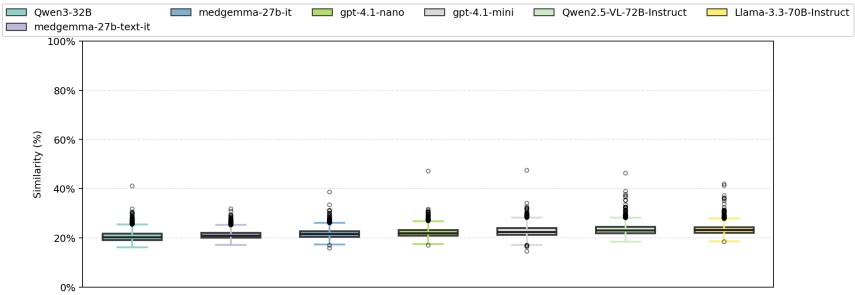

**Figure 18:** MELD similarity distributions on the MEDTHINKVQA test set across seven representative models. All distributions remain well below the 50% high-risk threshold, with similar behavior for text-only LLMs and VLMs.

## R    MELD-BASED DATA CONTAMINATION ANALYSIS (FULL DETAILS)

**Detector.**    We use MELD (Memorization Effects Levenshtein Detector) in a stricter, sliding-window form. For a model output $y$ and its corresponding question $x$, we compute normalized Levenshtein similarity over fixed-width windows on the longer string and take the maximum across windows. Scores are reported as percentages; higher values indicate longer, more verbatim copying. Following prior medical-QA practice Tang et al. (2025), samples with similarity $\geq 50\%$ are flagged as high-risk for contamination.

**Protocol.**    We run the exact inference setup used in our main experiments on the MEDTHINKVQA test set and apply MELD between each generated answer and its input question. We evaluate seven models spanning both LLMs and VLMs: Qwen3-32B, Med-Gemma-27B-*it*, Med-Gemma-27B-*text-it*, GPT-4.1-nano, GPT-4.1-mini, Qwen2.5-VL-72B-Instruct, and Llama-3.3-70B-Instruct.

**Results.**    Appendix Figure 18 plots the full distributions. Across all models, medians lie near ~20–24% with tight interquartile ranges, and the upper tails are short. Importantly, we do not observe any case with MELD similarity $\geq 50\%$; the largest outliers remain below that threshold. Text-only LLMs and VLMs exhibit highly similar distributions, suggesting that the presence of images does not drive overlap behavior.

**Context vs. prior benchmarks.**    MedAgentsBench Tang et al. (2025) reports broader spreads and heavier right tails (with many outliers above 50%) on several widely used QA datasets (e.g., MMLU, MedQA, MedMCQA). In contrast, MEDTHINKVQA shows uniformly low overlap and no high-similarity spikes, indicating a substantially lower contamination risk.

**Limitations.**    MELD is a surface-form detector; heavy paraphrasing or template-level memorization may evade detection. Our analysis should therefore be viewed as strong negative evidence for verbatim leakage rather than a proof of absence of all forms of contamination.

R.1  MELD AND OUR WINDOWED VARIANT

We first restate the original MELD procedure (Algorithm 1), and then present our implementation (Algorithm 2), which adds (i) a fixed-denominator Levenshtein ratio with respect to $|q_2|$, (ii) a length-$|q_2|$ sliding window over the model's continuation restricted to its early prefix, and (iii) length-aware bucketing and generation caps for efficient parallel decoding.

---

**Algorithm 1:** MELD (original reproduction)

---

**Data:** Generative model $g$; dataset $D$ of question–answer pairs; tokenizer $T$; threshold
        $Y \in [0, 1]$.
**Result:** $Z$: percentage (or average strength) of completions with overlap above $Y$.

1  Initialize an empty list $L$
2  **foreach** $(q, a) \in D$ **do**
3       Split $q$ into two halves: $q_1$ and $q_2$
4       Tokenize: $t_1 \leftarrow T(q_1)$ and $t_2 \leftarrow T(q_2)$
5       Set sampling temperature to 0 and pass $q_1$ as context to $g$
6       Let $k \leftarrow |t_2|$ and generate a continuation $x$ consisting of $k$ tokens from $g$
7       Compute the (paper-style) Levenshtein-based overlap ratio

$$\ell = \frac{\text{int}\big(\text{round}\big(\frac{2.0 \times M}{|q|} \times 100\big)\big)}{100},$$

     where $|q|$ is the total number of characters in both strings and $M$ is the number of matches.

8       **if** $\ell > Y$ **then**
9           |   append $\ell$ to $L$
10  $Z \leftarrow \text{mean}(L)$
11  **return** $Z$

---

**Algorithm 2:** MELD (ours, concise): windowed Levenshtein with length-aware batching

---

**Data:** Model $g$; dataset $D$; tokenizer $T$; threshold $Y$; cap multiplier $c \geq 1$; min gen tokens $m$;
        batch size $B$.
**Result:** $Z$ (near-exact rate), $\bar{\ell}$ (mean similarity).

1  **Build items.** For each $r \in D$: form text $q \leftarrow \mathsf{build}(r)$; if empty, continue. Tokenize $\text{ids} \leftarrow T(q)$;
    split at $h = \max(1, \lfloor |\text{ids}|/2 \rfloor)$; set $q_1 = T^{-1}(\text{ids}[:h])$, $q_2 = T^{-1}(\text{ids}[h:])$, $k = |\text{ids}| - h$. Collect
    tuples $(q_1, q_2, k, |q_2|)$.
2  **Bucket.** Group tuples into batches of size $\leq B$ with similar $k$ (length-aware).
3  **foreach** *batch* $b$ **do**
4       $G \leftarrow \max\big(m, c \cdot \max_{i \in b} k_i\big)$; set decoding (temp = 0, top-$p$ = 1, max tokens = $G$)
5       Generate in parallel $x_i \leftarrow g(q_{1,i})$ for all $i \in b$
6       **foreach** *item* $i$ *in* $b$ **do**
7           |   $L \leftarrow |q_{2,i}|$,
8           |   *region* $\leftarrow$ first $cL$ characters of $x_i$
9           |   $\rho_i \leftarrow \max\limits_{0 \leq j \leq |region| - L} \left(1 - \dfrac{\text{Lev}(region[j:j+L], q_{2,i})}{L}\right)$;
10           |   $s_i \leftarrow \mathbf{1}[\rho_i \geq Y]$
11  $Z \leftarrow \frac{1}{n} \sum_i s_i$;    $\bar{\ell} \leftarrow \frac{1}{n} \sum_i \rho_i$;
12  **return** $Z, \bar{\ell}$

---

## S  DISEASE CATEGORY BREAKDOWN

**Training set size**: $n = 7347$.
**Test set size**: $n = 720$.
**ICD coverage**: Train spans 22 chapters / 181 blocks / 743 categories; Test spans 19 chapters / 123 blocks / 308 categories.
**Train–test overlap**: 19/19 test chapters, 120/123 test blocks, 287/308 test categories appear in the training split.

SUBCATEGORY DETAIL (WITHIN EACH ICD-10 CHAPTER)

*Percentages are relative to the corresponding split; block counts are reported as Train / Test.*
**1. Certain infectious and parasitic diseases** (Train: $n = 364$; 5.0%, Test: $n = 55$; 7.6%)

- **1.1** A00–A09 Intestinal infectious diseases — 18 / 3
- **1.2** A15–A19 Tuberculosis — 88 / 9
- **1.3** A20–A28 Certain zoonotic bacterial diseases — 13 / 2
- **1.4** A30–A49 Other bacterial diseases — 64 / 12
- **1.5** A50–A64 Infections with a predominantly sexual mode of transmission — 6 / 1
- **1.6** A65–A69 Other spirochaetal diseases — 1 / 0
- **1.7** A70–A74 Other diseases caused by chlamydiae — 0 / 1
- **1.8** A80–A89 Viral infections of the central nervous system — 16 / 0
- **1.9** A92–A99 Arthropod-borne viral fevers and viral haemorrhagic fevers — 4 / 0
- **1.10** B00–B09 Viral infections characterized by skin and mucous membrane lesions — 7 / 0
- **1.11** B15–B19 Viral hepatitis — 2 / 1
- **1.12** B20–B24 Human immunodeficiency virus [HIV] disease — 5 / 4
- **1.13** B25–B34 Other viral diseases — 6 / 0
- **1.14** B35–B49 Mycoses — 40 / 8
- **1.15** B50–B64 Protozoal diseases — 5 / 2
- **1.16** B65–B83 Helminthiases — 87 / 12
- **1.17** B85–B89 Pediculosis, acariasis and other infestations — 1 / 0
- **1.18** B90–B94 Sequelae of infectious and parasitic diseases — 1 / 0

**2. Neoplasms** (Train: $n = 1934$; 26.3%, Test: $n = 202$; 28.1%)

- **2.1** C00–C14 Malignant neoplasms of lip, oral cavity and pharynx — 14 / 1
- **2.2** C15–C26 Malignant neoplasms of digestive organs — 187 / 13
- **2.3** C30–C39 Malignant neoplasms of respiratory and intrathoracic organs — 77 / 8
- **2.4** C40–C41 Malignant neoplasms of bone and articular cartilage — 51 / 8
- **2.5** C43–C44 Melanoma and other malignant neoplasms of skin — 12 / 1
- **2.6** C45–C49 Malignant neoplasms of mesothelial and soft tissue — 111 / 19
- **2.7** C50–C50 Malignant neoplasm of breast — 43 / 1
- **2.8** C51–C58 Malignant neoplasms of female genital organs — 45 / 2
- **2.9** C60–C63 Malignant neoplasms of male genital organs — 34 / 4
- **2.10** C64–C68 Malignant neoplasms of urinary tract — 74 / 4
- **2.11** C69–C72 Malignant neoplasms of eye, brain and other parts of central nervous system — 77 / 17

- **2.12** C73–C75 Malignant neoplasms of thyroid and other endocrine glands — 37 / 5
- **2.13** C76–C80 Malignant neoplasms of ill-defined, secondary and unspecified sites — 83 / 4
- **2.14** C81–C96 Malignant neoplasms, stated or presumed to be primary, of lymphoid, haematopoietic and related tissue — 161 / 13
- **2.15** C97–C97 Malignant neoplasms of independent (primary) multiple sites — 3 / 0
- **2.16** D00–D09 In situ neoplasms — 5 / 0
- **2.17** D10–D36 Benign neoplasms — 815 / 80
- **2.18** D37–D48 Neoplasms of uncertain or unknown behaviour — 105 / 22

**3. Diseases of the blood and blood-forming organs and certain disorders involving the immune mechanism** (Train: $n = 142$; 1.9%, Test: $n = 24$; 3.3%)

- **3.1** D50–D53 Nutritional anaemias — 2 / 0
- **3.2** D55–D59 Haemolytic anaemias — 15 / 1
- **3.3** D60–D64 Aplastic and other anaemias — 1 / 0
- **3.4** D65–D69 Coagulation defects, purpura and other haemorrhagic conditions — 10 / 0
- **3.5** D70–D77 Other diseases of blood and blood-forming organs — 79 / 11
- **3.6** D80–D89 Certain disorders involving the immune mechanism — 35 / 12

**4. Endocrine, nutritional and metabolic diseases** (Train: $n = 217$; 3.0%, Test: $n = 14$; 1.9%)

- **4.1** E00–E07 Disorders of thyroid gland — 21 / 0
- **4.2** E10–E14 Diabetes mellitus — 7 / 0
- **4.3** E15–E16 Other disorders of glucose regulation and pancreatic internal secretion — 3 / 0
- **4.4** E20–E35 Disorders of other endocrine glands — 81 / 7
- **4.5** E50–E64 Other nutritional deficiencies — 6 / 0
- **4.6** E65–E68 Obesity and other hyperalimentation — 5 / 0
- **4.7** E70–E90 Metabolic disorders — 94 / 7

**5. Mental and behavioural disorders** (Train: $n = 2$; 0.0%, Test: $n = 0$; 0.0%)

- **5.1** F10–F19 Mental and behavioural disorders due to psychoactive substance use — 1 / 0
- **5.2** F50–F59 Behavioural syndromes associated with physiological disturbances and physical factors — 1 / 0

**6. Diseases of the nervous system** (Train: $n = 359$; 4.9%, Test: $n = 37$; 5.1%)

- **6.1** G00–G09 Inflammatory diseases of the central nervous system — 40 / 6
- **6.2** G10–G14 Systemic atrophies primarily affecting the central nervous system — 11 / 1
- **6.3** G20–G26 Extrapyramidal and movement disorders — 18 / 1
- **6.4** G30–G32 Other degenerative diseases of the nervous system — 10 / 1
- **6.5** G35–G37 Demyelinating diseases of the central nervous system — 27 / 6
- **6.6** G40–G47 Episodic and paroxysmal disorders — 6 / 0
- **6.7** G50–G59 Nerve, nerve root and plexus disorders — 44 / 7
- **6.8** G60–G64 Polyneuropathies and other disorders of the peripheral nervous system — 5 / 0
- **6.9** G70–G73 Diseases of myoneural junction and muscle — 4 / 0

- **6.10** G80–G83 Cerebral palsy and other paralytic syndromes — 2 / 1
- **6.11** G90–G99 Other disorders of the nervous system — 192 / 14

**7. Diseases of the eye and adnexa** (Train: $n = 23$; 0.3%, Test: $n = 5$; 0.7%)

- **7.1** H00–H06 Disorders of eyelid, lacrimal system and orbit — 9 / 3
- **7.2** H15–H22 Disorders of sclera, cornea, iris and ciliary body — 0 / 1
- **7.3** H25–H28 Disorders of lens — 1 / 0
- **7.4** H30–H36 Disorders of choroid and retina — 1 / 1
- **7.5** H40–H42 Glaucoma — 1 / 0
- **7.6** H43–H45 Disorders of vitreous body and globe — 2 / 0
- **7.7** H46–H48 Disorders of optic nerve and visual pathways — 7 / 0
- **7.8** H49–H52 Disorders of ocular muscles, binocular movement, accommodation and refraction — 1 / 0
- **7.9** H55–H59 Other disorders of eye and adnexa — 1 / 0

**8. Diseases of the ear and mastoid process** (Train: $n = 22$; 0.3%, Test: $n = 2$; 0.3%)

- **8.1** H60–H62 Diseases of external ear — 3 / 0
- **8.2** H65–H75 Diseases of middle ear and mastoid — 7 / 1
- **8.3** H80–H83 Diseases of inner ear — 9 / 1
- **8.4** H90–H95 Other disorders of ear — 3 / 0

**9. Diseases of the circulatory system** (Train: $n = 759$; 10.3%, Test: $n = 58$; 8.1%)

- **9.1** I05–I09 Chronic rheumatic heart diseases — 2 / 0
- **9.2** I20–I25 Ischaemic heart diseases — 14 / 2
- **9.3** I26–I28 Pulmonary heart disease and diseases of pulmonary circulation — 42 / 5
- **9.4** I30–I52 Other forms of heart disease — 55 / 10
- **9.5** I60–I69 Cerebrovascular diseases — 120 / 10
- **9.6** I70–I79 Diseases of arteries, arterioles and capillaries — 368 / 20
- **9.7** I80–I89 Diseases of veins, lymphatic vessels and lymph nodes, not elsewhere classified — 155 / 11
- **9.8** I95–I99 Other and unspecified disorders of the circulatory system — 3 / 0

**10. Diseases of the respiratory system** (Train: $n = 222$; 3.0%, Test: $n = 23$; 3.2%)

- **10.1** J00–J06 Acute upper respiratory infections — 4 / 1
- **10.2** J09–J18 Influenza and pneumonia — 19 / 1
- **10.3** J30–J39 Other diseases of upper respiratory tract — 22 / 2
- **10.4** J40–J47 Chronic lower respiratory diseases — 22 / 1
- **10.5** J60–J70 Lung diseases due to external agents — 23 / 3
- **10.6** J80–J84 Other respiratory diseases principally affecting the interstitium — 50 / 9
- **10.7** J85–J86 Suppurative and necrotic conditions of lower respiratory tract — 5 / 0
- **10.8** J90–J94 Other diseases of pleura — 25 / 4
- **10.9** J95–J99 Other diseases of the respiratory system — 52 / 2

**11. Diseases of the digestive system** (Train: $n = 892$; 12.1%, Test: $n = 66$; 9.2%)

- **11.1** K00–K14 Diseases of oral cavity, salivary glands and jaws — 56 / 6
- **11.2** K20–K31 Diseases of oesophagus, stomach and duodenum — 115 / 6
- **11.3** K35–K38 Diseases of appendix — 50 / 1
- **11.4** K40–K46 Hernia — 106 / 7
- **11.5** K50–K52 Noninfective enteritis and colitis — 40 / 2
- **11.6** K55–K64 Other diseases of intestines — 237 / 20
- **11.7** K65–K67 Diseases of peritoneum — 56 / 5
- **11.8** K70–K77 Diseases of liver — 79 / 9
- **11.9** K80–K87 Disorders of gallbladder, biliary tract and pancreas — 131 / 9
- **11.10** K90–K93 Other diseases of the digestive system — 22 / 1

**12. Diseases of the skin and subcutaneous tissue** (Train: $n = 43$; 0.6%, Test: $n = 6$; 0.8%)

- **12.1** L00–L08 Infections of the skin and subcutaneous tissue — 16 / 0
- **12.2** L50–L54 Urticaria and erythema — 4 / 0
- **12.3** L60–L75 Disorders of skin appendages — 7 / 4
- **12.4** L80–L99 Other disorders of the skin and subcutaneous tissue — 16 / 2

**13. Diseases of the musculoskeletal system and connective tissue** (Train: $n = 524$; 7.1%, Test: $n = 61$; 8.5%)

- **13.1** M00–M03 Infectious arthropathies — 8 / 2
- **13.2** M05–M14 Inflammatory polyarthropathies — 24 / 5
- **13.3** M15–M19 Arthrosis — 6 / 1
- **13.4** M20–M25 Other joint disorders — 43 / 5
- **13.5** M30–M36 Systemic connective tissue disorders — 55 / 10
- **13.6** M40–M43 Deforming dorsopathies — 9 / 0
- **13.7** M45–M49 Spondylopathies — 19 / 2
- **13.8** M50–M54 Other dorsopathies — 10 / 1
- **13.9** M60–M63 Disorders of muscles — 55 / 3
- **13.10** M65–M68 Disorders of synovium and tendon — 38 / 2
- **13.11** M70–M79 Other soft tissue disorders — 73 / 13
- **13.12** M80–M85 Disorders of bone density and structure — 53 / 5
- **13.13** M86–M90 Other osteopathies — 90 / 11
- **13.14** M91–M94 Chondropathies — 33 / 1
- **13.15** M95–M99 Other disorders of the musculoskeletal system and connective tissue — 8 / 0

**14. Diseases of the genitourinary system** (Train: $n = 344$; 4.7%, Test: $n = 64$; 8.9%)

- **14.1** N00–N08 Glomerular diseases — 3 / 1
- **14.2** N10–N16 Renal tubulo-interstitial diseases — 37 / 6
- **14.3** N17–N19 Renal failure — 1 / 1

- **14.4** N20–N23 Urolithiasis — 15 / 1
- **14.5** N25–N29 Other disorders of kidney and ureter — 40 / 3
- **14.6** N30–N39 Other diseases of urinary system — 52 / 4
- **14.7** N40–N51 Diseases of male genital organs — 74 / 13
- **14.8** N60–N64 Disorders of breast — 34 / 14
- **14.9** N70–N77 Inflammatory diseases of female pelvic organs — 6 / 5
- **14.10** N80–N98 Noninflammatory disorders of female genital tract — 82 / 16

**15. Pregnancy, childbirth and the puerperium** (Train: $n = 35$; 0.5%, Test: $n = 7$; 1.0%)

- **15.1** O00–O08 Pregnancy with abortive outcome — 20 / 4
- **15.2** O10–O16 Oedema, proteinuria and hypertensive disorders in pregnancy, childbirth and the puerperium — 4 / 0
- **15.3** O20–O29 Other maternal disorders predominantly related to pregnancy — 0 / 1
- **15.4** O30–O48 Maternal care related to the fetus and amniotic cavity and possible delivery problems — 3 / 0
- **15.5** O60–O75 Complications of labour and delivery — 5 / 1
- **15.6** O85–O92 Complications predominantly related to the puerperium — 3 / 1

**16. Certain conditions originating in the perinatal period** (Train: $n = 25$; 0.3%, Test: $n = 0$; 0.0%)

- **16.1** P20–P29 Respiratory and cardiovascular disorders specific to the perinatal period — 6 / 0
- **16.2** P35–P39 Infections specific to the perinatal period — 1 / 0
- **16.3** P50–P61 Haemorrhagic and haematological disorders of fetus and newborn — 4 / 0
- **16.4** P70–P74 Transitory endocrine and metabolic disorders specific to fetus and newborn — 2 / 0
- **16.5** P75–P78 Digestive system disorders of fetus and newborn — 5 / 0
- **16.6** P90–P96 Other disorders originating in the perinatal period — 7 / 0

**17. Congenital malformations, deformations and chromosomal abnormalities** (Train: $n = 793$; 10.8%, Test: $n = 60$; 8.3%)

- **17.1** Q00–Q07 Congenital malformations of the nervous system — 96 / 10
- **17.2** Q10–Q18 Congenital malformations of eye, ear, face and neck — 25 / 2
- **17.3** Q20–Q28 Congenital malformations of the circulatory system — 228 / 11
- **17.4** Q30–Q34 Congenital malformations of the respiratory system — 48 / 7
- **17.5** Q35–Q37 Cleft lip and cleft palate — 1 / 0
- **17.6** Q38–Q45 Other congenital malformations of the digestive system — 65 / 10
- **17.7** Q50–Q56 Congenital malformations of genital organs — 41 / 2
- **17.8** Q60–Q64 Congenital malformations of the urinary system — 39 / 5
- **17.9** Q65–Q79 Congenital malformations and deformations of the musculoskeletal system — 126 / 9
- **17.10** Q80–Q89 Other congenital malformations — 118 / 4
- **17.11** Q90–Q99 Chromosomal abnormalities, not elsewhere classified — 6 / 0

**18. Symptoms, signs and abnormal clinical and laboratory findings, not elsewhere classified** (Train: $n = 36$; 0.5%, Test: $n = 1$; 0.1%)

- **18.1** R00–R09 Symptoms and signs involving the circulatory and respiratory systems — 1 / 0
- **18.2** R10–R19 Symptoms and signs involving the digestive system and abdomen — 11 / 0
- **18.3** R20–R23 Symptoms and signs involving the skin and subcutaneous tissue — 1 / 0
- **18.4** R30–R39 Symptoms and signs involving the urinary system — 3 / 1
- **18.5** R40–R46 Symptoms and signs involving cognition, perception, emotional state and behaviour — 1 / 0
- **18.6** R50–R69 General symptoms and signs — 5 / 0
- **18.7** R83–R89 Abnormal findings on examination of other body fluids, substances and tissues, without diagnosis — 1 / 0
- **18.8** R90–R94 Abnormal findings on diagnostic imaging and in function studies, without diagnosis — 12 / 0
- **18.9** R95–R99 Ill-defined and unknown causes of mortality — 1 / 0

**19. Injury, poisoning and certain other consequences of external causes** (Train: $n = 572$; 7.8%, Test: $n = 31$; 4.3%)

- **19.1** S00–S09 Injuries to the head — 38 / 1
- **19.2** S10–S19 Injuries to the neck — 30 / 0
- **19.3** S20–S29 Injuries to the thorax — 55 / 1
- **19.4** S30–S39 Injuries to the abdomen, lower back, lumbar spine and pelvis — 102 / 5
- **19.5** S40–S49 Injuries to the shoulder and upper arm — 30 / 2
- **19.6** S50–S59 Injuries to the elbow and forearm — 19 / 0
- **19.7** S60–S69 Injuries to the wrist and hand — 19 / 1
- **19.8** S70–S79 Injuries to the hip and thigh — 24 / 2
- **19.9** S80–S89 Injuries to the knee and lower leg — 34 / 8
- **19.10** S90–S99 Injuries to the ankle and foot — 18 / 2
- **19.11** T15–T19 Effects of foreign body entering through natural orifice — 40 / 1
- **19.12** T36–T50 Poisoning by drugs, medicaments and biological substances — 3 / 0
- **19.13** T51–T65 Toxic effects of substances chiefly nonmedicinal as to source — 14 / 1
- **19.14** T66–T78 Other and unspecified effects of external causes — 8 / 1
- **19.15** T79–T79 Certain early complications of trauma — 5 / 0
- **19.16** T80–T88 Complications of surgical and medical care, not elsewhere classified — 133 / 6

**20. External causes of morbidity and mortality** (Train: $n = 3$; 0.0%, Test: $n = 0$; 0.0%)

- **20.1** Y60–Y69 Misadventures to patients during surgical and medical care — 3 / 0

**21. Factors influencing health status and contact with health services** (Train: $n = 18$; 0.2%, Test: $n = 2$; 0.3%)

- **21.1** Z00–Z13 Persons encountering health services for examination and investigation — 10 / 0

- **21.2** Z40–Z54 Persons encountering health services for specific procedures and health care — 5 / 1
- **21.3** Z80–Z99 Persons with potential health hazards related to family and personal history and certain conditions influencing health status — 3 / 1

**22. Codes for special purposes** (Train: $n = 18$; 0.2%, Test: $n = 2$; 0.3%)

- **22.1** U00–U49 Provisional assignment of new diseases of uncertain etiology or emergency use — 18 / 2

*Note:* For each split, block counts within a chapter sum to the chapter total, and chapter totals across all chapters sum to the split size.

Abbreviations: NEC = not elsewhere classified.

# T    Rubric for Discussion Evaluation

## T.1    Rubric 1: Disease Overview & Core Definition (0–2 points)

**Focus:** Understanding of the disease's fundamental attributes, including: nomenclature, classification, and etiology.

- **0 points:** Unable to identify or define the disease.
- **1 point:** States the disease name, but classification or core etiology is vague or inaccurate.
- **2 points:** Accurately states the standard medical name, clearly defines its essential nature, and identifies principal etiologies or key risk factors.

## T.2    Rubric 2: Clinical Presentation & Pathophysiology (0–2 points)

**Focus:** How the disease manifests and its underlying mechanisms.

- **0 points:** Unable to describe any clinical features.
- **1 point:** Describes some common symptoms/signs but cannot explain the underlying pathophysiology, or omits critical features.
- **2 points:** Systematically outlines the typical clinical presentation and clearly explains the core pathophysiologic mechanisms.

## T.3    Rubric 3: Key Imaging Findings & Interpretation (0–2 points)

**Focus:** Recognition, description, and interpretation of disease-specific imaging features across modalities.

- **0 points:** Unable to describe any imaging characteristics.
- **1 point:** Provides only generic descriptors (e.g., "mass," "opacity") without modality-specific features (CT, MRI, radiography, ultrasound), or fails to distinguish key benign versus malignant signs.
- **2 points:** Clearly and accurately describes characteristic findings on one or more relevant modalities (e.g., morphology, attenuation/signal characteristics, margins, enhancement pattern, diffusion restriction), and interprets their clinical significance (e.g., stage, aggressiveness, complication risk).

## T.4    Rubric 4: Diagnostic Reasoning & Differential Diagnosis (0–2 points)

**Focus:** Integrating clinical and imaging data to reach a diagnosis and distinguish differential considerations.

- **0 points:** Unable to articulate a diagnostic approach.
- **1 point:** Arrives at the correct diagnosis but does not present a coherent, integrated reasoning process, or does not propose appropriate differential considerations.
- **2 points:** Clearly demonstrates how clinical information and imaging findings are synthesized to close the diagnostic loop, and lists at least two high-priority differential considerations with brief imaging discriminators (key features that separate each mimic from the index diagnosis).

## T.5    Rubric 5: Transferable Learning & Generalization (0–2 points)

**Focus:** Lessons that extend beyond a single case.

- **0 points:** Teaching points are confined to this case.
- **1 point:** Some generalizability is suggested but remains vague and lacks actionable takeaways.
- **2 points:** Clearly summarizes transferable learning points and explains how to avoid misinterpretation or improve diagnostic accuracy in similar future scenarios.

