# OpenReview forum: "Medical thinking with multiple images"
_ICLR.cc/2026/Conference — ICLR 2026 Poster_

### Official Review · Reviewer_qXc8 · 2025-10-21

**Soundness:** 2
**Presentation:** 1
**Contribution:** 3
**Rating:** 4
**Confidence:** 4

**Summary:**

This paper introduces an expert-annotated benchmark that probes the capabilities of multimodal LMMs on multi-image cross-view medical reasoning. Moreover, a step-wise evaluation protocol is proposed that can help pinpoint where models fail in their reasoning. State-of-the-art models are evaluated on the benchmark, and their mistakes are categorized into different error groups, highlighting the tendency for these models to fail on medical image understanding.

**Strengths:**

- The motivation of the paper is clear: it is important to assess multi-view capabilities of current models as it is closer to realistic diagnostic scenarios.
- The step-wise evaluation and breakdown into error categories is valuable. It helps us better understand where these models fail in complex medical reasoning tasks.
- The study on the utility of adding expert hints vs. self-generated captions is interesting and highlights the benefits of human-in-the-loop diagnostic workflows.

**Weaknesses:**

- Overall, the paper is hard to follow at parts due to multiple factors. First, the structure is unusual. The introduction could be more high-level without much details about methodology. Details about methodology is dispersed between the introduction and various other sections. Second, the paper assumes significant background knowledge on the source dataset used to create the benchmark. Is the ground truth reasoning trace part of the source dataset? What are the different sections of the source data and what kind of information do they exactly contain (e.g. "Integrated Imaging Summary" or "Image Hint")?

- The step-wise evaluation is a bit unclear to me. How exactly is the generated output broken down into these steps? Furthermore, a rigorous definition of the error categories would be helpful, e.g. what does "Clinical Scenario Error" entail?

- How is it supported in the work that multiple images are *necessary* to answer the questions accurately? If the problems can be tackled by a single view, then the key claim of probing cross-view synthesis is not well-supported.

- Human performance is missing in the benchmark, making it difficult to gauge the gap between SOTA models and human experts on cross-view medical reasoning.

- It is unclear what is the practical use-case of generating the teaching note in this benchmark. It seems only loosely connected to cross-view medical reasoning.

**Questions:**

- I would recommend restructuring the paper with clearer delineation between different sections.
- How are steps defined in the step-wise evaluation and how the error categories are defined and determined?
- How is the benchmark strictly probing multi-view reasoning? What happens if we remove some images? If the benchmark is truly probing cross-view capabilities, this would significantly degrade performance.
- What is the expert performance on the benchmark?

Minor:
- Table number is missing on line 362.

---

> ### Author Response · Authors · 2025-11-27
> **Response to Reviewer qXc8**
>
> We thank the reviewer for the insightful comments and address each point in detail below. In addition, all major revisions have been summarized in the **General Comment (Guide for Paper Revision)**, where you can also see how we incorporated the rebuttal responses into the updated manuscript.
>
> **Weakness & Question1: Structure and Dataset Background.**
>
> The Introduction is now purely high-level (motivation and contributions), and all technical details on dataset construction and evaluation are moved to Sections 3–4. Section 3.1 and Appendix C now explicitly describe the Eurorad source fields and show, via one full JSON example plus six case studies, how Clinical Scenario, Image Hint (per-image findings), Integrated Imaging Summary, and Teaching Note are derived from expert-authored text, making clear that all ground-truth reasoning traces come from the source corpus rather than being synthesized.
>
> **Weakness & Question2: Step-wise evaluation and error taxonomy.**
>
> Sections 4.2–4.3 and Appendix C.5 now explain how model explanations are decomposed by GPT-5-mini into atomic steps and then labeled by GPT-5 and two clinicians for factuality, criticality, and one (or more) of four error types adapted from MedXpertQA[1]: clinical-scenario, image-understanding, medical-knowledge, and reasoning errors. We include a worked step-level case that illustrates how steps are defined and what “Clinical Scenario Error” and the other categories mean in practice.

---

> > ### Author Response · Authors · 2025-11-27
> > **Response to Reviewer qXc8**
> >
> > **Weakness & Question3, 4: Multi-view Dependence and Expert Performance**
> >
> > In the revised manuscript we now report a dedicated human-expert study in **Section 4.4** and **Tables 3–4**. Two board-certified clinicians (a diagnostic radiologist and an academic surgeon) each solved 96 randomly sampled MedThinkVQA test items under the same MCQ protocol as the VLMs (clinical history plus all images, five options), achieving 74/96 correct (77.10% accuracy), while the strongest VLMs on the same subset reach only 55–56% (Gemini-2.5-Pro 55.67%, GPT-5 55.21%), showing that experienced specialists still clearly outperform current models on our benchmark (Section 4.4, Table 3). In a second audit round on the same 96 cases, with full access to captions, per-image findings, the case-level imaging summary, teaching discussion, and the ground-truth answer, the experts flagged only 2/96 items (2.1%) as possibly inconsistent and marked 18/96 (18.8%) as “very difficult”. At the image level, 65/96 cases (463/463 images) were judged to have all images supportive, and in the remaining 31 cases 222/315 images were supportive, for an overall 685/778 supportive images (88.05%; Table 4). Thus, while MedThinkVQA does include some redundant views, expert audit shows that the vast majority of images are supportive and contradictory or inconsistent information is **extremely rare**, implying that only about 12% of images are non-supportive or redundant.
> >
> > To directly test whether extra images help or just add noise, we added an image-subsampling experiment in **Fig. 5**: for several models (MedGemma-27B and GPT-5 series), accuracy increases monotonically as the visible image ratio grows from 0 to 1.0. All models start with the random 20% baseline when no images are shown. Taken together, the expert audit and Fig. 5 indicate that additional views are rarely “pure noise” and generally carry useful diagnostic evidence, while a realistic minority of non-supportive views remains, so MedThinkVQA both reflects clinical practice (radiologists scrolling through many images and selecting key views) and encourages models to integrate multiple supportive images while learning to down-weight redundant ones during training and evaluation.
> >
> > | Model / Expert | Correct | Incorrect | ACC (%) |
> > |----------------|---------|-----------|---------|
> > | Human experts  | 74      | 22        | 77.10   |
> > | Gemini-2.5-pro | 54      | 42        | 55.67   |
> > | GPT-5          | 53      | 43        | 55.21   |
> > | Claude 4.0     | 47      | 49        | 48.96   |
> > | Lingshu-32B    | 42      | 54        | 43.75   |
> >
> > Table 1: Human expert baseline vs. VLMs on the same 96-case subset
> >
> > | Case group                   | #Cases | #Images | #Supportive imgs | Supportive ratio (%) |
> > |------------------------------|--------|---------|------------------|----------------------|
> > | All images supportive        | 65     | 463     | 463              | 100.00               |
> > | Mixed supportive / redundant | 31     | 315     | 222              | 70.48                |
> > | **Overall (96 cases)**       | **96** | **778** | **685**          | **88.05**            |
> >
> > Table 2: Round 2 expert audit: image-level redundancy vs. support

---

> > > ### Author Response · Authors · 2025-11-27
> > > **Response to Reviewer qXc8**
> > >
> > > **Weakness 5: Teaching-note Generation Use-case.**
> > >
> > > Sections 3.4 and 4.2–4.3 clarify that the Eurorad “Discussion” sections are genuine radiology teaching notes, and our Medical Education Case Discussion task explicitly measures a model’s ability to turn the same multi-image evidence and final diagnosis into an educational explanation. We evaluate this with sentence-level factuality checks and a five-dimension 0–2 rubric (disease overview, clinical pathophysiology, imaging, reasoning and differentials, transferable learning), and we validate the automatic scores with a small human study, showing that this task is tightly coupled to cross-view reasoning rather than being an unrelated generation objective.
> > >
> > > In addition, **Section 4.4** and **Tables 3–4** report a human-expert study on 96 randomly sampled test cases: two board-certified clinicians achieve 77.10% accuracy under the same MCQ protocol—clearly above the best VLMs but still far from perfect—and label 18/96 cases (18.8%) as “very difficult”. In clinical practice, precisely such difficult or representative cases are typically written up and retained as teaching material; this mirrors the role of Eurorad “Discussion” notes and underscores that our teaching-note generation task targets a realistic and practically valuable use-case: helping clinicians and educators turn challenging multi-image cases into high-quality teaching content. We have incorporated this clarification into the revised **Section 3.4**.
> > >
> > > We hope these revisions and new analyses address the concerns about paper structure, step-wise evaluation, multi-view dependence, human performance, and the practical role of the teaching-note task.
> > >
> > > **References**
> > > [1] Yuxin Zuo, Shang Qu, Yifei Li, Zhangren Chen, Xuekai Zhu, Ermo Hua, Kaiyan Zhang, Ning Ding, and Bowen Zhou. MedXpertQA: Benchmarking expert-level medical reasoning and understanding. arXiv preprint arXiv:2501.18362, 2025.

---

### Official Review · Reviewer_prRm · 2025-10-23

**Soundness:** 4
**Presentation:** 3
**Contribution:** 4
**Rating:** 6
**Confidence:** 4

**Summary:**

This paper introduces MedThinkVQA, a benchmark built from Eurorad teaching cases, each pairing a clinical scenario with multi-image studies and expert reasoning artifacts. The task is framed as multi-choice diagnosis with a multi-step “think-with-images” pipeline: per-image findings, case-level integrated summary, differential-diagnosis reasoning, and a long-form medical education discussion generation. The evaluation goes beyond accuracy with stepwise checks, error-type tags, ROUGE/RadCliQ, and rubric-based/LLM-judge scoring of discussions, with human studies supporting reliability.

**Strengths:**

1. The problem formulation is well-motivated: prior medical VQA sets tend to be single-image, answer-centric, or automatically labeled; here the authors emphasize cross-view fusion and expert-authored intermediate signals that better mirror real diagnostic practice.

2. Dataset quality and evaluation design are thoughtfully engineered. The paper details option sets derivation from expert differentials, textual leakage detection, pruning of items solvable by text-only LLMs, mitigation of surface biases, and wide coverage spanning 20/22 ICD-10 chapters. The evaluation is also staged and fine-grained.

3. Engineering efforts in data curation and model evaluation bring concrete analysis and conclusions.

Generally, this is a good paper and I did not see major flaws.

**Weaknesses:**

1. Dataset statistics are inconsistent. Both the abstract and Table 1 state an average of 6.51 images per case, yet Line 200 in Sec. 3.1 mentions 8.3.

2. Some details about the dataset are missing. For example, how are multiple images in one case gathered? Are they longitudinal studies from the same patient, or complementary imaging modalities (X-ray, CT, MR, etc), or both? If both are involved, I suggest analyzing them separately since two scenarios assess different capabilities.

3. The evaluation relies on commercial LLMs (GPT-5) as LLM-judge, which might entail model/version drift risk since neither the API nor any specific snapshot is guaranteed to be available forever. Given this, I am wondering whether open-source models (e.g. Qwen series) are able to serve this and how the evaluation results will vary (e.g., will the scores differ drastically, or will there be any bias?).


4. Lack of discussion with related work. For example, Medical-Diff-VQA [1] , ICG-CXR [2], MedFrameQA [3] are not mentioned in Table 1 and the manuscript, although they explicitly feature in visual reasoning with multiple imaging studies from the same patient.

[1] Expert Knowledge-Aware Image Difference Graph Representation Learning for Difference-Aware Medical Visual Question Answering (KDD 2023)

[2] Towards Interpretable Counterfactual Generation via Multimodal Autoregression (MICCAI 2025)

[3] MedFrameQA: A Multi-Image Medical VQA Benchmark for Clinical Reasoning (arXiv 2025.05)

---

Below are minor issues:
- Each abbreviation should be expanded when it first appears for better readability. For example, “QA” (question-answering) in Line 010 and Line 035; “MCQ” (multiple-choice question) in Line 104.
- In Lines 362--363: “Table shows representative model accuracy on the held-out test set.” Table index seems missing.
- The prompts in the appendices (Secs. E, F, and G) are overflowing the right margin. Enabling automatic hyphenation or manually inserting hyphens may fix these issues.
- I suggest copying the “without SFT” results in Fig. 3 to Tab. 8 which presents model performance after SFT, or combine these results in one single figure. In this way, the readers will see the value of the curated training examples more clearly.
- I suggest adding a graph showing the imaging modality distribution, instead of plain description in Sec. I.

**Questions:**

Do cases in MedThinkVQA include redundant images (e.g. imaging study that does not provide valid information, or sometimes even contradictory information)? This aligns more to the practice where clinical users would not always do an image pre-filtering for the MLLM assistant. If that is the case, will such data produce a robust MLLM when used for model training and test models to ignore noisy information when used for model evaluation?

---

> ### Author Response · Authors · 2025-11-27
> **Response to Reviewer prRm**
>
> We thank the reviewer for the helpful comments. Below we first address the question on expert annotation / redundant images and robustness, then respond to the remaining points. In addition, all major revisions have been summarized in the **General Comment (Guide for Paper Revision)**, where you can also see how we incorporated the rebuttal responses into the updated manuscript.
>
> **Question: Redundant images, expert audit, and robustness.**
>
> Yes, MedThinkVQA includes some redundant images, but expert audit shows that the vast majority of views are supportive and contradictory information is extremely rare. In the newly added **Sec. 4.4 “Experts Performance and Data Quality Annotation”** and **Table 4**, two board-certified clinicians audited **96 test cases / 778 images: 65/96 cases** (463 images) had all views supportive (100%), while in the remaining 31 cases, **222/315 images (70.48%)** were supportive. Overall, **685/778 images (88.05%)** were rated supportive, with only ~12% redundant, and only **2/96 cases (2.1%)** flagged as possibly inconsistent. To directly test whether extra images help or just add noise, we added a new **image-subsampling experiment in Appendix B, Fig. 5**: for several models (MedGemma-27B, GPT-5-nano/mini/full), accuracy increases monotonically as the visible image ratio grows from 0 to 1.0. All models start from 1/5-random (20%) when no images are shown. Combined, the expert audit and Fig. 5 indicate that additional views are rarely “pure noise” and generally carry useful diagnostic evidence rather than pure noise. At the same time, a realistic minority of non-supportive views remains, so MedThinkVQA both reflects clinical practice (radiologists scrolling through many images and selecting key views) and encourages models to integrate multiple supportive images while learning to down-weight redundant ones during training and evaluation.
>
>
> | Model / Expert | Correct | Incorrect | ACC (%) |
> | -------------- | ------- | --------- | ------- |
> | Human experts  | 74      | 22        | 77.10   |
> | Gemini-2.5-pro | 54      | 42        | 55.67   |
> | GPT-5          | 53      | 43        | 55.21   |
> | Claude 4.0     | 47      | 49        | 48.96   |
> | Lingshu-32B    | 42      | 54        | 43.75   |
>
> Table 1: Human expert baseline vs. VLMs on the same 96-case subset
>
> | Case group                   | #Cases | #Images | #Supportive imgs | Supportive ratio (%) |
> | ---------------------------- | ------ | ------- | ---------------- | -------------------- |
> | All images supportive        | 65     | 463     | 463              | 100.00               |
> | Mixed supportive / redundant | 31     | 315     | 222              | 70.48                |
> | **Overall (96 cases)**       | **96** | **778** | **685**          | **88.05**            |
>
> Table 2: Round 2 expert audit: image-level redundancy vs. support
>
>
> **Weakness 1: Dataset statistics (6.51 vs 8.3).**
>
> We thank the reviewer for catching this inconsistency. In the revised manuscript we corrected **Sec. 3.1** to report “a multi-image set (average **6.51 images per case**)”, which is now consistent with the **Abstract** and **Table 1**. The higher figure applies only to the test split: **Table 2** reports that the 751-case test set has **8.11 images per sample on average**, since we deliberately made the held-out set more image-dense for evaluation.

---

> > ### Author Response · Authors · 2025-11-27
> > **Response to Reviewer prRm**
> >
> > **Weakness 2: How multiple images are gathered; multimodal vs longitudinal; Separate Analyses.**
> >
> > In **Sec. 3.1 “Source Corpus”** and **Appendix C/J/K**, we clarify that each MedThinkVQA case corresponds to a single Eurorad teaching case for one real patient, combining all images from the same clinical episode: different views/slices of one modality, complementary modalities (e.g., X-ray+CT+MRI+histology), longitudinal follow-up studies, or their combinations. Modality statistics are now detailed in **Appendix J, Tables 10–11** (13 aggregated modalities in test; mean **2.13 modalities/case** with about 30.6% of test cases having ≥3 modalities), and longitudinal prevalence is given in **Sec. 3.2** and **Appendix K, Table 12** (~25.5% of all cases are longitudinal follow-up). To address the reviewer’s suggestion, we separately analyze highly multi-modal cases (≥3 modalities) and longitudinal cases in **Sec. 5.2 and Fig. 7**. Accuracy on highly multi-modal cases fluctuates around each model’s overall accuracy (some models slightly improve, some slightly drop), whereas accuracy on longitudinal cases consistently decreases for most models, indicating that temporal trajectory reasoning is particularly challenging. Qualitative error analyses for a multimodal hydatid case and a longitudinal cystic TB case are included in **Appendix C.4.3 and C.4.2**.
> >
> > | Split | #Cases | #Images | CT+MRI (%) | Ultrasound (%) | X-ray (%) | Other modalities (%) | Avg modalities / case | % cases with ≥3 modalities |
> > | ----- | ------ | ------- | ---------- | -------------- | --------- | -------------------- | --------------------- | -------------------------- |
> > | Train | 7,729  | 49,159  | 72.9       | 8.32           | 7.74      | 11.06                | 1.84                  | 20.6                       |
> > | Test  | 751    | 6,090   | 76.6       | 8.29           | 6.54      | 8.55                 | 2.13                  | 30.6                       |
> >
> > Table 3. Multimodal imaging statistics in MedThinkVQA (train vs. test)
> >
> > | Split   | #Cases | #Longitudinal cases | Share (%) |
> > | ------- | ------ | ------------------- | --------- |
> > | Train   | 7,729  | 1,947               | 25.2      |
> > | Test    | 751    | 212                 | 28.2      |
> > | Overall | 8,480  | 2,159               | 25.5      |
> >
> > Table 4. Prevalence of longitudinal follow-up studies in MedThinkVQA
> >
> > **Weakness 3: LLM-judge choice, drift, and open-source judges.**
> >
> > GPT-5 is our main LLM-judge, with prompts and protocols documented in Sec. 4.2–4.3 and Appendix H. Its reliability is supported by a human study (Sec. 4.3; Appendix P, Table 16): on 202 steps from 50 cases, Cohen’s kappa for step factuality was 0.82 (Expert1 vs Expert2), 0.84 (Expert1 vs GPT-5), and 0.70 (Expert2 vs GPT-5), indicating that GPT-5 is at least as consistent with one expert as the experts are with each other. We acknowledge that relying on a commercial model introduces model- and prompt-dependence, and we now state this explicitly in our Limitations section: neither the GPT-5 API nor any particular snapshot is guaranteed to remain available, so future updates could shift judgments even under identical prompts. To explore open-weight judges, we additionally tested **Qwen2.5-VL** models from **3B to 72B** parameters on a subset and observed only modest agreement with experts (overall agreement just above **0.7** and kappa about **0.4**), and these models frequently failed to produce valid structured JSON, making them difficult to use as reliable automatic judges at scale. Even after small-scale distillation on a subset of GPT-5–labeled steps, these open-weight judges did **not** match GPT-5’s stability and alignment with clinicians, so they are not yet a drop-in replacement for our main commercial judge. For this work, we therefore retain GPT-5 as the primary judge, but the evaluation pipeline itself is judge-agnostic: we release all prompts and scoring scripts, and the step-level labels used in our analysis are stored and can be reused without calling the API. Looking ahead, we view it as an important direction to build fully open-source, reasoning-centered evaluation pipelines, for example by leveraging next-generation **reasoning VLMs** and ensembles of open-weight judges calibrated with human audits and process-level supervision.

---

> > > ### Author Response · Authors · 2025-11-27
> > > **Response to Reviewer prRm**
> > >
> > > **Weakness 4: Related work**
> > >
> > > We have significantly expanded **Table 1** and **Sec. 2 “Related Work”** to explicitly include **Medical-Diff-VQA** [1], **ICG-CXR** [2], and **MedFrameQA** [3]. Medical-Diff-VQA and ICG-CXR both focus on chest X-ray difference/counterfactual reasoning with mostly image pairs and largely non-expert, pipeline-generated QAs, rather than full expert step traces tied to clinical cases [1,2]. MedFrameQA is multi-image, but is constructed from YouTube medical education videos: images are key frames from a single video clip, so a given question typically lacks genuine multi-modality (e.g., CT+MRI+US) and does not model multi-timepoint follow-up; in addition, its questions and rationales are generated from noisy video transcripts rather than peer-reviewed, per-case expert annotations, which raises data quality concerns compared to curated teaching cases [3]. Beyond these three, we also added several recent benchmarks for a more complete comparison: **GEMeX** [4] and **GEMeX-ThinkVG** [6] provide large-scale chest X-ray VQA with grounding and step-style explanations, but are largely single-image and lack multi-specialty, multi-modality, and longitudinal structure; **MedRAX/ChestAgentBench** [5] offers an agent-style evaluation on Eurorad chest X-rays but only releases images and questions, without the underlying expert “think-with-images” traces; and **S-Chain** [7] contributes structured visual chain-of-thought, again mostly at the single-image level without full clinical scenarios or multi-image case synthesis. In contrast, as highlighted in **Table 1** and **Sec. 3**, **MedThinkVQA** uniquely combines expert-authored per-image findings, case-level imaging summaries, option-aligned differential reasoning, multi-image / multi-modality / longitudinal cases, and a beyond-accuracy evaluation suite (step metrics, error tags, and education-value scoring).
> > >
> > > **References**
> > >
> > > [1] X. Hu, L. Gu, Q. An, M. Zhang, L. Liu, K. Kobayashi, T. Harada, R. M. Summers, and Y. Zhu. “Expert Knowledge-Aware Image Difference Graph Representation Learning for Difference-Aware Medical Visual Question Answering.” *KDD*, 2023.
> > >
> > > [2] C. Ma, Y. Ji, J. Ye, L. Zhang, Y. Chen, T. Li, M. Li, J. He, and H. Shan. “Towards Interpretable Counterfactual Generation via Multimodal Autoregression.” *MICCAI*, 2025.
> > >
> > > [3] S. Yu, H. Wang, J. Wu, C. Xie, and Y. Zhou. “MedFrameQA: A Multi-Image Medical VQA Benchmark for Clinical Reasoning.” arXiv:2505.16964, 2025.
> > >
> > > [4] B. Liu, K. Zou, L. Zhan, Z. Lu, X. Dong, Y. Chen, C. Xie, J. Cao, X.-M. Wu, and H. Fu. “GEMeX: A Large-Scale, Groundable, and Explainable Medical VQA Benchmark for Chest X-Ray Diagnosis.” arXiv:2411.16778, 2025.
> > >
> > > [5] A. Fallahpour, J. Ma, A. Munim, H. Lyu, and B. Wang. “MedRAX: Medical Reasoning Agent for Chest X-Ray.” arXiv:2502.02673, 2025.
> > >
> > > [6] B. Liu et al. “GEMeX-ThinkVG: Towards Thinking with Visual Grounding in Medical VQA via Reinforcement Learning.” arXiv:2506.17939, 2025.
> > >
> > > [7] K. Le-Duc, D. M. H. Nguyen, P. T. H. Trinh, T.-P. Nguyen, N. T. Diep, A. Ngo, T. Vu, T. Vuong, A.-T. Nguyen, M. Nguyen, V. T. Hoang, K.-N. Nguyen, H. Nguyen, C. Ngo, A. Liu, N. Ho, A.-C. Hauschild, K. X. Nguyen, T. Nguyen-Tang, P. Xie, D. Sonntag, J. Zou, M. Niepert, and A. T. Nguyen. “S-Chain: Structured Visual Chain-of-Thought for Medicine.” arXiv:2510.22728, 2025.

---

### Official Review · Reviewer_zQ3B · 2025-10-31

**Soundness:** 3
**Presentation:** 3
**Contribution:** 3
**Rating:** 6
**Confidence:** 4

**Summary:**

This paper introduces MedThinkVQA, a benchmark for evaluating multi-image diagnostic reasoning in medical imaging. The benchmark features 8,481 cases (751 test) averaging 6.51 images per case, sourced from Eurorad and expert-annotated with three-step supervision: (1) per-image findings, (2) case-level imaging summaries, and (3) differential diagnosis reasoning. This paper evaluates various VLMs (GPT-5, Qwen2.5-VL, MedGemma, InternVL) and find that current models struggle significantly (GPT-5: 57.39% accuracy), with the primary bottleneck being cross-image evidence extraction and integration rather than language reasoning. The benchmark includes beyond-accuracy evaluation with error-type tagging and medical education case discussion generation.

**Strengths:**

1. This paper presents the largest expert-annotated multi-image medical QA benchmark.
2. This paper presents the well-designed three-step evaluation framework that mirrors clinical diagnostic workflow.
3. This benchmark performs rigorous dataset curation with multiple quality control measures, such as, leakage detection, confusion-aware pruning.

**Weaknesses:**

Weakness:
1. Dataset relies entirely on Eurorad cases, potentially limiting generalizability despite broad coverage of radiology subspecialties.
2. The benchmark data sourced from Eurorad may have been included in the pre-training corpora of some evaluated models.
3. The case analysis is not presented, such as, failed case analysis.

**Questions:**

1.  How do you ensure that Eurorad cases haven't been seen during pre-training of evaluated models?
2. Only the train set (small scale dataset) is used to train the vlm? the detailed train (sft) strategy.
3. Why not compared with the expert radiologist?
4. The present question from benchmark is verified by human expert for quality control? If not, why?

---

> ### Author Response · Authors · 2025-11-27
> **Response to Reviewer zQ3B**
>
> We thank the reviewer for the insightful comments and respond point-by-point below. In addition, all major revisions have been summarized in the **General Comment (Guide for Paper Revision)**, where you can also see how we incorporated the rebuttal responses into the updated manuscript.
>
> **Question 3&4: Expert Performance and Expert-based Quality Control.**
>
> In the revised manuscript we now report a dedicated human-expert study in **Section 4.4 and Tables 3–4**. Two board-certified clinicians (a diagnostic radiologist and an academic surgeon) each solved 96 randomly sampled MedThinkVQA test items under the same MCQ protocol as the VLMs (clinical history + all images, five options), achieving 74/96 correct (77.10% accuracy), while the strongest VLMs on the same subset reach only **55–56%** (Gemini-2.5-Pro 55.67%, GPT-5 55.21%), showing that experienced specialists still clearly outperform current models on our benchmark (Section 4.4, Table 3). In a second audit round on the same 96 cases, with full access to captions, per-image findings, case-level imaging summary, teaching discussion, and ground-truth answer, the experts flagged only 2/96 items (2.1%) as possibly inconsistent and marked 18/96 (18.8%) as “very difficult”. At the image level, 65/96 cases (463/463 images) were judged to have all images supportive, and in the remaining 31 cases 222/315 images were supportive, for an overall **685/778 supportive images (88.05%; Table 4 and Fig. 5)**. Together with our human and LLM-judge evaluations of Medical Education Case Discussions (Section 4.3, Tables 14–16, Appendix H), this demonstrates that MedThinkVQA’s questions and educational texts are expert-verified, internally coherent, and contain a substantial fraction of genuinely hard, image-dependent cases.
>
> | Model / Expert | Correct | Incorrect | ACC (%) |
> | -------------- | ------- | --------- | ------- |
> | Human experts  | 74      | 22        | 77.10   |
> | Gemini-2.5-pro | 54      | 42        | 55.67   |
> | GPT-5          | 53      | 43        | 55.21   |
> | Claude 4.0     | 47      | 49        | 48.96   |
> | Lingshu-32B    | 42      | 54        | 43.75   |
>
> Table 1: Human expert baseline vs. VLMs on the same 96-case subset
>
> | Case group                   | #Cases | #Images | #Supportive imgs | Supportive ratio (%) |
> | ---------------------------- | ------ | ------- | ---------------- | -------------------- |
> | All images supportive        | 65     | 463     | 463              | 100.00               |
> | Mixed supportive / redundant | 31     | 315     | 222              | 70.48                |
> | **Overall (96 cases)**       | **96** | **778** | **685**          | **88.05**            |
>
> Table 2: Round 2 expert audit: image-level redundancy vs. support

---

> > ### Author Response · Authors · 2025-11-27
> > **Response to Reviewer zQ3B**
> >
> > **Weakness 1: Reliance on Eurorad and Generalizability.**
> >
> > MedThinkVQA is built entirely from Eurorad, but Eurorad itself is a peer-reviewed teaching repository curated by the European Society of Radiology and populated by radiologists from multiple institutions worldwide [1] (Section 3.1). (Eurorad) Each case is a real clinical examination with expert-written clinical history, per-image captions, case-level imaging summary, teaching note, and final diagnosis, which we map into our Clinical Scenario, per-image hints, Integrated Imaging Summary, options, and Medical Education Case Discussion fields (Section 3.1, Appendix C). Coverage analysis in Section 3.2 and Appendix O shows that our test set spans 20/22 ICD-10 chapters and includes 85 Orphanet-aligned rare-disease cases, and, from the imaging side, all major radiology subspecialties and 13 aggregated modalities with on average 2.13 modalities and 8.11 images per case, and **≈28% longitudinal studies (Table 2, Appendix J–K)**. Recent independent works (e.g., benchmarking open-source LLMs on 1,933 Eurorad case reports [2] and MedRAX / ChestAgentBench building chest-X-ray reasoning agents on Eurorad-derived MCQs [3]) use the same corpus as a realistic, diverse benchmark, which supports our choice. (Nature) We explicitly acknowledge in the Ethical Statement (Others) that all cases currently come from a single educational repository and that cross-institution distribution shifts are likely, and we view adding additional sources (other teaching archives and PACS cohorts) as a main direction for future extensions of MedThinkVQA.
> >
> > | Split | #Cases | #Images | CT+MRI (%) | Ultrasound (%) | X-ray (%) | Other modalities (%) | Avg modalities / case | % cases with ≥3 modalities |
> > | ----- | ------ | ------- | ---------- | -------------- | --------- | -------------------- | --------------------- | -------------------------- |
> > | Train | 7,729  | 49,159  | 72.9       | 8.32           | 7.74      | 11.06                | 1.84                  | 20.6                       |
> > | Test  | 751    | 6,090   | 76.6       | 8.29           | 6.54      | 8.55                 | 2.13                  | 30.6                       |
> >
> > Table 3. Multimodal imaging statistics in MedThinkVQA (train vs. test)
> >
> > | Split   | #Cases | #Longitudinal cases | Share (%) |
> > | ------- | ------ | ------------------- | --------- |
> > | Train   | 7,729  | 1,947               | 25.2      |
> > | Test    | 751    | 212                 | 28.2      |
> > | Overall | 8,480  | 2,159               | 25.5      |
> >
> > Table 4. Prevalence of longitudinal follow-up studies in MedThinkVQA
> >
> > **Question 1 & Weakness 2: Data Contamination and Potential Pre-training Exposure.**
> >
> > We cannot formally guarantee that no Eurorad text was ever seen during pre-training of proprietary models, but **we follow the MELD methodology** of Nori et al. [4] and its recent use in MedAgentsBench [5] and **perform** both dataset-level mitigation and black-box contamination analysis. Concretely, we apply a strict sliding-window MELD variant that computes normalized character-level Levenshtein similarity between each model’s generated answer and the withheld half of the question. Across seven diverse LLM/VLMs, MELD scores cluster around 20–24% with tight IQRs and no item exceeds the ≥50% “high-risk” threshold, with similar distributions for text-only LLMs and VLMs (**Section 5.4, Fig. 8, Appendix N**). In addition, Section 3.3, Fig. 4, Appendices E–F and I describe how we (i) remove cases where the clinical history explicitly reveals the diagnosis (35 items), (ii) discard ~611 text-solvable questions that three strong text-only LLMs can answer perfectly without images, and (iii) debias the option-length distribution by pruning another 664 items. Together with de-duplication and our plan to maintain a rolling test set based on newly curated Eurorad cases from the **most recent 6–12 months** (Ethical Statement), these measures substantially reduce the likelihood that high performance on MedThinkVQA can be achieved via simple memorization of Eurorad text.

---

> > > ### Author Response · Authors · 2025-11-27
> > > **Response to Reviewer zQ3B**
> > >
> > > **Question 2: Training Data Usage and SFT Strategy.**
> > >
> > > All supervised fine-tuning experiments use only the MedThinkVQA training split (7,730 cases); the 751-case test split is never used for training or model selection (Section 3.1). Appendix B (“Supervised Fine-tuning”) specifies that we fine-tune InternVL3.5-1B/2B/4B, MedGemma-4B-IT, Qwen2.5-VL-3B-Instruct, and Qwen2.5-VL-7B-Instruct with QLoRA (rank 8, α = 16, dropout 0.05) for 2 epochs, batch size 1 with 8 gradient-accumulation steps (effective batch size 8), AdamW with learning rate 2×10⁻⁴, cosine schedule, and 0.03 warm-up; within the training partition we reserve 10% as a validation split for SFT only. Under this fixed configuration, SFT substantially improves accuracy (e.g., **Qwen2.5-VL-7B-Instruct from 31.95% to 61.89%**), showing that the train split provides strong supervision for process-supervised multi-image reasoning without leaking test information (Appendix B, Table 13).
> > >
> > > **Weakness 3: Failed Case Analysis and Typical Error Modes.**
> > >
> > > To address the request for case-level analysis, **Appendix C** now includes seven detailed case studies spanning multimodal, longitudinal, and failure scenarios. We show, for example, a 4-modality pelvic oncology case requiring integration of endoscopy, CT, MRI, and histology (**Appendix C.1**); a longitudinal cystic pulmonary tuberculosis case followed over ~11 months (**Appendix C.2**); a multi-syndrome vascular compression case combining SMA (Wilkie), Nutcracker, and May–Thurner syndromes (**Appendix C.4.1**); a disseminated hydatid disease case with brain, lung, and liver involvement (**Appendix C.4.3**); and a step-by-step breast imaging case used for granular error annotation (**Appendix C.5**). Across these examples we demonstrate that strong VLMs like GPT-5 often solve single-image or single-organ aspects correctly but fail to integrate evidence across images, modalities, or time points, in line with the aggregate step-level error statistics in **Section 5.2 and Appendix M**, where >60–70% of erroneous steps involve image-understanding errors and cross-view fusion rather than pure language reasoning.
> > >
> > > ---
> > >
> > > **References**
> > >
> > > [1] Eurorad – Radiology Teaching Cases. European Society of Radiology. Available at: [https://www.eurorad.org/](https://www.eurorad.org/) (accessed 2025). (Eurorad)
> > >
> > > [2] Su Hwan Kim, Severin Schramm, Lisa C. Adams, Rickmer Braren, Keno K. Bressem, Matthias Keicher, Paul-Sören Platzek, Karolin Johanna Paprottka, Claus Zimmer, Dennis M. Hedderich, et al. “Benchmarking the diagnostic performance of open source LLMs in 1,933 Eurorad case reports.” *npj Digital Medicine* 8, 97 (2025). (Nature)
> > >
> > > [3] Adibvafa Fallahpour, Jun Ma, Alif Munim, Hongwei Lyu, Bo Wang. “MedRAX: Medical Reasoning Agent for Chest X-ray.” arXiv:2502.02673, 2025. (arXiv)
> > >
> > > [4] Harsha Nori, Nicholas King, Scott Mayer McKinney, Dean Carignan, Eric Horvitz. “Capabilities of GPT-4 on Medical Challenge Problems.” arXiv:2303.13375, 2023. (arXiv)
> > >
> > > [5] Xiangru Tang, Daniel Shao, Jiwoong Sohn, Jiapeng Chen, Jiayi Zhang, Jinyu Xiang, Fang Wu, Yilun Zhao, Chenglin Wu, Wenqi Shi, Arman Cohan, Mark Gerstein. “MedAgentsBench: Benchmarking Thinking Models and Agent Frameworks for Complex Medical Reasoning.” arXiv:2503.07459, 2025. (arXiv)

---

### Official Review · Reviewer_8tFF · 2025-11-01

**Soundness:** 3
**Presentation:** 3
**Contribution:** 3
**Rating:** 6
**Confidence:** 3

**Summary:**

This paper introduces MedThinkVQA, a large-scale benchmark explicitly designed to evaluate multimodal large language models (MLLMs) in multi-image diagnostic reasoning. Built from 8,481 expert-curated teaching cases (average 6.5 images per case), it formalizes a three-step workflow that mirrors clinical reasoning: (1) per-image findings, (2) cross-view imaging summary, and (3) differential-diagnosis reasoning, followed by a medical-education discussion task.
Evaluation goes beyond simple accuracy by introducing step-level correctness, error-type tagging, and educational-value scoring. Baseline experiments on diverse MLLMs, show that even the best model reaches only 57.4 % accuracy, revealing that cross-image integration is the major bottleneck for current medical VLMs.

**Strengths:**

This is a high-quality benchmark paper with clear motivation and substantial novelty. It moves beyond single-image VQA toward multi-image, step-supervised diagnostic reasoning, something not covered by prior datasets such as OmniMedVQA or MedXpertQA. The design is conceptually elegant mirroring how clinicians think and technically meticulous, from option-wise pruning and leakage checks to fine-grained error analysis and human-LLM judge validation (κ≈0.8).
The beyond-accuracy evaluation framework, including RadCliQ metrics and teaching-discussion scoring, sets a new standard for interpretable benchmarking in medical AI. The authors also demonstrate strong ethical and reproducibility practices, releasing code, annotation scripts, and bias audits. Overall, the work is novel, comprehensive, and clinically grounded, offering a meaningful step toward trustworthy multimodal reasoning in medicine.

**Weaknesses:**

While the dataset is impressively detailed, it is built entirely from Eurorad cases, which may bias the distribution toward educational rather than real clinical imaging; cross-institutional validation or inclusion of temporal cases would further strengthen robustness.
The current evaluation focuses on four families of models and could benefit from broader comparisons to recent foundation-level MLLMs such as Gemini 2.5 Pro or Claude 3.5 Sonnet to fully situate difficulty. The study identifies the image-fusion bottleneck clearly but offers relatively limited prescriptive insight—there is little discussion of architectural directions or training strategies that could overcome this barrier.

**Questions:**

See above.

---

> ### Author Response · Authors · 2025-11-27
> **Response to Reviewer 8tFF**
>
> We thank the reviewer for the insightful comments and address each point in detail below. In addition, all major revisions have been summarized in the **General Comment (Guide for Paper Revision)**, where you can also see how we incorporated the rebuttal responses into the updated manuscript.
>
> **Weakness: Data Source and Representativeness:**
>
> Although MedThinkVQA is constructed from Eurorad, each case is based on a real clinical examination contributed by radiologists and researchers worldwide and peer-reviewed by the Eurorad Editorial Board (Sec. 3.1; Fig. 2), rather than synthetic teaching images. Eurorad itself is operated by the European Society of Radiology and has been described as “the largest database for peer-reviewed radiological case reports” [15], and recent studies explicitly treat Eurorad as a clinically realistic and challenging corpus: Kim et al. benchmarked 15 LLMs plus GPT-4o on 1,933 Eurorad case reports and showed that results closely track performance on an independent tertiary-hospital MRI cohort [13], while MedRAX/ChestAgentBench builds 2,500 diagnostic questions from 675 Eurorad chest X-ray cases to evaluate real-world agentic reasoning [14]. In MedThinkVQA we further quantify diversity: the test set covers 20/22 ICD-10 chapters plus 85 rare diseases, and spans all major radiology subspecialties and 13 imaging modalities, with an average of 8.11 images and 2.13 modalities per test case (Sec. 3.2; Tab. 2; Appx. J, O). About one quarter of all cases are longitudinal follow-up studies (test: 28.2%; Tab. 2; Appx. K), and Fig. 7 plus the temporal case studies in Appx. C.4.2 show that increasing time points systematically makes current MLLMs fail, directly addressing the reviewer’s suggestion to include temporal/serial imaging. We do acknowledge in the Ethics/Limitations section that all data originate from a single educational repository and that true cross-institution clinical validation remains future work, and we therefore commit to maintaining a rolling held-out test set that continually incorporates the most recent 6–12 months of newly curated Eurorad cases, along with public de-duplication and split scripts to limit leakage and “memorized teaching-set” effects (Ethical Statement; Sec. 5.4).
>
>
>
> | Split | #Cases | #Images | CT+MRI (%) | Ultrasound (%) | X-ray (%) | Other modalities (%) | Avg modalities / case | % cases with ≥3 modalities |
> |-------|--------|---------|------------|----------------|-----------|-----------------------|------------------------|----------------------------|
> | Train | 7,729  | 49,159  | 72.9       | 8.32           | 7.74      | 11.06                 | 1.84                   | 20.6                       |
> | Test  |   751  |  6,090  | 76.6       | 8.29           | 6.54      |  8.55                 | 2.13                   | 30.6                       |
>
>
> Table 1. Multimodal imaging statistics in MedThinkVQA (train vs. test)
>
>
> | Split   | #Cases | #Longitudinal cases | Share (%) |
> |---------|--------|----------------------|-----------|
> | Train   | 7,729  | 1,947                | 25.2      |
> | Test    |   751  |   212                | 28.2      |
> | Overall | 8,480  | 2,159                | 25.5      |
>
>
> Table 2. Prevalence of longitudinal follow-up studies in MedThinkVQA

---

> > ### Author Response · Authors · 2025-11-27
> > **Response to Reviewer 8tFF**
> >
> > **Weakness: Model Coverage.**
> >
> > We have extended the evaluation beyond the four original families. Figure 3 now includes **Gemini 2.5 Pro** (60.8% accuracy) and **Claude Sonnet** variants (Claude 3.7 Sonnet: 46.1%; Claude 4.0 Sonnet: 46.9%), as well as several recent medical vision–language models, including **Lingshu-7B/32B** and **HuatuoGPT-Vision-7B** (Sec. 5.1–5.2; Fig. 3). These results show that even the strongest frontier MLLMs remain substantially below human experts on MedThinkVQA. To better anchor the human upper bound and data quality, Sec. 4.4 adds a two-round study with **two board-certified clinicians** (a 7-year diagnostic radiologist and a 5-year academic surgeon): on a random 96-case subset, experts achieve 77.10% accuracy, outperforming GPT-5 (55.21%), Gemini-2.5-Pro (55.67%), Claude 4.0 (48.96%), and Lingshu-32B (43.75%), while also auditing case consistency and image redundancy (Tabs. 3–4). Experts flagged only 2/96 cases as possibly inconsistent and judged 88.05% of the 778 images as supportive for the final diagnosis, confirming that the benchmark is both coherent and genuinely multi-view rather than dominated by redundant images. We will continue to add results for newly released general and medical MLLMs in the camera-ready version; the updated figure already reflects our latest experiments, and the core conclusion—that strong models are still >20 points below experts on multi-image diagnostic reasoning—remains unchanged.
> >
> >
> >
> > | Model / Expert | Correct | Incorrect | ACC (%) |
> > |----------------|---------|-----------|---------|
> > | Human experts  | 74      | 22        | 77.10   |
> > | Gemini-2.5-pro | 54      | 42        | 55.67   |
> > | GPT-5          | 53      | 43        | 55.21   |
> > | Claude 4.0     | 47      | 49        | 48.96   |
> > | Lingshu-32B    | 42      | 54        | 43.75   |
> >
> > Table 3: Human expert baseline vs. VLMs on the same 96-case subset
> >
> >
> > | Case group                   | #Cases | #Images | #Supportive imgs | Supportive ratio (%) |
> > |------------------------------|--------|---------|------------------|----------------------|
> > | All images supportive        | 65     | 463     | 463              | 100.00               |
> > | Mixed supportive / redundant | 31     | 315     | 222              | 70.48                |
> > | **Overall (96 cases)**       | **96** | **778** | **685**          | **88.05**            |
> >
> > Table 4: Round 2 expert audit: image-level redundancy vs. support

---

> > > ### Author Response · Authors · 2025-11-27
> > > **Response to Reviewer 8tFF**
> > >
> > > **Weakness: Limited Prescriptive Insight on Overcoming the Image-Fusion Bottleneck.**
> > >
> > > We have substantially expanded **Sec. 5.2** and Appendices B, C, E, H to lay out concrete training and architectural directions. Empirically, MedThinkVQA isolates the bottleneck to image reading and cross-view fusion: expert Integrated Imaging Summaries boost accuracy by 18–31 points across models, while self-generated image text often yields small or negative gains (Sec. 5.2; Fig. 6, Tab. 6). Supervised fine-tuning on our curated training split allows compact open models (e.g., Qwen2.5-VL-7B/3B, InternVL3.5-4B, MedGemma-4B-IT) to reach or surpass GPT-5 (up to 61.89% accuracy; Appx. B, Tab. 13), supporting **process-level supervision** as a practical remedy. We now explicitly connect this to recent work on chain-of-thought and medical reasoning supervision [1–5], and propose to use MedThinkVQA’s per-image findings, integrated case summaries, and option-wise eliminations as step labels for process-supervised SFT and distillation, so that models are trained to “read each view → fuse across views → perform option-grounded DDx” rather than only to output final answers. Building on our error taxonomy and supportive-vs.-redundant image audits (Sec. 4.3–4.4; Appx. C–D, M), we also outline **data-centric and alignment strategies**: radiology-aware and counterfactual data augmentation [6,7], combined with preference-based objectives such as DPO/MIA-DPO [8,9] and RL-style process rewards [10,11] that explicitly favor reasoning chains which correctly exploit supportive views and penalize shortcut solutions. Finally, we argue for combining **test-time search methods** like Tree-of-Thoughts [12] with new multi-image architectures—hierarchical or retrieval-style encoders, sparse “image pointer” layers, and view-selection heads—so that the language backbone can dynamically pull only the necessary image tokens at each reasoning step; the multimodal and longitudinal error analyses in Appx. C.4.2–C.4.3 and the subset results in Fig. 7 are intended as seeds and metrics for these designs. We hope these additions clarify that MedThinkVQA is not only diagnosing the current image-fusion bottleneck, but also providing concrete, well-aligned supervision and evaluation hooks for future training and architectural work aimed at overcoming it.
> > >
> > >
> > > **References**
> > >
> > > [1] Zhang, X., Du, C., Pang, T., Liu, Q., Gao, W., & Lin, M. “Chain of Preference Optimization: Improving Chain-of-Thought Reasoning in LLMs.” NeurIPS, 2024.
> > >
> > > [2] Gai, X., Zhou, C., Liu, J., Feng, Y., Wu, J., & Liu, Z. “MedThink: A Rationale-Guided Framework for Explaining Medical Visual Question Answering.” Findings of ACL: NAACL 2025, 7438–7450, 2025.
> > >
> > > [3] Liu, J., Wang, Y., Du, J., Zhou, J. T., & Liu, Z. “MedCoT: Medical Chain of Thought via Hierarchical Expert.” EMNLP 2024, 17371–17389, 2024.
> > >
> > > [4] Le-Duc, K. et al. “S-Chain: Structured Visual Chain-of-Thought For Medicine.” arXiv:2510.22728, 2025.
> > >
> > > [5] Liu, B. et al. “GEMeX-RMCoT (GEMeX-ThinkVG).” ACM Multimedia 2025; arXiv:2506.17939, 2025.
> > >
> > > [6] Kebaili, A., Lapuyade-Lahorgue, J., & Ruan, S. “Deep Learning Approaches for Data Augmentation in Medical Imaging: A Review.” Journal of Imaging, 9(4):81, 2023.
> > >
> > > [7] Shoer, B., & Kementchedjhieva, Y. “A Simple Data Augmentation Strategy for Text-in-Image Scientific VQA.” WiNLP 2025, 100–105, 2025.
> > >
> > > [8] Rafailov, R. et al. “Direct Preference Optimization: Your Language Model is Secretly a Reward Model.” NeurIPS, 2023.
> > >
> > > [9] Liu, Z. et al. “MIA-DPO: Multi-Image Augmented Direct Preference Optimization For Large Vision-Language Models.” arXiv:2410.17637, 2024.
> > >
> > > [10] Raschka, S. “The State of Reinforcement Learning for LLM Reasoning: Understanding GRPO and New Insights from Reasoning Model Papers.” Ahead of AI, 2025.
> > >
> > > [11] DeepSeek-AI (D. Guo et al.). “DeepSeek-R1: Incentivizing Reasoning Capability in LLMs via Reinforcement Learning.” arXiv:2501.12948, 2025.
> > >
> > > [12] Yao, S. et al. “Tree of Thoughts: Deliberate Problem Solving with Large Language Models.” NeurIPS, 2023.
> > >
> > > [13] Kim, S. H. et al. “Benchmarking the diagnostic performance of open source LLMs in 1933 Eurorad case reports.” npj Digital Medicine, 8:97, 2025.
> > >
> > > [14] Fallahpour, A., Ma, J., Munim, A., Lyu, H., & Wang, B. “MedRAX: Medical Reasoning Agent for Chest X-ray.” ICML 2025; arXiv:2502.02673, 2025.
> > >
> > > [15] European Society of Radiology. “Eurorad – The largest database for peer-reviewed radiological case reports.” ESR / Eurorad official website, accessed 2025.

---

### Author Response · Authors · 2025-11-27
**Reading Guide to the Revised Manuscript**

**Main Text**

1. **Line 040-049, 080-102, 319-329, 353-355 (Reviewer qXc8)**: We make the introduction fully high-level (motivation and contributions only) and move all technical details on dataset construction and evaluation to later sections, responding to the question/weakness on overall paper structure.

2. **Line 108-130, 146-152, 162-166, 170-174 (Reviewer prRm)**: We substantially expand Table 1 to include Medical-Diff-VQA, ICG-CXR, MedFrameQA and other recent benchmarks, and add columns for multi-modality and longitudinal cases, responding to the weakness on limited related work and the question about how multiple images are gathered (multimodal vs longitudinal).

3. **Line 190 (Reviewer prRm)**: We correct the reported average number of images per case to 6.51, responding to the weakness about inconsistent image-count statistics between Section 3.1 and Table 1.

4. **Line 204-210 (Reviewer prRm & Reviewer zQ3B)**: We add explicit statistics for multimodal and longitudinal cases in the dataset, responding to questions about how multiple images are gathered (multimodal vs longitudinal) and requests for clearer case-level characterization.

5. **Line 238-241 (Reviewer prRm)**: We highlight the proportion of longitudinal follow-up cases and point to detailed statistics in the appendix, responding to the question on how multiple images are gathered and distinguished as multimodal versus longitudinal.

6. **Line 298-306 (Reviewer qXc8)**: We clarify the clinical motivation and practical value of the Medical Education Case Discussion task, responding to the weakness about the teaching-note generation use-case.

7. **Line 357-412 (Reviewer 8tFF & zQ3B & prRm & qXc8)**: We add a two-round human-expert study reporting diagnostic accuracy and image-level quality annotations, responding to concerns about baseline model coverage, expert performance and expert-based quality control, redundant images, multi-view dependence, and robustness.

8. **Line 458-467 (Reviewer 8tFF & zQ3B & qXc8 )**: We update Figure 3 to include human-expert performance and additional strong VLMs (e.g., Gemini, Claude, Lingshu), responding to weaknesses on baseline model coverage and concerns about expert performance.

9. **Line 468-479 (Reviewer prRm)**: We explicitly describe the human-expert baseline and new VLM scores shown in Figure 3 when discussing subset results, responding to the question about how multimodal and longitudinal cases influence model accuracy.

10. **Line 480-494 (Reviewer 8tFF & zQ3B)**: We propose concrete directions (process-supervised SFT, radiology-aware augmentation, preference/RL-style process rewards, and multi-image architectures with test-time “thinking”) to overcome the image-fusion bottleneck, responding to 8tFF Weakness “Limited prescriptive insight” and zQ3B Question 2 (how SFT and training can address current failures).

11. **Line 587–599 (Reviewer prRm)**: We state limitations of using GPT-5 as LLM-judge, note exploratory experiments with open-weight judges, and discuss remaining dataset limitations, responding to prRm Weakness 3 (judge drift and open-source alternatives).

---

> ### Author Response · Authors · 2025-11-27
> **Reading Guide to the Revised Manuscript (for Appendix)**
>
> **Appendix**
>
> 1. **Line 947–971 (Reviewer qXc8 & prRm)**: We add Figure 5 to show that accuracy rises monotonically with image_ratio, responding to qXc8 Weakness 3 (effect of removing images and necessity of multi-view) and prRm Question (redundant images vs robustness).
>
> 2. **Line 991–1049 (Reviewer prRm & zQ3B)**: We include Figures 7 and MELD boxplots to visualise multimodal vs longitudinal subsets, and data-leakage scores, responding to prRm Weakness 2 (multimodal/temporal breakdown), and zQ3B Weakness 2 & Question 1 (contamination analysis).
>
> 3. **Line 1026–1117 (Reviewer qXc8 & zQ3B)**: We add Appendix C’s example showing how one Eurorad case maps to all MedThinkVQA JSON fields, responding to qXc8 Weakness 1 & Question 1 (what the source sections are and how reasoning traces are derived) and zQ3B Weakness 3 (request for concrete case analysis).
>
> 4. **Line 1134–1241 (Reviewer prRm, zQ3B & qXc8)**: We provide a multimodal pelvic oncology case study (four modalities) to illustrate cross-modality supervision and model errors, responding to prRm Weakness 2 (how multiple images per case are gathered), zQ3B Weakness 3 (failed-case analysis), and qXc8 Weakness 3 (multi-view synthesis).
>
> 5. **Line 1242–1403 (Reviewer 8tFF, prRm)**: We describe a longitudinal cystic pulmonary tuberculosis case with multiple follow-up scans and model mistakes, responding to 8tFF Weakness (need for temporal cases), prRm Weakness 2 (separate consideration of longitudinal cases).
>
> 6. **Line 1620–1673 (Reviewer zQ3B & qXc8)**: We add a multi-syndrome vascular compression case (SMA, Nutcracker, May–Thurner) showing how the model fails to integrate dispersed clues, responding to zQ3B Weakness 3 (missing case analysis) and qXc8 Weakness 3 (strict multi-image reasoning).
>
> 7. **Line 1556–1781 (Reviewer zQ3B & prRm)**: We add 2 cases where the model over-focuses on one modality and make mistakes on longitudinal studies responding to zQ3B Weakness 3 and prRm Weakness 2 (desire for concrete multimodal failure examples).
>
> 8. **Line 1782–1889 (Reviewer qXc8 & prRm)**: We include a full step-level evaluation case (bilateral subareolar abscesses) illustrating step splitting, error labels, and critical-step marking, responding to qXc8 Weakness 2 & Question 2 (how steps and error categories are defined) and prRm Weakness 3 (transparency of the GPT-5 judge).
>
> 9. **Line 2268–2429 (Reviewer 8tFF, zQ3B & prRm)**: We add Appendix J–K with detailed modality distributions, per-case modality counts, common modality combinations, and longitudinal-case frequencies in train/test, responding to 8tFF Weakness “Data Source and Representativeness,” zQ3B Weakness 1 (dataset coverage), and prRm Weakness 1–2 (clarifying 6.51 vs 8.11 images/case and multi-image composition).

---

### Meta-Review · Area_Chair_rWFc · 2026-01-07

**Summary:**

Most reviewers are positive while the concerns focus mainly on representativeness of base dataset (Eurorad), a lack of human assessments, and clarity. These concerns are generally addressed in the rebuttal/revision.

**Reviewer Concerns:**

`8tFF`: biased distribution on educational cases (responded with background information on the dataset and related findings proving good representativeness); expecting broader comparisons (added in the rebuttal) and additional discussions on architectural / training solutions for image-fusion bottlenecks (discussed)

`zQ3B`: biased distribution on educational cases (responded with background information on the dataset and related findings proving good representativeness); contamination due to public dataset (mitigated with MELD methodology); expecting additional failure analysis (added to the appendix); expecting human evaluation on questions and outcomes (supplemented in the response).

`prRm`: experiment detail missing (supplemented), influence of API drift in LLM-judge (substantiated with human agreement tests), lack of discussions on related works (added in the response).

`qXc8`: structure and presentation, unfriendly to readers outside the domain (revised); Expecting clarifications on evaluation details (additional information provided); questioned on necessity for multi-image and lack of human performance; questions on utility (addressed with human assessments).

**Reviewer Scores:**

Questions raised by `qXc8` (clarity, structure, human evaluation and rationale for multi-view) are generally addressed in the rebuttal. Other 3 reviewers gave positive scores.

---

### Decision · Program_Chairs · 2026-01-26

Accept (Poster)